# Gene regulation by gonadal hormone receptors underlies brain sex differences

B. Gegenhuber[1,2], M. V. Wu[1], R. Bronstein[1] & J. Tollkuhn[1✉]

Oestradiol establishes neural sex differences in many vertebrates[1–3] and modulates mood, behaviour and energy balance in adulthood[4–8]. In the canonical pathway, oestradiol exerts its effects through the transcription factor oestrogen receptor-α (ERα)[9]. Although ERα has been extensively characterized in breast cancer, the neuronal targets of ERα, and their involvement in brain sex differences, remain largely unknown. Here we generate a comprehensive map of genomic ERα-binding sites in a sexually dimorphic neural circuit that mediates social behaviours. We conclude that ERα orchestrates sexual differentiation of the mouse brain through two mechanisms: establishing two male-biased neuron types and activating a sustained male-biased gene expression program. Collectively, our findings reveal that sex differences in gene expression are defined by hormonal activation of neuronal steroid receptors. The molecular targets we identify may underlie the effects of oestradiol on brain development, behaviour and disease.

In mammals, gonadal steroid hormones regulate sex differences in neural activity and behaviour. These hormones establish sex-typical neural circuitry during critical periods of development and activate the display of innate social behaviours in adulthood. Among these hormones, oestradiol is the principal regulator of brain sexual differentiation in mice. In males, the testes briefly activate at birth, generating a sharp rise in testosterone that subsides within hours[10]. Neural aromatase converts circulating testosterone to 17β-oestradiol, which acts through ERα in discrete neuronal populations to specify sex differences in cell number and connectivity[1,3,11]. Despite extensive characterization of the neural circuits controlling sex-typical behaviours[12,13], the underlying genomic mechanisms by which steroid hormone receptors act in these circuits remain unknown. Recent advancements in low-input and single-cell chromatin profiling methods have provided transformative insights into how transcription factors (TFs) regulate gene expression in small numbers of cells[14]. We set out to use these methods to discover the neuronal genomic targets of ERα and how they coordinate brain sexual differentiation.

## Genomic targets of ERα in the brain

To determine the genomic targets of ERα in the brain, we used an established hormone starvation and replacement paradigm that reproducibly elicits sex-typical behaviours[2] and replicates the medium conditions required to detect ERα genomic binding in cell lines[15]. At 4 h after treatment with oestradiol benzoate (E2) or vehicle control, we profiled ERα binding in three interconnected limbic brain regions in which ERα regulates sex-typical behaviours: the posterior bed nucleus of the stria terminalis (BNSTp), medial pre-optic area and posterior medial amygdala[11,12,16] (Fig. 1a). We used the low-input TF profiling method CUT&RUN, which we first validated in MCF-7 breast cancer cells by comparing to a previous dataset for chromatin immunoprecipitation with sequencing (ChIP–seq) of ERα (Extended Data Fig. 1).

We detected 1,930 E2-induced ERα-bound loci in the brain (Fig. 1b, Extended Data Fig. 2 and Supplementary Table 1). The most enriched TF-binding motif in these peaks was the oestrogen response element (ERE), the canonical binding site of oestrogen receptors (Extended Data Fig. 2c, d). Comparison of these ERα-binding sites to those previously detected in peripheral mouse tissues revealed that most are specific to the brain (Fig. 1c and Extended Data Fig. 2f). Brain-specific ERα binding events were uniquely enriched for synaptic and neurodevelopmental disease Gene Ontology terms, including neurotransmitter receptors, ion channels, neurotrophin receptors and extracellular matrix genes (Fig. 1d, Extended Data Fig. 2h–k and Supplementary Table 1). We also found evidence supporting direct crosstalk between oestradiol and neuroprotection, as ERα directly binds loci for the neurotrophin receptors *Ntrk2* (also known as *Trkb*) and *Ntrk3* (Extended Data Fig. 2k and Supplementary Table 1). Moreover, ERα targets the genes encoding androgen and progesterone receptors (*Ar* and *Pgr*; Supplementary Table 1).

To determine the effects of ERα binding on gene expression and chromatin state, we focused on a single brain region, the BNSTp, given its central role in the regulation of sex-typical behaviours. The BNSTp receives olfactory input through the accessory olfactory bulb and projects to the medial pre-optic area, medial amygdala, hypothalamus and mesolimbic reward pathway[11,17]. We used our oestradiol treatment paradigm and performed translating ribosome affinity purification (TRAP), followed by RNA sequencing (RNA-seq), on the BNSTp from *Esr1*[Cre/+];*Rpl22*[HA/+] mice, enabling selective capture of ribosome-bound transcripts from *Esr1*+ cells. We identified 358 genes regulated by oestradiol, including genes known to be induced by E2 in breast cancer, such as *Pgr* and *Nrip1* (Fig. 1e and Supplementary Table 2). We then validated several of these E2-regulated genes by in situ hybridization (Fig. 1f, Extended Data Fig. 3 and Extended Data Table 1). Genes that contribute to neuron wiring (*Brinp2*, *Unc5b* and *Enah*) and synaptic plasticity (*Rcn1* and *Irs2*) were robustly induced by oestradiol in the

[1]Cold Spring Harbor Laboratory, Cold Spring Harbor, NY, USA. [2]Cold Spring Harbor Laboratory School of Biological Sciences, Cold Spring Harbor, NY, USA. ✉e-mail: tollkuhn@cshl.edu

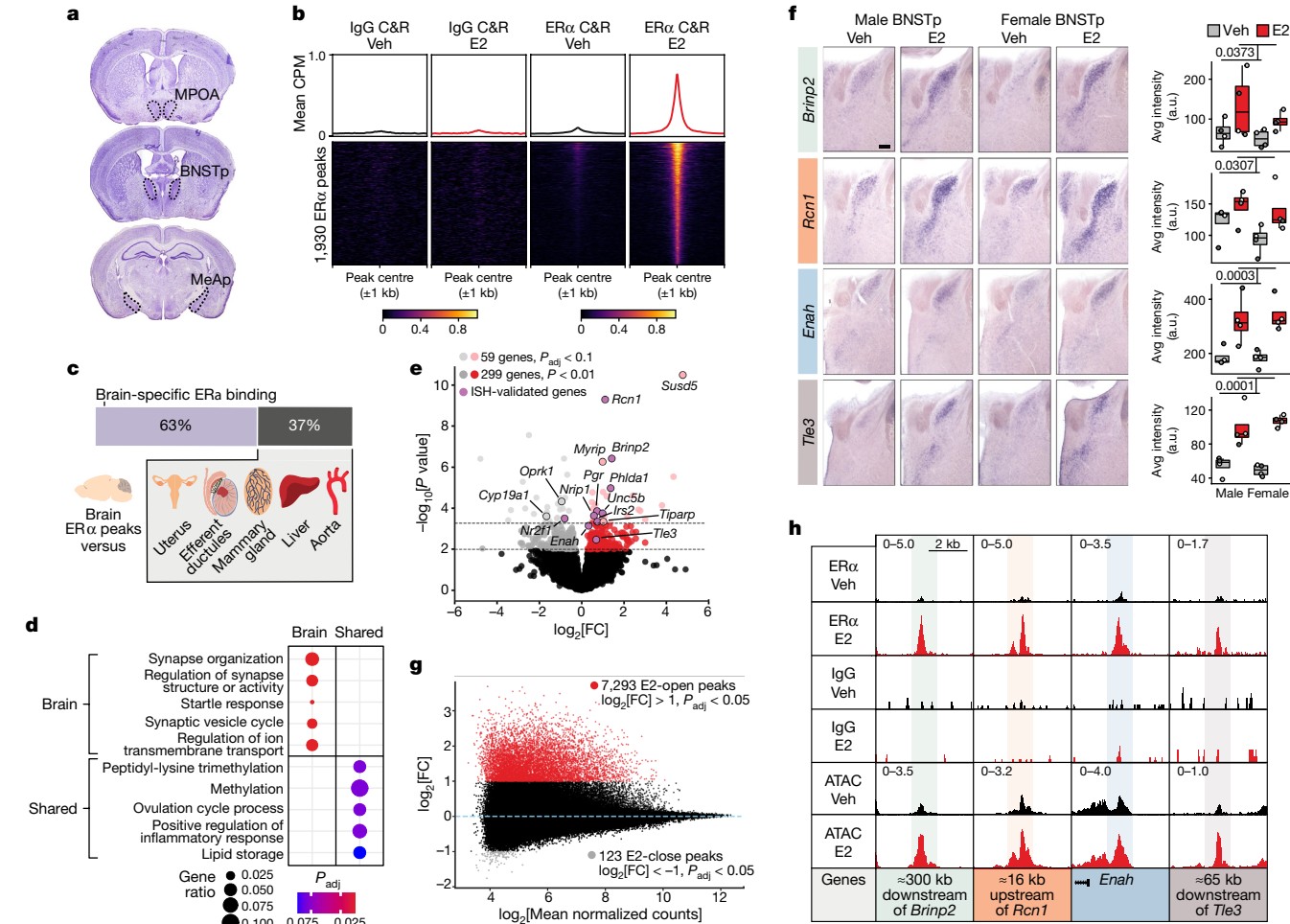

**Fig. 1 | Genomic targets of ERα in sexually dimorphic neuronal populations.**
**a**, Coronal sections containing sexually dimorphic brain areas used for ERα CUT&RUN. MPOA, medial pre-optic area; BNSTp, posterior bed nucleus of the stria terminalis; MeAp, posterior medial amygdala. **b**, Line plots (top) and heatmaps (bottom) of mean IgG and ERα CUT&RUN (C&R) CPM ±1 kb around E2-induced ERα CUT&RUN peaks (DiffBind edgeR, $P_{adj} < 0.1$). The heatmaps are sorted by E2 ERα CUT&RUN signal. Colour scale is counts per million (CPM). Veh, vehicle. **c**, Cross-tissue ERα comparison, showing the proportion of ERα peaks detected specifically in brain. **d**, Top Gene Ontology biological process terms associated with genes nearest to brain-specific or shared (≥4 other tissues) ERα CUT&RUN peaks (clusterProfiler, $P_{adj} < 0.1$). **e**, Combined sex E2 versus vehicle RNA-seq in BNSTp *Esr1*+ cells; light grey and red dots (DESeq2, $P_{adj} < 0.1$), dark grey and red dots (DESeq2, $P < 0.01$), purple dots (validated by in situ hybridization (ISH)). FC, fold change. Positive FC is E2-upregulated, negative FC is E2-downregulated. **f**, Images (left panels) and quantitative analysis (right panels) of ISH for select genes induced by E2 in both sexes. Boxplot centre, median; box boundaries, first and third quartiles; whiskers, 1.5 × IQR from boundaries. Two-way analysis of variance: *Brinp2* $P = 0.0373$, *Rcn1* $P = 0.0307$, *Enah* $P = 0.0003$, *Tle3* $P = 0.0001$; $n = 4$ per condition; scale bar, 200 μm. **g**, MA plot of E2-regulated ATAC–seq peaks in BNSTp *Esr1*+ cells; red dots are E2-open peaks (DiffBind edgeR, $\log_2[FC] > 1$, $P_{adj} < 0.05$), grey dots are E2-close peaks (DiffBind edgeR, $\log_2[FC] < -1$, $P_{adj} < 0.05$). **h**, Example ERα peaks at E2-induced genes. Top left number is the y-axis range in CPM. Shaded band indicates peak region.

BNSTp, illustrating how oestradiol signalling may sculpt sexual differentiation of BNSTp circuitry.

To identify oestradiol-responsive chromatin regions, which may involve signalling pathways other than direct ERα binding[18], we used our oestradiol treatment paradigm and performed assay for transposase-accessible chromatin with sequencing (ATAC–seq) on BNSTp *Esr1*+ cells collected from *Esr1*[Cre/+];*Sun1−GFP*[lx/+] mice. Across sexes, we detected 7,293 chromatin regions that increase accessibility with treatment (E2-open) as well as 123 regions that decrease accessibility (E2-close; Fig. 1g, Extended Data Fig. 4a–e and Supplementary Table 3). Motif enrichment analysis of these E2-open regions, which occurred primarily at distal enhancer elements (Extended Data Fig. 4c), showed that 89% contain an ERE (Extended Data Fig. 4f), consistent with the observation that nearly all ERα-binding sites overlapped an E2-open region (Extended Data Fig. 4g). These results indicate that direct oestrogen receptor binding, rather than indirect signalling pathways, drives most E2-responsive chromatin regions in the BNSTp[19].

After examining the relationship between oestradiol-regulated chromatin loci and gene expression, we noted that E2-open regions localized at both E2-upregulated and E2-downregulated genes (Extended Data Fig. 5a). E2-open regions at downregulated genes contained EREs yet lacked widespread ERα binding (Extended Data Fig. 5b, c), suggesting that transient ERα recruitment may contribute to gene repression[20]. E2-upregulated genes with corresponding E2-responsive chromatin loci include *Brinp2*, *Rcn1*, *Enah* and *Tle3* (Fig. 1h); E2-downregulated genes include *Astn2*, a regulator of synaptic trafficking, and *Nr2f1* (Extended Data Figs. 3 and 5d).

Although most oestradiol regulation events were shared between sexes in our treatment paradigm, we noted certain sex-dependent effects. Pairwise comparison by sex revealed nearly 300 differential genes between females and males in our TRAP RNA-seq data (Supplementary Table 2). Moreover, we observed 306 genes with a differential response to oestradiol between sexes (Extended Data Fig. 5e, f and Supplementary Table 2). These sex-dependent, E2-responsive genes lacked

enrichment of E2-responsive chromatin regions (Extended Data Fig. 5g), which may indicate further oestradiol regulation at the translational level[21]. Likewise, across ERα CUT&RUN and ATAC–seq modalities, we observed negligible sex differences and sex-dependent, E2-responsive loci (Extended Data Fig. 5h–j and Supplementary Table 3), demonstrating that females and males mount a similar genomic response to exogenous oestradiol on removal of the hormonal milieu.

## Sex differences in gene regulation

Across rodents and humans, the BNSTp of males is approximately 1.5–2 times larger than that of females[22,23]. In mice, this structural dimorphism arises from male-specific neonatal ERα activation, which promotes neuron survival[24,25]. Although BNSTp *Esr1*+ neurons are known to be GABAergic[16], the identity of male-biased GABAergic neuron types remains unclear. To characterize these cells, we reanalysed a single-nucleus RNA-seq (snRNA-seq) dataset collected from the BNST of adult, gonadally intact females and males[26]. Seven BNSTp *Esr1*+ transcriptomic neuron types emerged from this analysis, and two of these marked by *Nfix* (i1:Nfix) and *Esr2* (i3:Esr2) are more abundant in males than in females (Fig. 2a, b and Extended Data Fig. 6a, b). Although a male bias in *Esr2*/ERβ-labelled cells is known[27], Nfix expression has not been described previously in the BNSTp. Immunofluorescent staining confirmed that males have twice as many ERα+Nfix+ neurons as females (Fig. 2c and Extended Data Fig. 6c).

To interpret the functional relevance of BNSTp *Esr1*+ neuron types, we compared their gene expression profiles to the mouse cortical and hippocampal single-cell RNA-seq atlas using MetaNeighbor[28,29]. i1:Nfix neurons uniquely matched the identity of *Lamp5*+ neurogliaform interneurons[30,31] (Fig. 2d and Extended Data Fig. 6d, e) and also shared markers (*Moxd1* and *Cplx3*; Extended Data Fig. 6b, f, g) with a male-biased neuron type (i20:Gal/Moxd1) in the sexually dimorphic nucleus of the preoptic area (SDN-POA) that is selectively activated during male-typical mating, inter-male aggression and parenting behaviours[32]. Beyond these two genes, i1:Nfix and i20:Gal/Moxd1 neuron types share a transcriptomic identity, consistent with observed Nfix immunofluorescence in both the BNSTp and SDN-POA (Fig. 2e and Extended Data Fig. 6h). Together, these results define male-biased neurons in the BNSTp and reveal a common *Lamp5*+ neurogliaform identity between the BNSTp and SDN-POA.

We next examined sex differences in gene expression and found extensive and robust (false discovery rate < 0.1) sex-biased expression across BNST neuron types (Fig. 2f, Extended Data Fig. 7a–d and Supplementary Table 4). Most sex differences were specific to individual types (for example, *Dlg2*/PSD-93 and *Kctd16* in i1:Nfix neurons), whereas select differences were detected in multiple populations (for example, *Tiparp* and *Socs2*; Extended Data Fig. 7b, c). Relative to all other TF genes in the genome, *Esr1*, along with coexpressed hormone receptors, *Ar* and progesterone receptor (*Pgr*), correlated best with sex-biased gene expression (Fig. 2g and Extended Data Fig. 7e, f), indicating potential regulatory function.

To identify chromatin regions controlling sex differences in BNSTp gene expression, we performed ATAC–seq on BNSTp *Esr1*+ cells collected from gonadally intact *Esr1*^Cre/+;*Sun1−GFP*^lx/+ mice. Approximately 18,000 regions differed in accessibility between sexes; moreover, these regions localized at sex-biased genes detected in *Esr1*+ neuron types (Fig. 2h, Extended Data Fig. 7g, h and Supplementary Table 5). By contrast, gonadectomy reduced the number of sex-biased regions to 71 (Fig. 2h and Supplementary Table 5). We compared chromatin accessibility across sexes and gonadal hormone status using *k*-means clustering and discovered male-specific, but not female-specific, responses to gonadectomy (Fig. 2i and Extended Data Fig. 7i–k). Notably, chromatin regions that close specifically in males on gonadectomy (cluster 1) primarily contained the androgen response element, whereas regions closing across both sexes (cluster 2) were enriched for the ERE and

strongly overlapped E2-open regions (Fig. 2i, j). Thus, in the BNSTp, oestradiol maintains chromatin in an active state across both sexes, whereas testosterone promotes chromatin activation and repression in males. Collectively, these data indicate that gonadal hormone receptors drive adult sex differences in gene expression, largely as a consequence of acute hormonal state.

## ERα drives neonatal chromatin state

Sexual dimorphism in BNSTp wiring emerges throughout a 2-week window following birth, well after neural oestradiol has subsided in males. To determine the genomic targets of the neonatal surge, we performed ATAC–seq on BNSTp *Esr1*+ cells at postnatal day 4 (P4), which corresponds to the onset of male-biased BNSTp cell survival and axonogenesis[33,34]. We detected about 2,000 sex differences in chromatin loci at this time, and nearly all sex differences were dependent on neonatal oestradiol (NE; Fig. 3a, Extended Data Fig. 8a, b and Supplementary Table 6). NE-open regions were similarly induced by oestradiol in our adult dataset (Extended Data Fig. 8c, d). To determine whether ERα drives male-typical chromatin opening, we performed ERα CUT&RUN on *Esr1*+ cells from females treated acutely with vehicle or oestradiol on the day of birth. Oestradiol rapidly recruited ERα to NE-open regions (Fig. 3a, Extended Data Fig. 8e–h and Supplementary Table 7). Our results demonstrate that ERα activation controls neonatal sex differences in the chromatin landscape.

Previous studies have proposed that adult sex differences in behaviour arise from permanent epigenomic modifications induced during the neonatal hormone surge[35]. Our datasets allowed us to examine whether chromatin regions regulated by neonatal hormone maintain sex-biased accessibility into adulthood. Only a small proportion of NE-regulated regions (about 10%) are maintained as sex biased in gonadally intact adults (Extended Data Fig. 9a), implying substantial reprogramming of sex differences as a result of hormonal production during puberty (Fig. 2h). Notably, although most NE-open loci did not maintain male-biased accessibility after puberty, they still localized at adult male-biased genes and clustered around adult male-biased ATAC peaks (Extended Data Fig. 9b–d). These results suggest that certain male-biased genes undergo sequential regulation by ERα and AR in early life and adulthood, respectively.

## Sustained sex-biased gene expression

Our identification of approximately 2,000 chromatin regions controlled by the neonatal hormone surge suggests that ERα drives extensive sex differences in the expression of genes that control brain sexual differentiation. To identify these genes, and assess the longevity of their expression, we performed single-nucleus multiome (RNA and ATAC) sequencing on female and male BNST *Esr1*+ cells collected at P4 and P14, after the closure of the neonatal critical period[3] (Fig. 3b and Extended Data Fig. 10a, b). We profiled 14,836 cells and found that *Esr1*+ neuron identity is largely the same across P4, P14 and adulthood[36] (Fig. 3b and Extended Data Fig. 10c–f).

To identify TFs regulating *Esr1*+ neuron identity, we ranked TFs on their potential to control chromatin accessibility and their expression specificity across neuron types[37] (Extended Data Fig. 10g). This approach uncovered canonical GABAergic identity TF genes a priori, including *Lhx6*, *Prox1* and *Nkx2-1*, as well as regulators *Zfhx3* and *Nr4a2* (Extended Data Fig. 10g). In addition, *Nfix* was predicted to regulate the identity of the male-biased i1:Nfix neuron type (Fig. 3c and Extended Data Fig. 10f, g). Profiling Nfix binding in the adult BNSTp confirmed that the binding sites of this factor, including at the *Nfix* locus itself, are maintained in an active state primarily in i1:Nfix neurons (Fig. 3c, Extended Data Fig. 10h, i and Supplementary Table 8). Further examination of NE-responsive chromatin regions showed that NE-open regions vary as a function of neuron identity, with NE-open regions in i1:Nfix neurons preferentially

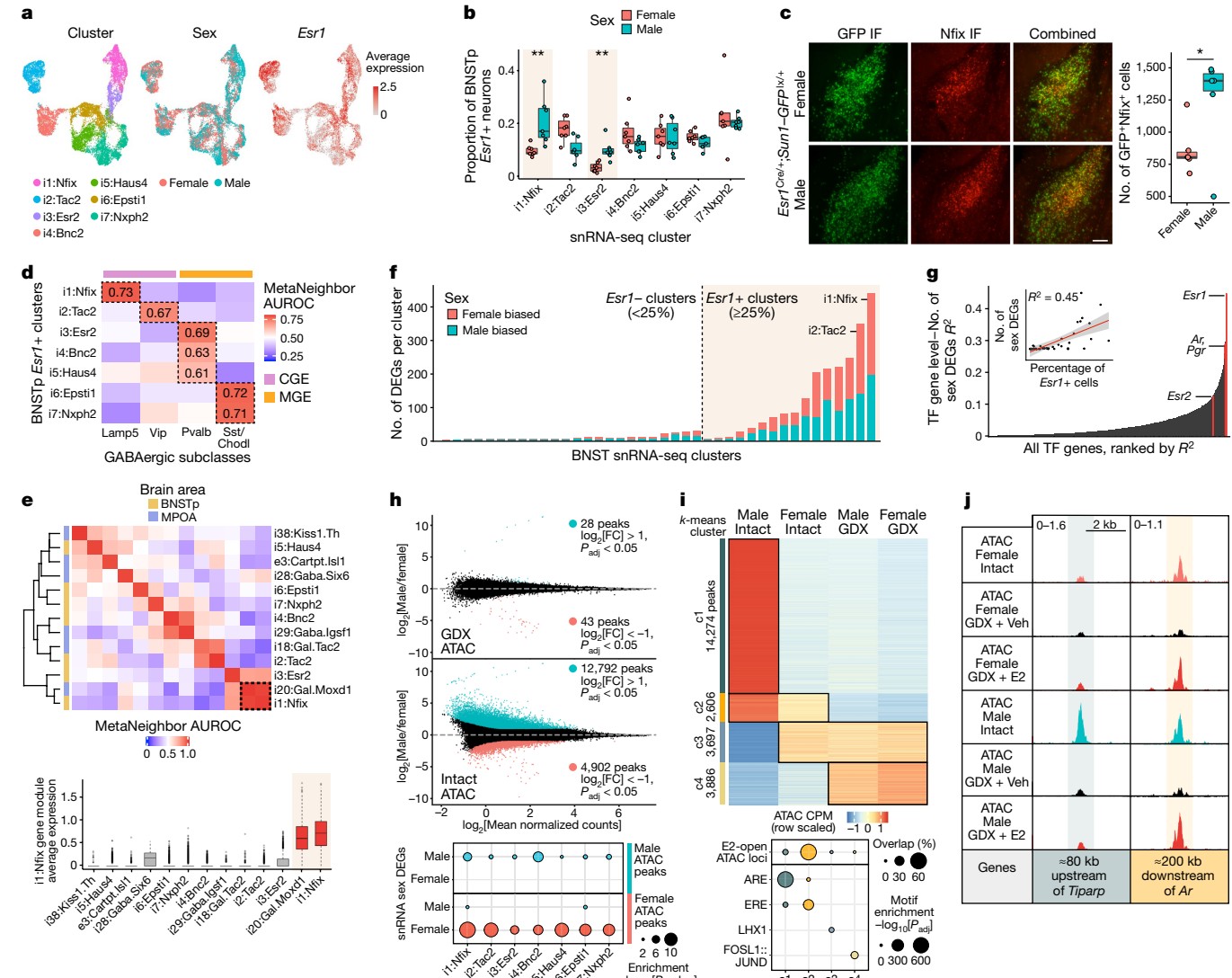

**Fig. 2 | Sex differences in cell type abundance and gene regulation in BNSTp Esr1+ cells. a**, Uniform manifold approximation and projection (UMAP) visualization of BNSTp *Esr1*+ snRNA-seq inhibitory neuron clusters, coloured by identity (left), sex (middle) and *Esr1* expression (right). **b**, Proportion of BNSTp *Esr1*+ nuclei in each BNSTp *Esr1*+ inhibitory neuron cluster per sex. Higher proportions of i1:Nfix ($P_{adj}$ = 0.002) and i3:Esr2 ($P_{adj}$ = 0.002) neurons are in males than females. Boxplot centre, median; box boundaries, first and third quartiles; whiskers, 1.5 × IQR from boundaries, $n$ = 7, **$P_{adj}$ < 0.01, one-sided, Wilcoxon rank-sum test, adjusted with the Benjamini–Hochberg procedure. **c**, BNSTp immunofluorescence (IF) staining for GFP (left micrographs) and Nfix (middle micrographs) in P14 female and male *Esr1*^Cre/+;*Sun1–GFP*^lx/+ animals (scale bar, 100 μm), with combined images (right micrographs) and their quantification (boxplots; right). Boxplot centre, median; box boundaries, first and third quartiles; whiskers, 1.5 × IQR from boundaries, $n$ = 6, $P$ = 0.0422, *$P$ < 0.05, two-sided, unpaired *t*-test. **d**, Heatmap of median MetaNeighbor area under the receiver operating characteristic curve (AUROC) values for BNSTp *Esr1*+ clusters and cortical/hippocampal GABAergic neuron subclasses. The colour bar indicates the developmental origin of GABAergic subclasses. CGE, caudal ganglionic eminence; MGE, medial ganglionic eminence. **e**, Top: heatmap of MetaNeighbor AUROC values

for BNSTp and MPOA *Esr1*+ clusters. Bottom: average expression of i1:Nfix marker genes across BNSTp and MPOA *Esr1*+ clusters. Dotted box indicates shared identity of i1:Nfix and i20:Gal.Moxd1 cells. $n$ = 297 i20:Gal.Moxd1 cells, 2,459 i1:Nfix cells. Boxplot centre, median; box boundaries, first and third quartiles; whiskers, 1.5 × IQR from boundaries. **f**, Number of differentially expressed genes (DEGs) between females and males (DESeq2, $P_{adj}$ < 0.1) per BNST neuron snRNA-seq cluster. **g**, $R^2$ between percentage of TF gene expression and number of sex DEGs per cluster across snRNA-seq clusters. The inset shows correlation for the top-ranked TF gene, *Esr1*. The error band represents the 95% confidence interval. **h**, Differential ATAC sites between gonadectomized (GDX), vehicle-treated females and males (top) and gonadally intact females and males (middle). Blue dots (edgeR, $\log_2[FC]$ > 1, $P_{adj}$ < 0.05), red dots (edgeR, $\log_2[FC]$ < –1, $P_{adj}$ < 0.05). Bottom: enrichment analysis of sex-biased ATAC peaks at sex DEGs. **i**, Top: *k*-means clustering (c1–c4) of differentially accessible ATAC peaks across four conditions(edgeR, $P_{adj}$ < 0.01). Bottom: dotplot showing the percentage of sites per cluster overlapping E2-open ATAC loci and motif enrichment analysis of peaks in each cluster (AME algorithm). ARE, androgen response element. **j**, Example ATAC peaks in *k*-means clusters 1 and 2. Top left number is the y-axis range in CPM. Shaded band indicates peak region.

containing Nfix binding events (Fig. 3d). These data suggest that, in addition to specifying the chromatin landscape, identity TFs may dictate the cellular response to neonatal oestradiol by influencing ERα binding.

Differential expression analysis across *Esr1*+ neuron types on P4 identified >400 sex-biased genes (Fig. 3e, Extended Data Fig. 11a and

Supplementary Table 9). Performing RNA-seq on BNSTp *Esr1*+ cells collected from females treated at birth with vehicle or oestradiol showed that these sex differences largely arise as a consequence of the neonatal surge (Extended Data Fig. 11b–e and Supplementary Table 10). Notably, oestradiol-dependent sex differences in gene expression

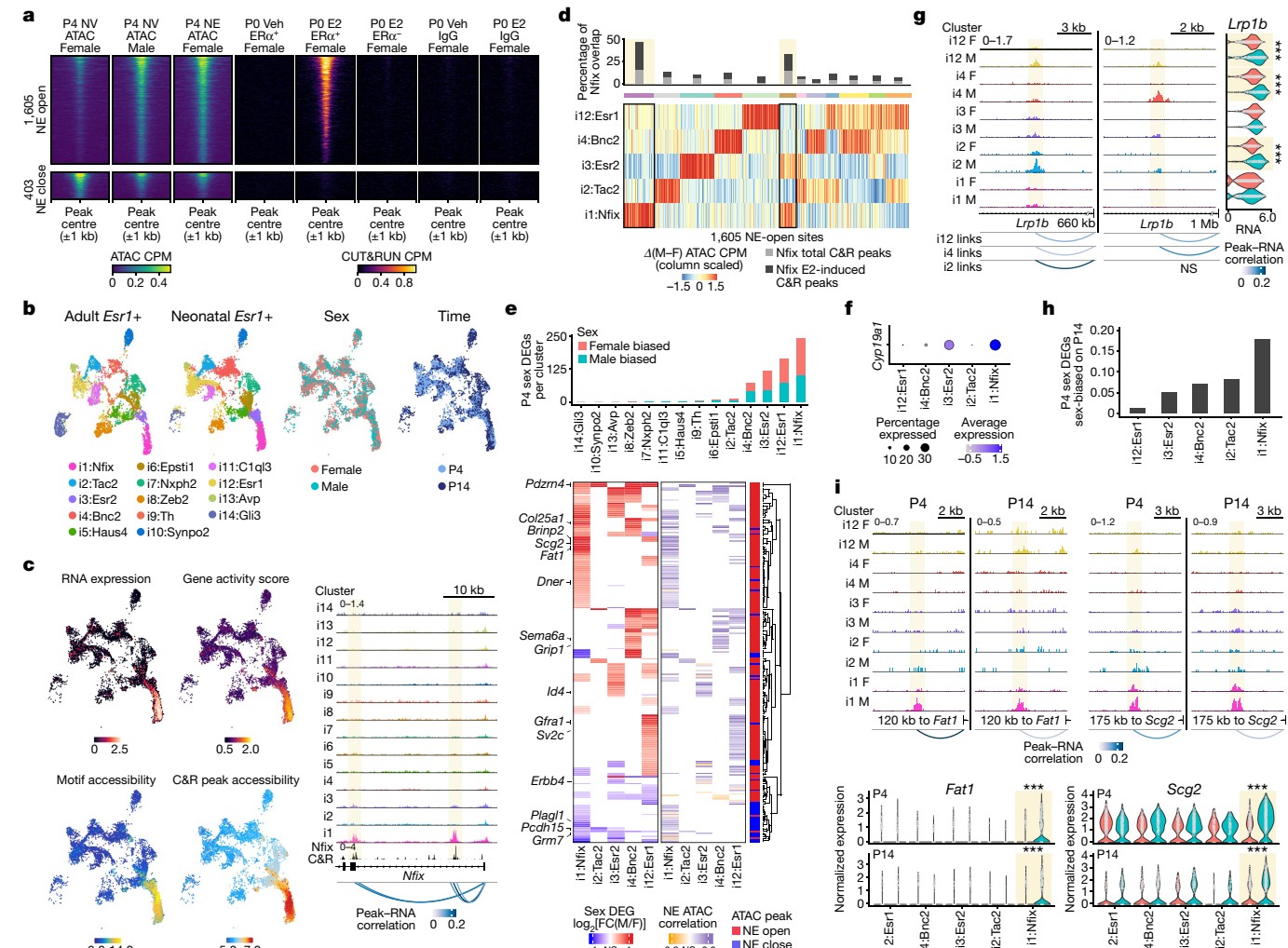

**Fig. 3 | Neonatal ERα genomic binding drives a sustained male-biased gene expression program. a**, Heatmap of P4 BNST *Esr1*+ ATAC, P0 IgG CUT&RUN and P0 ERα CUT&RUN CPM ±1 kb around 1,605 NE-open and 403 NE-close ATAC peaks (edgeR, $P_{adj}$ < 0.1). ERα⁺, Sun1–GFP⁺ nuclei; ERα⁻, Sun1–GFP⁻ nuclei. **b**, UMAPs of adult (left) and neonatal (middle left) BNST *Esr1*+ snRNA-seq clusters; neonatal snRNA-seq clusters coloured by sex (middle right) and time point (right). **c**, Left: UMAPs of *Nfix* expression (top left), gene activity score (top right), motif chromVAR deviation score (bottom left) and CUT&RUN chromVAR deviation score (bottom right). Right: neonatal single-nucleus ATAC (snATAC) and adult BNSTp Nfix CUT&RUN tracks at the *Nfix* locus. Top left number is the y-axis range in CPM. Shaded band indicates peak region. Peak–RNA correlation indicates correlation coefficient for snATAC peaks correlated with *Nfix* expression. **d**, Heatmap of differential snATAC CPM between males (M) and females (F) at 1,605 NE-open sites, scaled across snRNA-seq clusters and grouped using *k*-means clustering. The barplot indicates the percentage of overlap for each *k*-means cluster with total and E2-induced BNSTp Nfix CUT&RUN peaks. **e**, Top: number of sex DEGs (MAST, $P_{adj}$ < 0.05) in P4 multiome

clusters. Bottom: heatmaps indicating RNA log₂[FC] of P4 sex DEGs (left) and Pearson's correlation coefficient of NE-open (red) and NE-close (blue) ATAC peaks (right) linked to sex DEGs in each cluster. Genes without significant differential expression or correlation coefficients (not significant (NS)) are shown in white. **f**, *Cyp19a1*/aromatase expression on P4. **g**, Left: NE-open ATAC peaks correlating with *Lrp1b* expression in *Cyp19a1*⁻ clusters, i2:Tac2 and i12:Esr1. Top left number is the y-axis range in CPM. Shaded band indicates peak region. Right, sex difference in *Lrp1b* expression in i2:Tac2 (*n* = 260 female, 153 male, $P_{adj}$ = 2.13 × 10⁻⁸), i4:Bnc2 (*n* = 437 female, 373 male, $P_{adj}$ = 5.62 × 10⁻³⁷), i12:Esr1 (*n* = 803 female, 507 male, $P_{adj}$ = 1.09 × 10⁻¹²) cells. ***$P_{adj}$ < 0.001, MAST. **h**, Proportion of P4 sex DEGs detected as sex biased on P14. **i**, Top: i1:Nfix-specific, NE-open ATAC peaks at *Fat1* and *Scg2* loci on P4 and P14. Top left number is the y-axis range in CPM. Shaded band indicates peak region. Bottom: Sex difference in i1:Nfix *Fat1* and *Scg2* expression on P4 (*Fat1*, $P_{adj}$ = 1.28 × 10⁻³⁷; *Scg2*, $P_{adj}$ = 1.54 × 10⁻⁴⁶; *n* = 887 female, 676 male) and P14 (*Fat1*, $P_{adj}$ = 1.13 × 10⁻¹¹; *Scg2*, $P_{adj}$ = 1.52 × 10⁻⁵; *n* = 554 female, 829 male). ***$P_{adj}$ < 0.001, MAST.

and chromatin state occurred in neurons lacking *Cyp19a1*/aromatase expression (Fig. 3e–g), indicative of non-cell-autonomous oestradiol signalling.

To link our chromatin and gene expression data, we constructed a gene regulatory map across *Esr1*+ neuron types consisting of sex-biased genes and NE-regulated enhancers with correlated accessibility (Fig. 3e and Extended Data Fig. 11f, g). This map demonstrates both divergent responses across neuron types as well as neuron-type-specific enhancers for common sex-biased targets. Notably, we identified *Arid1b*, an autism spectrum disorder candidate gene, among genes regulated by distinct enhancers across neuron types (Extended Data Fig. 11g).

Further examination showed that about 40% of high-confidence (family-wise error rate ≤ 0.05) autism spectrum disorder candidate genes[38], including *Grin2b, Scn2a1* (also known as *Scn2a*) and *Slc6a1*, contained NE-open chromatin regions and ERα occupancy (Extended Data Fig. 8j and Supplementary Table 6).

We also examined whether sex-biased genes, and their corresponding enhancers, are sustained across the neonatal critical period by comparing *Esr1*+ neurons between P4 and P14. Although the total number of sex-biased genes declined between P4 and P14, a subset persisted as sex biased throughout the neonatal critical window (Fig. 3h and Supplementary Table 10). In i1:Nfix neurons, about 20% of differentially

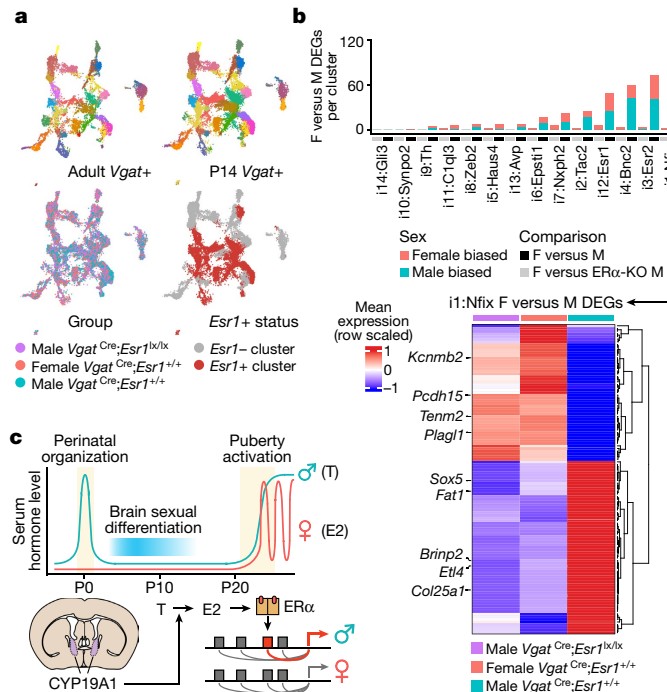

**Fig. 4 | ERα is required for sustained sex differences in gene expression.**
**a**, UMAPs of adult (top left) and P14 (top right) BNST *Vgat*+ snRNA-seq clusters; P14 *Vgat*+ snRNA clusters coloured by group (bottom left) and *Esr1*+ status (bottom right). **b**, Top: number of female versus male sex DEGs (MAST, $P_{adj} < 0.05$) in P14 snRNA clusters (black bar). Number of female versus male sex DEGs detected in female versus male KO comparison (grey bar). Bottom: heatmap of mean expression of i1:Nfix sex DEGs, scaled across control males, control females and conditional ERα-KO males. **c**, Neonatal ERα activation drives a sustained male-typical gene expression program.

expressed genes on P4 persisted as sex biased on P14. These genes regulate distinct components of neural circuit development, including neurite extension (*Klhl1* and *Pak7* (also known as *Pak5*)), axon pathfinding (*Epha3* and *Nell2*), neurotransmission (*Kcnab1* and *Scg2*) and synapse formation (*Il1rap* and *Tenm2*; Fig. 3h and Extended Data Fig. 11h). Together, these results show that neonatal ERα activation drives the epigenetic maintenance of a gene expression program that facilitates sexual differentiation of neuronal circuitry.

## Sustained sex differences require ERα

The adult display of male mating and territoriality behaviours requires ERα expression in GABAergic neurons[16]. To determine whether ERα is also required for sustained sex differences in gene expression, we performed snRNA-seq on 38,962 BNST GABAergic neurons isolated from P14 conditional mutant males lacking ERα (*Vgat*^Cre;*Esr1*^lx/lx;*Sun1–GFP*^lx), and littermate control females and males (*Vgat*^Cre;*Esr1*^+/+;*Sun1–GFP*^lx; Fig. 4a and Extended Data Fig. 12a–d). GABAergic neurons in ERα-mutant males did not deviate from P14 control or adult BNST neuron types (Fig. 4a and Extended Data Fig. 12b), indicating that ERα is dispensable for neuron identity. However, the abundance of male-biased i1:Nfix and i3:Esr2 neurons dropped to female levels in *Vgat*^Cre;*Esr1*^lx/lx males (Fig. 4b and Extended Data Fig. 12d), suggesting that neonatal ERα activation is essential for their male-typical abundance.

Differential expression analysis between control females and control or conditional ERα-knockout (KO) males in each neuron type established that ERα is required for nearly all sexually dimorphic gene expression, with the exception of those located on the Y chromosome or escaping X inactivation (Fig. 4b, Extended Data Fig. 12e and Supplementary Table 11). Notably, ERα-KO males exhibited feminized

expression of sex-biased genes (Fig. 4c and Extended Data Fig. 12f). Together, these findings demonstrate that the neonatal hormone surge drives a sustained male-typical gene expression program through activation of a master regulator TF, ERα (Fig. 4c).

## Discussion

Here we identify the genomic targets of ERα in the brain and demonstrate that BNSTp sexual differentiation is defined by both male-biased cell number and gene expression. We find that sexual dimorphism in the BNSTp equates to increased numbers of i1:Nfix and i3:Esr2 neurons in males. The transcriptomic identity of i1:Nfix neurons resembles that of cortical *Lamp5*+ neurogliaform interneurons, which provide regional inhibition through synaptic and ambient release of GABA[39]. As all BNSTp neurons and much of the POA are GABAergic, we predict that higher numbers of i1:Nfix inhibitory neurons enables stronger disinhibition of downstream projection sites. The net result is a gain of responses to social information, leading to male-typical levels of mounting or attacking[40]. Male-biased populations of inhibitory neurons also modulate sex-typical behaviours in *Drosophila*, but they do not rely on gonadal hormones to specify sex-biased enhancers[41–45]. In vertebrates, hormone receptor signalling may have evolved to coordinate gene regulation throughout a neural circuit as a strategy for controlling context-dependent behavioural states. Moreover, the association between hormone receptor target genes identified here and human neurological and neurodevelopmental conditions may explain the notable sex biases of these diseases.

Our data show that the neonatal hormone surge activates ERα to drive a sustained male-biased gene expression program in the developing brain. We speculate that this program establishes male-typical neuronal connectivity across the neonatal critical period and potentially primes the response to hormone receptor activation at puberty. In the adult brain, gonadectomy ablated sex differences in chromatin accessibility, and under these conditions, *Esr1*+ neurons of both sexes exhibited a similar genomic response to exogenous oestradiol. Together, these findings suggest that although sex differences in developmental gonadal hormone signalling establish dimorphisms in BNSTp circuitry, the genome remains responsive to later alterations in the hormonal milieu. Likewise, manipulating hormonal status, circuit function or individual genes consistently demonstrates that both sexes retain the potential to engage in behaviours typical of the opposite sex[46–49]. This study implicates puberty as a further critical period for sexual differentiation of gene regulation and provides an archetype for studying hormone receptor action across life stages, brain regions and species.

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

## Methods

### Animals

All animals were maintained on a 12-h light/12-h dark cycle and provided food and water ad libitum. All mouse experiments were performed under strict guidelines set forth by the CSHL Institutional Animal Care and Use Committee. All animals were randomly assigned to experimental groups. *Esr1*[Cre] (ref. [50]), *Rpl22*[HA] (ref. [51]), *ROSA26*[CAG-Sun1-sfGFP–Myc] (ref. [52]); abbreviated as *Sun1–GFP*), *Vgat*[Cre] (ref. [53]) and C57Bl6/J wild-type mice were obtained from Jackson Labs. *Esr1*[lx] mice were received from S. A. Khan[54]. Adult male and female mice were used between 8 and 12 weeks of age. For adult hormone treatment experiments, animals were euthanized for tissue collection 4 h after subcutaneous administration of 5 µg E2 (Sigma E8515) suspended in corn oil (Sigma C8267) or vehicle 3 weeks post-gonadectomy. For neonatal CUT&RUN, ATAC–seq and RNA-seq experiments, animals were treated with 5 µg E2 or vehicle on P0 and collected 4 h later (ERα CUT&RUN) or 4 days later (ATAC–seq and nuclear RNA-seq). For neonatal multiome, snRNA-seq and IF quantification, animals were collected on P4 (multiome) or P14 (multiome, snRNA-seq and IF staining).

### Cell lines

Cell lines include mHypoA clu-175 clone (Cedarlane Labs) and MCF-7 (ATCC). Cell lines were not tested for mycoplasma contamination. Cells were maintained in standard DMEM supplemented with 10% FBS and penicillin/streptomycin. Before CUT&RUN, MCF7 cells were grown in phenol-red-free DMEM medium containing 10% charcoal-stripped FBS and penicillin/streptomycin for 48 h and then treated with 20 nM 17-β-oestradiol or vehicle (0.002% ethanol) for 45 min.

### Adult RNA-seq and in situ hybridization

Experiments were performed as previously described[55]. Briefly, the BNSTp was microdissected following rapid decapitation of deeply anaesthetized adult *Esr1*[Cre/+];*Rpl22*[HA/+] mice. Tissue homogenization, immunoprecipitation and RNA extraction were performed, and libraries were prepared from four biological replicate samples (each consisting of 8–9 pooled animals) using NuGEN Ovation RNA-Seq kits (7102 and 0344). Multiplexed libraries were sequenced with 76-bp single-end reads on the Illumina NextSeq. Validation by in situ hybridization staining and quantification was performed by an investigator blinded to experimental condition, as previously described[16,55]. Riboprobe sequences are listed in Extended Data Table 1.

### Isolation of nuclei from adult mice for ATAC–seq

Adult *Esr1*[Cre/+];*Sun1–GFP*[lx/+] mice (four pooled per condition) were deeply anaesthetized with ketamine/dexmedetomidine. Sections of 500 µm spanning the BNSTp were collected in an adult mouse brain matrix (Kent Scientific) on ice. The BNSTp was microdissected and collected in 1 ml of cold supplemented homogenization buffer (250 mM sucrose, 25 mM KCl, 5 mM MgCl$_2$, 120 mM tricine-KOH, pH 7.8), containing 1 mM dithiothreitol, 0.15 mM spermine, 0.5 mM spermidine and 1× EDTA-free PIC (Sigma Aldrich 11873580001). The tissue was dounce homogenized 15 times in a 1-ml glass tissue grinder (Wheaton) with a loose pestle. Next, 0.3% IGEPAL CA-630 was added, and the suspension was homogenized five times with a tight pestle. The homogenate was filtered through a 40-µm strainer and then centrifuged at 500*g* for 15 min at 4 °C. The pellet was resuspended in 0.5 ml homogenization buffer containing 1 mM dithiothreitol, 0.15 mM spermine, 0.5 mM spermidine and 1× EDTA-free PIC. A total of 30,000 GFP$^+$ nuclei were collected into cold ATAC-RSB (10 mM Tris-HCl pH 7.5, 10 mM NaCl, 3 mM MgCl$_2$) using the Sony SH800S Cell Sorter (purity mode) with a 100-µm sorting chip. After sorting, 0.1% Tween-20 was added, and the nuclei were centrifuged at 500*g* for 5 min at 4 °C. The pellet of nuclei was directly resuspended in transposition reaction mix.

### ATAC–seq library preparation

Tn5 transposition was performed using the OMNI-ATAC protocol[56]. A 2.5 µl volume of Tn5 enzyme (Illumina 20034197) was used in the transposition reaction. Libraries were prepared with NEBNext High-Fidelity 2× PCR Master Mix (NEB M0541L), following the standard protocol. After the initial five cycles of amplification, another four cycles were added, on the basis of qPCR optimization. Following amplification, libraries were size selected (0.5×–1.8×) twice with AMPure XP beads (Beckman Coulter A63880) to remove residual primers and large genomic DNA. Individually barcoded libraries were multiplexed and sequenced with paired-end 76-bp reads on an Illumina NextSeq, using either the Mid or High Output Kit.

### Cell line CUT&RUN

To collect cells for CUT&RUN, cells were washed twice with Hank's buffered salt solution (HBSS) and incubated for 5 min with pre-warmed 0.5% trypsin–EDTA (10×) at 37 °C/5% CO$_2$. Trypsin was inactivated with DMEM supplemented with 10% FBS and penicillin/streptomycin (mHypoA cells) or phenol-red-free DMEM supplemented with 10% charcoal-stripped FBS and penicillin/streptomycin (MCF-7 cells). After trypsinizing, cells were centrifuged at 500*g* in a 15-ml conical tube and resuspended in fresh medium. CUT&RUN was performed as previously described[14], with minor modifications. Cells were washed twice in wash buffer (20 mM HEPES, pH 7.5, 150 mM NaCl, 0.5 mM spermidine, 1× PIC, 0.02% digitonin). Cell concentration was measured on a Countess II FL Automated Cell Counter (Thermo Fisher). A total of 25,000 cells were used per sample. Cells were bound to 20 µl concanavalin A beads (Bangs Laboratories, BP531), washed twice in wash buffer, and incubated overnight with primary antibody (ERα: Santa Cruz sc-8002 or EMD Millipore Sigma 06-935, Nfix: Abcam ab101341) diluted 1:100 in antibody buffer (wash buffer containing 2 mM EDTA). The following day, cells were washed twice in wash buffer, and 700 ng ml$^{-1}$ protein A-MNase (pA-MNase, prepared in-house) was added. After 1 h incubation at 4 °C, cells were washed twice in wash buffer and placed in a metal heat block on ice. pA-MNase digestion was initiated with 2 mM CaCl$_2$. After 90 min, digestion was stopped by mixing 1:1 with 2× stop buffer (340 mM NaCl, 20 mM EDTA, 4 mM EGTA, 50 µg ml$^{-1}$ RNase A, 50 µg ml$^{-1}$ glycogen, 0.02% digitonin). Digested fragments were released by incubating at 37 °C for 10 min, followed by centrifuging at 16,000*g* for 5 min at 4 °C. DNA was purified from the supernatant by phenol–chloroform extraction, as previously described[14].

### Adult brain CUT&RUN

Nuclei were isolated from microdissected POA, BNSTp and MeAp from gonadectomized C57Bl6/J mice, following anatomic designations[57] (Fig. 1a), as described previously[52]. Following tissue douncing, brain homogenate was mixed with a 50% OptiPrep solution and underlaid with 4.8 ml of 30% then 40% OptiPrep solutions, in 38.5-ml Ultra-clear tubes (Beckman-Coulter C14292). Ultracentrifugation was performed with a Beckman SW-28 swinging-bucket rotor at 9,200 r.p.m. for 18 min at 4 °C. Following ultracentrifugation, an ≈1.5-ml suspension of nuclei was collected from the 30/40% OptiPrep interface by direct tube puncture with a 3-ml syringe connected to an 18-gauge needle. Nucleus concentration was measured on a Countess II FL Automated Cell Counter. For ERα CUT&RUN (1:100, EMD Millipore Sigma 06-935), 400,000 nuclei were isolated from BNST, MPOA and MeA of five animals. For Nfix CUT&RUN (1:100, Abcam ab101341), 200,000 nuclei were isolated from BNSTp of five animals. A total of 400,000 cortical nuclei were used for the CUT&RUN IgG control (1:100, Antibodies-Online ABIN101961). Before bead binding, 0.4% IGEPAL CA-630 was added to the nucleus suspension to increase affinity for concanavalin A magnetic beads. All subsequent steps were performed as described above, with a modified wash buffer (20 mM HEPES, pH 7.5, 150 mM NaCl, 0.1% BSA, 0.5 mM spermidine, 1× PIC).

## CUT&RUN library preparation

Cell line CUT&RUN libraries were prepared using the SMARTer Thru-PLEX DNA-seq Kit (Takara Bio R400676), with the following PCR conditions: 72 °C for 3 min, 85 °C for 2 min, 98 °C for 2 min, (98 °C for 20 s, 67 °C for 20 s, 72 °C for 30 s) × 4 cycles, (98 °C for 20 s, 72 °C for 15 s) × 14 cycles (MCF7) or 10 cycles (mHypoA). Brain CUT&RUN libraries were prepared using the same kit with 10 PCR cycles. All samples were size selected with AMPure XP beads (0.5×–1.7×) to remove residual adapters and large genomic DNA. Individually barcoded libraries were multiplexed and sequenced with paired-end 76-bp reads on an Illumina NextSeq, using either the Mid or High Output Kit. For the mHypoA experiment, samples were sequenced with paired-end 25-bp reads on an Illumina MiSeq.

## Nfix immunofluorescence staining

Brains were dissected from perfused P14 $Esr1^{Cre/+}$;$Sun1–GFP^{lx/+}$ animals and cryosectioned at 40 µm before immunostaining with primary antibodies to GFP (1:1,000, Aves GFP-1020) and Nfix (1:1,000, Thermo Fisher PA5-30897), and secondary antibodies against chicken (1:300, Jackson Immuno 703-545-155) and rabbit (1:800, Jackson Immuno 711-165-152), as previously described[16]. A Zeiss Axioimager M2 System equipped with MBF Neurolucida Software was used to take 20× wide-field image stacks spanning the BNSTp (five sections, both sides). The number of Nfix+, GFP+ and Nfix+GFP+ cells was quantified using Fiji/ImageJ from the centre three optical slices by an investigator blinded to condition.

## Neonatal bulk ATAC–seq

Female and male $Esr1^{Cre/+}$;$Sun1–GFP^{lx/+}$ mice were injected subcutaneously with 5 µg E2 or vehicle on P0 and collected on P4 (4–5 animals pooled per condition and per replicate). The BNSTp was microdissected, as described above, and collected in 300 µl of cold, supplemented homogenization buffer. Nuclei were extracted as described for the adult brain. After filtering through a 40-µm strainer, the nuclei were diluted 3:1 with 600 µl of cold, supplemented homogenization buffer and immediately used for sorting. A total of 30,000 GFP+ nuclei were collected into cold ATAC-RSB buffer using the Sony SH800S Cell Sorter (purity mode) with a 100-µm sorting chip. After sorting, nuclei transposition and library preparation were performed, as described above.

## P0 ERα CUT&RUN

Female $Esr1^{Cre/+}$;$Sun1–GFP^{lx/+}$ mice were injected subcutaneously with 5 µg E2 or vehicle on P0 and collected 4 h later (5 animals pooled per condition and per replicate). The BNSTp, MPOA and MeA were microdissected, and nuclei were extracted, as described for the neonatal bulk ATAC–seq experiment. After filtering through a 40-µm strainer, the nuclei were diluted 3:1 with 600 µl of cold, supplemented homogenization buffer. A 2 mM concentration of EDTA was added, and the sample was immediately used for sorting. A total of 150,000 GFP+ nuclei were collected into cold CUT&RUN wash buffer using the Sony SH800S Cell Sorter (purity mode) with a 100-µm sorting chip. GFP− events were collected into cold CUT&RUN wash buffer, and 150,000 nuclei were subsequently counted on the Countess II FL Automated Cell Counter for ERα– and IgG negative-control CUT&RUN. All subsequent steps were performed as described for the adult brain CUT&RUN experiments. P0 CUT&RUN libraries were prepared with 10 PCR cycles.

## Neonatal single-nucleus multiome sequencing

The BNST was microdissected fresh from P4 and P14 female and male $Esr1^{Cre/+}$;$Sun1–GFP^{lx/+}$ mice, as described above (4–5 animals pooled per condition). Nuclei were extracted and prepared for sorting, as performed for the neonatal bulk ATAC–seq experiment, with the inclusion of 1 U µl$^{-1}$ Protector RNase inhibitor (Sigma) in the homogenization buffer. A total of 40,000–50,000 GFP+ nuclei were collected into 1 ml of cold ATAC-RSB buffer, supplemented with 0.1% Tween-20, 0.01%

digitonin, 2% sterile-filtered BSA (Sigma A9576) and 1 U µl$^{-1}$ Protector RNase inhibitor. The nuclei were centrifuged in a swinging-bucket rotor at 500g for 10 min at 4 °C. About 950 µl of supernatant was carefully removed, and 200 µl 10x Genomics dilute nuclei buffer was added to the side of the tube without disturbing the pellet. The nuclei were centrifuged again at 500g for 10 min at 4 °C. About 240 µl of supernatant was carefully removed, and the nuclei were resuspended in the remaining volume (about 7 µl). Samples were immediately used for the 10x Genomics Single Cell Multiome ATAC + Gene Expression kit (1000285), following the manufacturer's instructions. snRNA-seq and snATAC–seq libraries were sequenced on an Illumina NextSeq, using the High Output kit. Each sample was sequenced to a depth of about 40,000–80,000 mean reads per cell for the snATAC library and about 40,000–50,000 mean reads per cell for the snRNA library.

## P14 snRNA-seq

The BNSTp was microdissected from P14 female and male $Vgat^{Cre}$;$Esr1^{+/+}$;$Sun1–GFP^{lx}$ and male $Vgat^{Cre}$;$Esr1^{lx/lx}$;$Sun1–GFP^{lx}$ mice. Tissue samples from individual animals were immediately flash frozen in an ethanol dry-ice bath and stored at −80 °C until $n$ = 3 animals were collected per group. On the day of the experiment, tissue samples were removed from −80 °C and maintained on dry ice. With the tissue still frozen, cold, supplemented homogenization buffer was added to the tube, and the tissue was immediately transferred to a glass homogenizer and mechanically dounced and filtered, as described for our other neonatal experiments. A total of 80,000–90,000 GFP+ nuclei were collected into 100 µl of cold ATAC-RSB buffer, supplemented with 1% sterile-filtered BSA (Sigma A9576), and 1 U µl$^{-1}$ Protector RNase inhibitor, in a 0.5-ml DNA lo-bind tube (Eppendorf) pre-coated with 30% BSA. After collection, nuclei were pelleted with two rounds of gentle centrifugation (200g for 1 min) in a swinging-bucket centrifuge at 4 °C. After the second round, the supernatant was carefully removed, leaving about 40 µl in the tube. The nuclei were gently resuspended in this remaining volume and immediately used for the 10x Genomics Single Cell 3′ Gene Expression kit v3 (1000424), following the manufacturer's instructions. Each biological sample was split into two 10× lanes, producing 6 libraries that were pooled and sequenced on an Illumina NextSeq 2000 to a depth of about 45,000–60,000 mean reads per cell.

## Neonatal nuclear RNA-seq

Female $Esr1^{Cre/+}$;$Sun1–GFP^{lx/+}$ mice were injected subcutaneously with 5 µg E2 or vehicle on P0. Four days later, animals were rapidly decapitated, and 400-µm sections were collected in cold homogenization buffer using a microtome (Thermo Scientific Microm HM 650V). The BNST was microdissected (4 animals pooled per condition) and collected in 1 ml of cold, supplemented homogenization buffer containing 0.4 U ml$^{-1}$ RNAseOUT (Thermo Fisher, 10777019). Nuclei were isolated as described for neonatal bulk ATAC–seq. A total of 12,000 GFP+ nuclei were collected into cold Buffer RLT Plus supplemented 1:100 with β-mercaptoethanol (Qiagen, 74034) using the Sony SH800S Cell Sorter (purity mode) with a 100-µm sorting chip. Nuclei lysates were stored at −80 °C until all replicates were collected. Nuclei samples for all replicates were thawed on ice, and RNA was isolated using the Qiagen RNeasy Plus Micro Kit (74034). Strand-specific RNA-seq libraries were prepared using the Ovation SoLo RNA-seq system (Tecan Genomics, 0501-32), following the manufacturer's guidelines. Individually barcoded libraries were multiplexed and sequenced with single-end 76-bp reads on an Illumina NextSeq, using the Mid Output Kit.

## Bioinformatics and data analysis

**CUT&RUN data processing.** Paired-end reads were trimmed to remove Illumina adapters and low-quality basecalls (cutadapt -q 30)[58]. Trimmed reads were aligned to mm10 using Bowtie2 (ref. [59]) with the following flags: --dovetail --very-sensitive-local --no-unal --no-mixed --no-discordant --phred33. Duplicate reads were removed using Picard

(http://broadinstitute.github.io/picard/) MarkDuplicates (REMOVE_DUPLICATES = true). Reads were filtered by mapping quality[60] (samtools view -q 40) and fragment length[61] (deepTools alignmentSieve --maxFragmentLength 120). Reads aligning to the mitochondrial chromosome and incomplete assemblies were also removed using SAMtools. After filtering, peaks were called on individual replicate BAM files using MACS2 callpeak (--min-length 25 -q 0.01)[62]. To identify consensus Nfix peaks across samples, MACS2 callpeak was performed on BAM files merged across biological replicates ($n = 2$) and subsequently intersected across treatment and sex. TF peaks that overlapped peaks called in the IgG control were removed using bedtools intersect (-v)[63] before downstream analysis.

**CUT&RUN data analysis.** CUT&RUN differential peak calling was performed with DiffBind v2.10.0(ref. [64]). A count matrix was created from individual replicate BAM and MACS2 narrowpeak files ($n = 2$ per condition). Consensus peaks were recentred to ±100 bp around the point of highest read density (summits = 100). Contrasts between sex and treatment were established (categories = c(DBA_TREATMENT, DBA_CONDITION)), and edgeR[65] was used for differential peak calling. Differential ERα peaks with $P_{adj} < 0.1$ were used for downstream analysis. For Nfix, differential peaks with a $P_{adj} < 0.1$ and abs(log$_2$[FC]) > 1 were used for downstream analysis. Differential peak calling for the MCF-7 CUT&RUN experiment was performed with DESeq2 ($P_{adj} < 0.1$) in DiffBind. Differential peak calling for the P0 ERα CUT&RUN experiment was performed with DESeq2 ($P_{adj} < 0.01$) in DiffBind. To identify sex-dependent, oestradiol-responsive peaks for adult brain ERα CUT&RUN, the DiffBind consensus peakset count matrix was used as input to edgeR, and an interaction between sex and treatment was tested with glmQLFTest.

Brain E2-induced ERα CUT&RUN peaks were annotated to NCBI RefSeq mm10 genes using ChIPseeker[66]. DeepTools plotHeatmap was used to plot ERα CUT&RUN (Fig. 1b), representing CPM-normalized bigwig files pooled across replicate and sex per condition, at E2-induced ERα peaks. Heatmaps of individual ERα CUT&RUN replicates are shown in Extended Data Fig. 2. CUT&RUNTools[67] was used to plot ERα CUT&RUN fragment ends surrounding ESR1 motifs (JASPAR MA0112.3) in E2-induced ERα ChIP–seq peaks. BETA (basic mode, -d 500000)[68] was used to determine whether ERα peaks were significantly overrepresented at E2-regulated RNA-seq genes ($P < 0.01$), as well as sex-dependent E2-regulated genes ($P < 0.01$), compared to non-differential, expressed genes. Motif enrichment analysis of ERα peaks was performed with AME[69] using the 2020 JASPAR core non-redundant vertebrate database. Motif enrichment analysis was performed using a control file consisting of shuffled primary sequences that preserves the frequency of $k$-mers (--control --shuffle--). The following seven ERα ChIP–seq files were lifted over to mm10 using UCSC liftOver and intersected with E2-induced ERα peaks to identify brain-specific and shared (≥4 intersections) ERα-binding sites: uterus (intersection of GEO: GSE36455 (uterus 1)[70] and GEO: GSE49993 (uterus 2)[71]), liver (intersection of GEO: GSE49993 (liver 1)[71] and GEO: GSE52351 (liver 2)[72]), aorta[72] (GEO: GSE52351), efferent ductules[73] (Supplementary Information) and mammary gland[74] (GEO: GSE130032). ClusterProfiler[75] was used to identify associations between brain-specific and shared ERα peak-annotated genes and Gene Ontology (GO) biological process terms (enrichGO, ont = 'BP', $P_{adj} < 0.1$). For Disease Ontology (DO) and HUGO Gene Nomenclature Committee (HGNC) gene family enrichment, brain-specific ERα peak-associated gene symbols were converted from mouse to human using bioMart[76] and then analysed with DOSE[77]; enrichDO, $P_{adj} < 0.1$) and enricher ($P_{adj} < 0.1$). Log-odds ESR1 and ESR2 motif scores in brain-specific and shared ERα peaks were calculated with FIMO[78], using default parameters.

MCF7 ERα CUT&RUN data were compared to MCF7 ERα ChIP–seq data from ref. [79] (GEO: GSE59530). Single-end ChIP–seq fastq files for two vehicle-treated and two 17β-oestradiol (E2)-treated IP and input samples were accessed from the Sequence Read Archive and processed identically to ERα CUT&RUN data, with the exception of fragment size filtering. Differential ERα ChIP–seq peak calling was performed using DiffBind DESeq2 ($P_{adj} < 0.01$). DeepTools was used to plot CPM-normalized ERα CUT&RUN signal at E2-induced ERα ChIP–seq binding sites. DREME[80] and AME were used to compare de novo and enriched motifs between E2-induced MCF7 ERα CUT&RUN and ChIP–seq peaks.

**Adult RNA-seq data processing and analysis.** Reads were adapter trimmed and quality filtered ($q > 30$) (http://hannonlab.cshl.edu/fastx_toolkit/), and then mapped to the mm10 reference genome using STAR[81]. The number of reads mapping to the exons of each gene was counted with featureCounts[82], using the NCBI RefSeq mm10 gene annotation. Differential gene expression analysis was performed using DESeq2 (ref. [83]) with the following designs: effect of treatment (design = ~ batch + hormone), effect of sex (design = ~ batch + sex), two-way comparison of treatment and sex (design = ~ batch + hormone_sex), four-way comparison (design = ~ 0 + hormone_sex) and sex–treatment interaction (design = ~ batch + sex + hormone + sex:hormone).

**ATAC–seq data processing.** ATAC–seq data were processed using the ENCODE ATAC–seq pipeline (https://github.com/ENCODE-DCC/atac-seq-pipeline) with default parameters. To generate CPM-normalized bigwig tracks, quality-filtered, Tn5-shifted BAM files were converted to CPM-normalized bigwig files using DeepTools bamCoverage (--binSize 1 --normalizeUsing CPM).

**Adult GDX treatment ATAC–seq data analysis.** ATAC–seq differential peak calling was performed with DiffBind v2.10.0. A DiffBind dba object was created from individual replicate BAM and MACS2 narrowPeak files ($n = 3$ per condition). A count matrix was created with dba.count, and consensus peaks were recentred to ±250 bp around the point of highest read density (summits = 250). Contrasts between sex and treatment were established (categories = c(DBA_TREATMENT, DBA_CONDITION)), and edgeR was used for differential peak calling. Differential peaks with an FDR < 0.05 and abs(log$_2$[FC]) > 1 or abs(log$_2$[FC] )> 0 were used for downstream analysis. DeepTools computeMatrix and plotHeatmap were used to plot mean ATAC CPM at E2-open ATAC peaks. To identify sex-dependent, oestradiol-responsive peaks, the DiffBind consensus peakset count matrix was used as input to edgeR, and an interaction between sex and treatment was tested with glmQLFTest. E2-open ATAC peaks and total vehicle or E2 ATAC peaks (intersected across replicate and sex for each treatment condition) were annotated to NCBI RefSeq mm10 genes using ChIPseeker. ClusterProfiler was used to calculate the enrichment of GO biological process terms. DO and HGNC gene family enrichment was performed on E2-open ATAC peak-associated genes, as described above for ERα CUT&RUN analysis. BETA (basic mode, -d 500000)[68] was used to determine whether E2-open ATAC peaks were significantly overrepresented at E2-regulated RNA-seq genes ($P < 0.01$), as well as sex-dependent E2-regulated genes ($P < 0.01$), compared to non-differential, expressed genes. Motif enrichment analysis of E2-open ATAC peaks was performed with AME, using the 2020 JASPAR core non-redundant vertebrate database. FIMO was used to determine the percentage of E2-open ATAC peaks containing the enriched motifs shown in Extended Data Fig. 4h, i.

**Adult gonadally intact ATAC–seq analysis.** ATAC–seq differential peak calling and comparison between gonadally intact (abbreviated as intact) and GDX ATAC samples were performed with DiffBind v2.10.0 and edgeR. A DiffBind dba object was created from individual replicate BAM and MACS2 narrowPeak files for the four groups: female intact ($n = 2$), male intact ($n = 2$), female GDX vehicle treated ($n = 3$), male GDX vehicle treated ($n = 3$). A count matrix was created with dba.count, and consensus peaks were recentred to ±250 bp around the point of

highest read density (summits = 250). The consensus peakset count matrix was subsequently used as input to edgeR. Differential peaks ($abs(\log_2[FC]) > 1$, $P_{adj} < 0.05$) were calculated between female intact and male intact and between female GDX vehicle treated and male GDX vehicle-treated groups using glmQLFTest. BETA was used to assess statistical association between gonadally intact, sex-biased ATAC peaks and sex DEGs called in BNSTp *Esr1*+ snRNA-seq clusters (top 500 genes per cluster, ranked by $P_{adj}$). Sex DEGs ranked by ATAC regulatory potential score[68], a metric that reflects the number of sex-biased peaks and distance of sex-biased peaks to the TSS, are shown in Extended Data Fig. 7g. HGNC gene family enrichment was performed on sex DEGs, using a background of expressed genes in any of the seven BNSTp *Esr1*+ clusters.

To identify differential peaks across the four conditions, an ANOVA-like design was created in edgeR by specifying multiple coefficients in glmQLFTest (coefficient = 2:4). A matrix of normalized counts in these differential peaks ($P_{adj} < 0.01$) was clustered using *k*-means clustering (kmeans function in R), with *k* = 4 and iter. max = 50. For each *k*-means cluster, the cluster centroid was computed, and outlier peaks in each cluster were excluded on the basis of having low Pearson's correlation with the cluster centroid ($R < 0.8$). Depth-normalized ATAC CPM values in these peak clusters are shown in Fig. 2i (mean across biological replicates per group) and Extended Data Fig. 7 (individual biological replicates). Peak cluster overlap with E2-open ATAC loci ($abs(\log_2[FC]) > 0$, $P_{adj} < 0.05$) was computed with bedtools intersect (-wa). For each peak cluster, motif enrichment analysis was performed by first generating a background peak list (matching in GC content and accessibility) from the consensus ATAC peak matrix using chromVAR (addGCBias, getBackgroundPeaks)[84], and then calculating enrichment with AME using the background peak list as the control (--control background peaks). In Fig. 2i, the JASPAR 2020 AR motif (MA0007.3) is labelled as ARE, and the ESR2 motif (MA0258.2) is labelled as ERE.

**Adult snRNA-seq and single-cell RNA-seq analysis.** Mouse BNST snRNA-seq data containing 76,693 neurons across 7 adult female and 8 adult male biological replicates[26] were accessed from GEO: GSE126836 and loaded into a Seurat object[85]. Mouse MPOA single-cell RNA-seq data containing 31,299 cells across 3 adult female and 3 adult male biological replicates[32] were accessed from GEO: GSE113576 and loaded into a Seurat object. Cluster identity, replicate and sex were added as metadata features to each Seurat object. Pseudo-bulk RNA-seq analysis was performed to identify sex differences in gene expression in the BNST snRNA-seq dataset. Briefly, the Seurat object was converted to a SingleCellExperiment object (as.SingleCellExperiment). Genes were filtered by expression (genes with >1 count in ≥5 nuclei). NCBI-predicted genes were removed. For each cluster, nuclei annotated to the cluster were subsetted from the main Seurat object. Biological replicates containing ≤20 nuclei in the subsetted cluster were excluded. Gene counts were summed for each biological replicate in each cluster. Differential gene expression analysis across sex in each cluster was performed on the filtered, aggregated count matrix using DESeq2 (design = ~ sex) with alpha = 0.1. The BNSTp_Cplx3 cluster was excluded, as none of the replicates in this cluster contained more than 20 nuclei. Clusters containing ≥25% nuclei with ≥1 *Esr1* counts in the main Seurat object were classified as *Esr1*+ (i1:Nfix, i2:Tac2, i3:Esr2, i4:Bnc2, i5:Haus4, i6:Epsti1, i7:Nxph2, i8:Zeb2, i9:Th, i10:Synpo2, i11:C1ql3, i12:Esr1, i13:Avp, i14:Gli3). To identify TFs that correlate with sex DEG number per cluster (Fig. 2g), a linear regression model with percentage of TF expression as the predictor variable and sex DEG number per cluster as the response variable was generated using the lm function in R stats (formula = percentage of TF expression ~ DEG number). This model was tested for all TFs in the SCENIC[86] mm10 database. All TFs were then ranked by $R^2$ to identify those most predictive of sex DEG number, and the ranked $R^2$ values are shown in Fig. 2g.

To visualize BNSTp *Esr1*+ snRNA-seq data (Fig. 2a), BNSTp *Esr1*+ clusters were subsetted from the main Seurat object. Gene counts were normalized and log transformed (LogNormalize), and the top 2,000 variable features were identified using FindVariableFeatures (selection.method = vst). Gene counts were scaled, and linear dimensionality reduction was performed by principal component analysis (runPCA, npcs = 10). BNSTp *Esr1*+ clusters were visualized with UMAP (runUMAP, dims = 10). To generate the heatmaps in Extended Data Fig. 7a, pseudo-bulk counts for each biological replicate included in the analysis were normalized and transformed with variance-stabilizing transformation (DESeq2 vst), subsetted for sex-biased genes in each cluster, and *z*-scaled across pseudo-bulk replicates.

To examine differential abundance of BNSTp *Esr1*+ clusters between sexes (Fig. 2b), the proportion of total nuclei in each BNSTp *Esr1*+ cluster was calculated for each biological replicate. After calculating the proportions of nuclei, sample MALE6 was excluded as an outlier for having no detection (0 nuclei) of i1:Nfix and i2:Tac2 clusters and overrepresentation of the i5:Haus4 cluster. The one-sided Wilcoxon rank-sum test (wilcox.test in R stats) was used to test for male-biased abundance of nuclei across biological replicates in each cluster. *P* values were adjusted for multiple hypothesis testing using the Benjamini–Hochberg procedure (method = fdr).

To identify marker genes enriched in the i1:Nfix cluster relative to the remaining six BNSTp *Esr1*+ clusters (Extended Data Fig. 6b), differential gene expression analysis was performed using DESeq2 with design = ~ cluster_id (betaPrior = TRUE), alpha = 0.01, lfcThreshold = 2, altHypothesis = greater.

To identify the enrichment of *Lamp5*+ subclass markers in BNSTp and MPOA *Esr1*+ clusters (Extended Data Fig. 6e), a Seurat object was created from the Allen Brain Atlas Cell Types dataset. Gene counts per cell were normalized and log transformed (LogNormalize), and subclass-level marker genes were calculated with the Wilcoxon rank-sum test (FindAllMarkers, test.use = wilcox, min.diff.pct = 0.2). The mean expression of *Lamp5*+ subclass markers ($avg\_log[FC] > 0.75$, $P_{adj} < 0.05$, <40% in non-*Lamp5*+ subclasses) was calculated in BNSTp and MPOA *Esr1*+ clusters and visualized using pheatmap.

To generate the UMAP plots shown in Extended Data Fig. 6g, BNSTp *Esr1*+ clusters were integrated with MPOA/BNST *Esr1*-expressing clusters (e3: Cartpt_Isl1, i18: Gal_Tac2, i20: Gal_Moxd1, i28: Gaba_Six6, i29: Gaba_Igsf1, i38: Kiss1_Th) using Seurat. Anchors were identified between cells from the two datasets, using FindIntegrationAnchors. An integrated expression matrix was generated using IntegrateData (dims = 1:10). The resulting integrated matrix was used for downstream PCA and UMAP visualization (dims = 1:10).

**MetaNeighbor analysis.** MetaNeighbor[28] was used to quantify the degree of similarity between BNSTp *Esr1*+ clusters and MPOA *Esr1*+ clusters and between BNSTp *Esr1*+ clusters and cortical/hippocampal GABAergic neuron subclasses from the Allen Brain Atlas Cell Types database[29]. Briefly, the BNST and MPOA Seurat objects were subsetted for *Esr1*+ clusters, and then transformed and merged into one Single-CellExperiment object. For the BNSTp and cortex comparison, BNSTp *Esr1*+ clusters were merged into a SingleCellExperiment with cortical/hippocampal GABAergic cortical clusters. Unsupervised MetaNeighbor analysis was performed between BNST and MPOA clusters, and between BNST and cortical/hippocampal clusters, using highly variable genes identified across datasets (called with the variableGenes function). The median AUROC value per cortical/hippocampal GABAergic subclass across Allen Brain Atlas datasets for each BNSTp *Esr1*+ cluster is shown in Fig. 2d.

**Neonatal bulk ATAC–seq analysis.** Differential peak calling on the neonatal bulk ATAC–seq experiment was performed with DiffBind v2.10.0 and edgeR. A count matrix was created from individual replicate BAM and MACS2 narrowpeak files (*n* = 3 per condition). Consensus

peaks were recentred to ±250 bp around the point of highest read density (summits = 250), and the consensus peakset count matrix was subsequently used as input to edgeR. Differential peaks across the three treatment groups (NV female, NV male, NE female) were calculated by specifying multiple coefficients in glmQLFTest (coefficient = 4:5). To identify accessibility patterns across differential peaks ($P_{adj} < 0.05$), a matrix of normalized counts in differential peaks was hierarchically clustered using pheatmap, and the resulting dendrogram tree was cut with $k$ = 6 to achieve 6 peak clusters (Extended Data Fig. 8a). The two largest clusters were identified as having higher accessibility in NV males and NE females compared to NV females (cluster 3, labelled as NE open), or lower accessibility in NV male and NE female compared to NV females (cluster 5, labelled as NE close). Motif enrichment analysis of NE-open peaks was performed with AME using the 2020 JASPAR core non-redundant vertebrate database. GO biological process, DO and HGNC gene family enrichment analyses were performed, as described above for adult GDX treatment ATAC–seq data analysis.

**Neonatal single-nucleus multiome data processing and analysis.**
Raw sequencing data were processed using the Cell Ranger ARC pipeline (v2.0.0) with the cellranger-arc mm10 reference. Default parameters were used to align reads, count unique fragments or transcripts, and filter high-quality nuclei. Individual HDF5 files for each sample containing RNA counts and ATAC fragments per cell barcode were loaded into Seurat (Read10X_h5). Nuclei with lower-end ATAC and RNA QC metrics (<1,000 ATAC fragments, <500 counts, nucleosomal signal > 3, TSS enrichment < 2) were removed. DoubletFinder[87] was then used to remove predicted doublets from each sample (nExp = 9% of nuclei per sample). Following doublet removal, nuclei surpassing upper-end ATAC and RNA QC metrics (>60,000 ATAC fragments, >20,000 RNA counts, >6,000 genes detected) were removed. After filtering, Seurat objects for each sample were subsetted for the RNA assay and merged. Gene counts were normalized and log transformed (LogNormalize), and the top 2,000 variable features were identified using FindVariableFeatures (selection.method = 'vst'). Gene counts were scaled, regressing out the following variables: number of RNA counts, number of RNA genes, percentage of mitochondrial counts and biological sex. Linear dimensionality reduction was performed by principal component analysis (runPCA, npcs = 25). A $k$-nearest-neighbours graph was constructed on the basis of Euclidean distance in PCA space and refined (FindNeighbors, npcs = 25), and then the nuclei were clustered using the Louvain algorithm (FindClusters, resolution = 0.8). snRNA clusters were visualized with UMAP (runUMAP, dims = 25). To reduce the granularity of clustering, a phylogenetic tree of cluster identities was generated from a distance matrix constructed in PCA space (BuildClusterTree) and visualized as a dendrogram (PlotClusterTree). DEGs between clusters in terminal nodes of the phylogenetic tree were calculated (FindMarkers, test.use = 'wilcox', $P_{adj} < 0.05$), and clusters were merged if they had fewer than 10 DEGs with the following parameters: >0.5 avg_log[FC], <10% expression in negative nuclei, and >25% expression in positive nuclei. The final de novo snRNA-seq clusters are shown in Extended Data Fig. 10c.

Inhibitory neuron clusters (*Slc32a1/Gad2*+) from the neonatal multiome dataset were subsequently assigned to adult BNST *Esr1*+ cluster labels using Seurat. Adult BNST *Esr1*+ clusters (as defined above) were subsetted from the adult snRNA-seq object and randomly downsampled to 5,000 nuclei. Normalization, data scaling and linear dimensionality reduction were performed with the same parameters as for neonatal and adult *Esr1*+ inhibitory neuron clusters. Anchor cells between adult (reference) and neonatal (query) datasets were first identified using FindTransferAnchors. Reference cluster labels, as well as the corresponding UMAP structure, were subsequently transferred to the neonatal dataset using MapQuery. Prediction scores, which measure anchor consistency across the neighbourhood structure of reference and query datasets as previously described[85], were used to quantify the confidence of label transfer from adult to neonatal nuclei. Extended Data Fig. 10d shows the prediction scores per reference cluster and time point of nuclei mapped onto adult reference cluster labels as well as the percentage of nuclei from each de novo cluster mapped onto each adult reference cluster (prediction score > 0.5). To further validate the quality of label transfer between adult and neonatal datasets, we computed DEGs between neonatal clusters post label transfer (FindMarkers, test.use = 'wilcox', $P_{adj} < 0.05$, min.diff.pct = 0.1, avg_log[FC] > 0.5) and calculated their background-subtracted, average expression (AddModuleScore) in neonatal and adult BNST *Esr1*+ nuclei (visualized in Extended Data Fig. 10e).

To generate pseudo-bulk, normalized ATAC bigwig tracks for each snATAC cluster, we first re-processed the cellranger ARC output BAM file for each sample using SAMtools (-q 30 -f 2) and removed duplicate reads per cell barcode using picard MarkDuplicates (BARCODE_TAG=CB REMOVE_DUPLICATES = true). Sinto (https://timoast.github.io/sinto/) was used to split ATAC alignments for each cluster into individual BAM files using cell barcodes extracted from the Seurat object. CPM-normalized bigwig files were computed for each pseudo-bulk BAM file using DeepTools bamCoverage (--binSize 1 --normalizeUsing CPM).

To analyse the neonatal multiome snATAC data, we used ArchR[88]. Separate Arrow files were created for each multiome sample, and then merged into a single ArchR project. Gene activity scores per nucleus were calculated at the time of Arrow file creation (addGeneScoreMat = TRUE). Metadata (cluster label, sex, time and QC metrics) were transferred from the previously generated Seurat object to the ArchR project by cell barcode-matching. Dimensionality reduction was performed on the snATAC data using ArchR's iterative Latent Semantic Indexing approach (addIterativeLSI). Per-nucleus imputation weights were added using MAGIC[89] in ArchR (addImputeWeights) to denoise sparse ATAC data for UMAP visualization. Cluster-aware ATAC peak calling was performed using ArchR's iterative overlap peak merging approach (addReproduciblePeaks, groupBy = 'cluster'). Following peak calling, CISBP human motif annotations were added for each peak (addPeakAnnotation), and chromVAR deviation scores (addDeviationsMatrix) were calculated for each motif. In addition, chromVAR was used to calculate per-nucleus deviation scores for consensus BNSTp Nfix CUT&RUN peaks. To perform neuron identity regulator analysis (Extended Data Fig. 10g), the correlation between TF RNA expression and motif deviation score was calculated for all TFs in the CISBP motif database (correlateMatrices). TFs with a correlation coefficient >0.5 and a maximum TF RNA log₂[FC] value between each cluster in the top 50% were classified as neuron identity regulators (coloured pink in Extended Data Fig. 10g).

For visualization of gene activity and CISBP motif deviation scores (Fig. 3c and Extended Data Fig. 10g), scores were imputed (imputeMatrix), transferred to the original Seurat object by cell barcode matching, and visualized using FeaturePlot. Signac[90] was used to generate and store peak-by-cell count matrices for each sample. snATAC markers for each cluster were calculated (FindAllMarkers, test.use = 'LR', vars.to.regress = 'nCount_ATAC', min.pct = 0.1, min.diff.pct = 0.05, logfc.threshold = 0.15). Pseudo-bulk snATAC cluster CPM was computed for each marker peak using DeepTools multiBigwigSummary and visualized with pheatmap (Extended Data Fig. 10f). Motif enrichment analysis of snATAC marker peaks for each cluster was performed using FindMotifs. The top three enriched motifs per snATAC cluster are shown in Extended Data Fig. 10f.

To identify sex-biased enrichment of NE-open loci across P4 snATAC clusters (Fig. 3d), we first filtered out low-abundance P4 snATAC clusters (<400 nuclei), and then computed the difference in ATAC CPM between males and females at NE-open loci in each cluster. Differential ATAC CPM values were scaled across clusters, then grouped using $k$-means clustering ($k$ = 12, iter.max = 50) and visualized with pheatmap (Fig. 3d). To call sex DEGs ($P_{adj} < 0.05$) in each cluster and time point, we used

MAST[91] in Seurat (FindMarkers, test.use = 'MAST', min.pct = 0.05, logfc. threshold = 0.2, latent.vars = 'nFeature_RNA', 'nCount_RNA').

To link NE-regulated loci to sex DEGs at P4 and P14 (Fig. 3e and Extended Data Fig. 11h), we computed the Pearson correlation coefficient between sex DEG expression and NE-regulated peak accessibility for each cluster (LinkPeaks, min.distance = 2,000, distance = 1,000,000, min.cells = 2% of cluster size). Sex DEG $\log_2$[FC] values and NE-regulated ATAC site correlation coefficients were hierarchically clustered and visualized using ComplexHeatmap[92].

**P14 snRNA-seq data processing and analysis.** Raw sequencing data were processed using the Cell Ranger pipeline (v6.0.0) with the refdata-gex-mm10-2020-A reference. Default parameters were used to align reads, count unique transcripts and filter high-quality nuclei. Individual HDF5 files for each sample were loaded into Seurat. Nuclei with lower-end RNA QC metrics (<1,000 counts) were removed. DoubletFinder[87] was then used to remove predicted doublets from each sample (nExp = 9% of nuclei per sample). Following doublet removal, nuclei surpassing upper-end RNA QC metrics (>20,000 counts, >6,000 genes detected) were removed. After filtering, Seurat objects were merged. Gene counts were normalized and scaled, as described for the single-nucleus multiome data processing.

The P14 snRNA-seq dataset was assigned to adult BNST inhibitory cluster labels using Seurat. Adult BNST inhibitory clusters were subsetted from the adult snRNA-seq object and randomly downsampled to 10,000 nuclei. Normalization, data scaling and linear dimensionality reduction were performed with the same parameters for P14 and adult inhibitory neuron clusters. Label transfer was then performed as described for the single-nucleus multiome data processing. Extended Data Fig. 12b shows the prediction scores of P14 nuclei mapped onto adult reference cluster labels. To validate the quality of label transfer between adult and P14 datasets, we computed DEGs between P14 clusters post label transfer, as described above, and calculated their background-subtracted, average expression (AddModuleScore) in P14 and adult BNST inhibitory clusters (shown in Extended Data Fig. 12c). Sex DEGs between control females and and control or conditional ERα KO males were calculated for each P14 cluster, as described above for the multiome analysis. Cluster abundance for each group was computed and is plotted in Extended Data Fig. 12d.

**Neonatal bulk nuclear RNA-seq data processing and analysis.** Reads were trimmed to remove Illumina adapters and low-quality basecalls (cutadapt -q 30), and then mapped to the mm10 reference genome using STAR. Technical duplicate reads (identical start and end positions with the same strand orientation and identical molecular identifiers) were removed using the nudup.py python package (https://github.com/tecangenomics/nudup). The number of reads mapping to each gene (including introns) on each strand (-s 1) was calculated with feature-Counts[82], using the mm10.refGene.gtf file. Differential gene expression analysis was performed using DESeq2 (design = ~ treatment) after prefiltering genes by expression (rowMeans ≥ 5).

**Reporting summary**
Further information on research design is available in the Nature Research Reporting Summary linked to this paper.

**Data availability**
All sequencing data generated in this study have been deposited in GEO (GSE144718). The following publicly available datasets were also analysed: MCF7 ERα ChIP–seq (GSE59530), mouse liver ERα ChIP–seq (GSE49993), mouse liver ERα ChIP–seq (GSE52351), mouse uterus ERα ChIP–seq (GSE36455), mouse uterus ERα ChIP–seq dataset (GSE49993), mouse aorta ERα ChIP–seq (GSE52351), mouse mammary gland ERα ChIP–seq (GSE130032), BNST snRNA-seq (GSE126836),

MPOA single-cell RNA-seq (GSE113576) and the Allen Brain Institute Cell Type Database (https://portal.brain-map.org/atlases-and-data/rnaseq/mouse-whole-cortex-and-hippocampus-10x). Source data are provided with this paper.

## Code availability
Custom scripts can be found at https://github.com/gegenhu/estrogen_gene_reg.

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

**Acknowledgements** We thank E. J. Clowney and H. A. Ingraham for comments and discussions, and C. Regan and J. Preall for assistance with single-cell sequencing. This work was performed with assistance from Cold Spring Harbor Laboratory Shared Resources, including the Animal, Next Generation Genomics, Flow Cytometry, Microscopy and Single Cell Core Facilities, which are supported by the National Institutes of Health Cancer Center Support Grant 5P30CA045508. National Institutes of Health grant S10OD028632-01 supports the graphical processing units and large-memory nodes at Cold Spring Harbor Laboratory. This work was supported by funding to J.T. (Stanley Family Foundation, R01 MH113628 and SFARI600568) and B.G. (2T32GM065094 and F31MH124365).

**Author contributions** performed the Nfix CUT&RUN. B.G. and J.T. conceived the study, designed the experiments and wrote the manuscript.

**Competing interests** The authors declare no competing interests.

**Additional information**
**Correspondence and requests for materials** should be addressed to J. Tollkuhn.

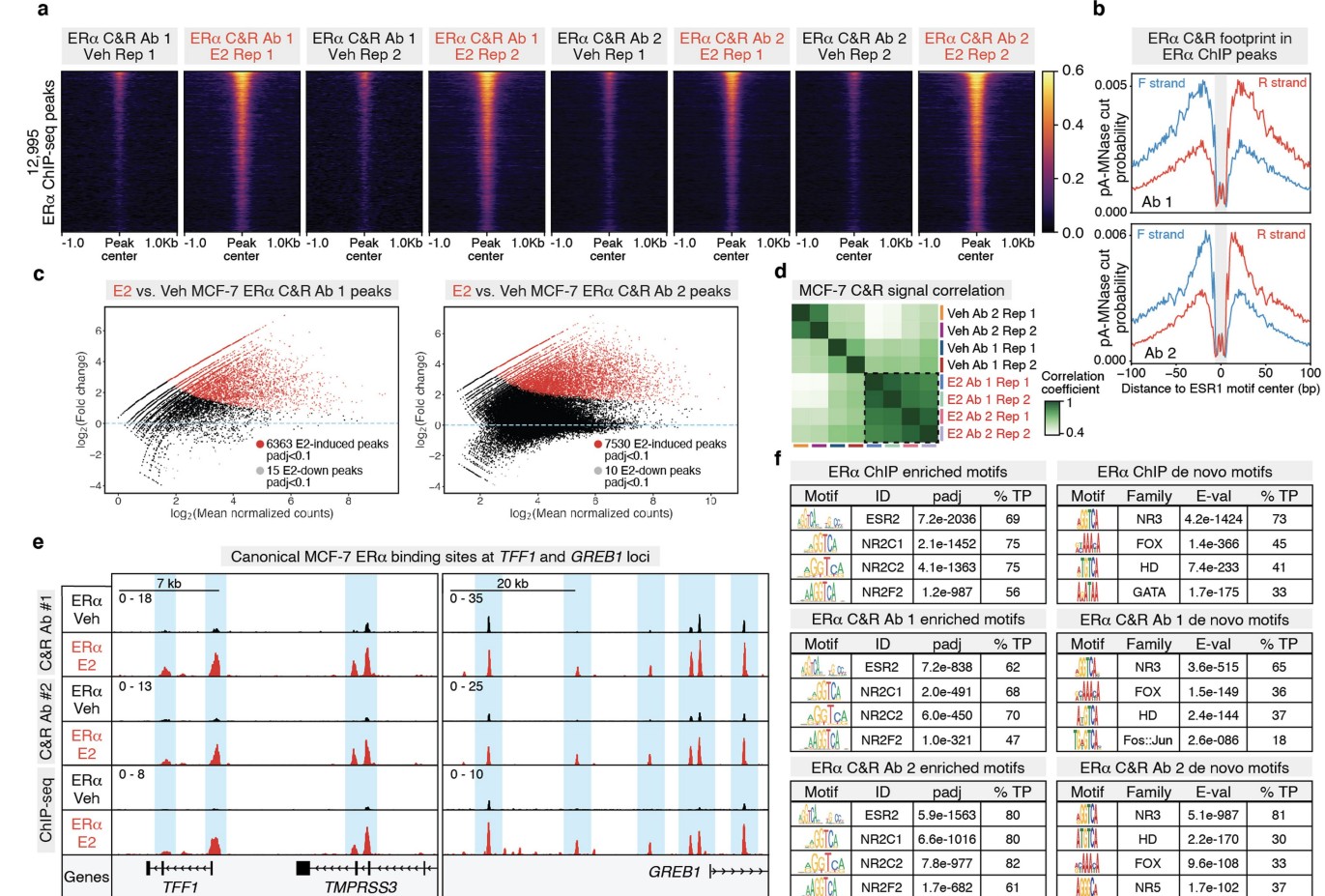

**Extended Data Fig. 1 | Validation of ERα CUT&RUN in MCF-7 cells. a**, Heatmap of mean MCF-7 ERα CUT&RUN CPM ±1Kb around 12,995 17β-estradiol (E2)-induced MCF-7 ERα ChIP-seq peaks (DiffBind, DESeq2, padj < 0.01) for individual replicates (n = 2 per condition and antibody). **b**, pA-MNase-cut footprint (CUT&RUNTools) around ESR1 motif sites (FIMO) detected in ERα ChIP-seq peaks. **c**, MA plots of differential ERα CUT&RUN peaks (DiffBind, DESeq2, padj < 0.1) for ERα antibody #1 (Santa Cruz sc-8002) and ERα antibody #2 (EMD Millipore Sigma 06-935). **d**, Pearson correlation coefficient of CPM-normalized CUT&RUN signal within the consensus peak matrix across ERα CUT&RUN samples. Red text indicates E2 treatment group. **e**, ERα CUT&RUN (both antibodies) and ChIP-seq tracks at canonical MCF-7 ERα target genes (*TFF1*, *GREB1*). **f**, (left) Top enriched motifs (AME) and (right) top *de novo* motifs (DREME) within ERα ChIP-seq peaks, Ab #1 ERα CUT&RUN peaks, and Ab #2 ERα CUT&RUN peaks. % TP = % of peaks called as positive for the indicated motif. *De novo* motifs were classified into motif families using TomTom.

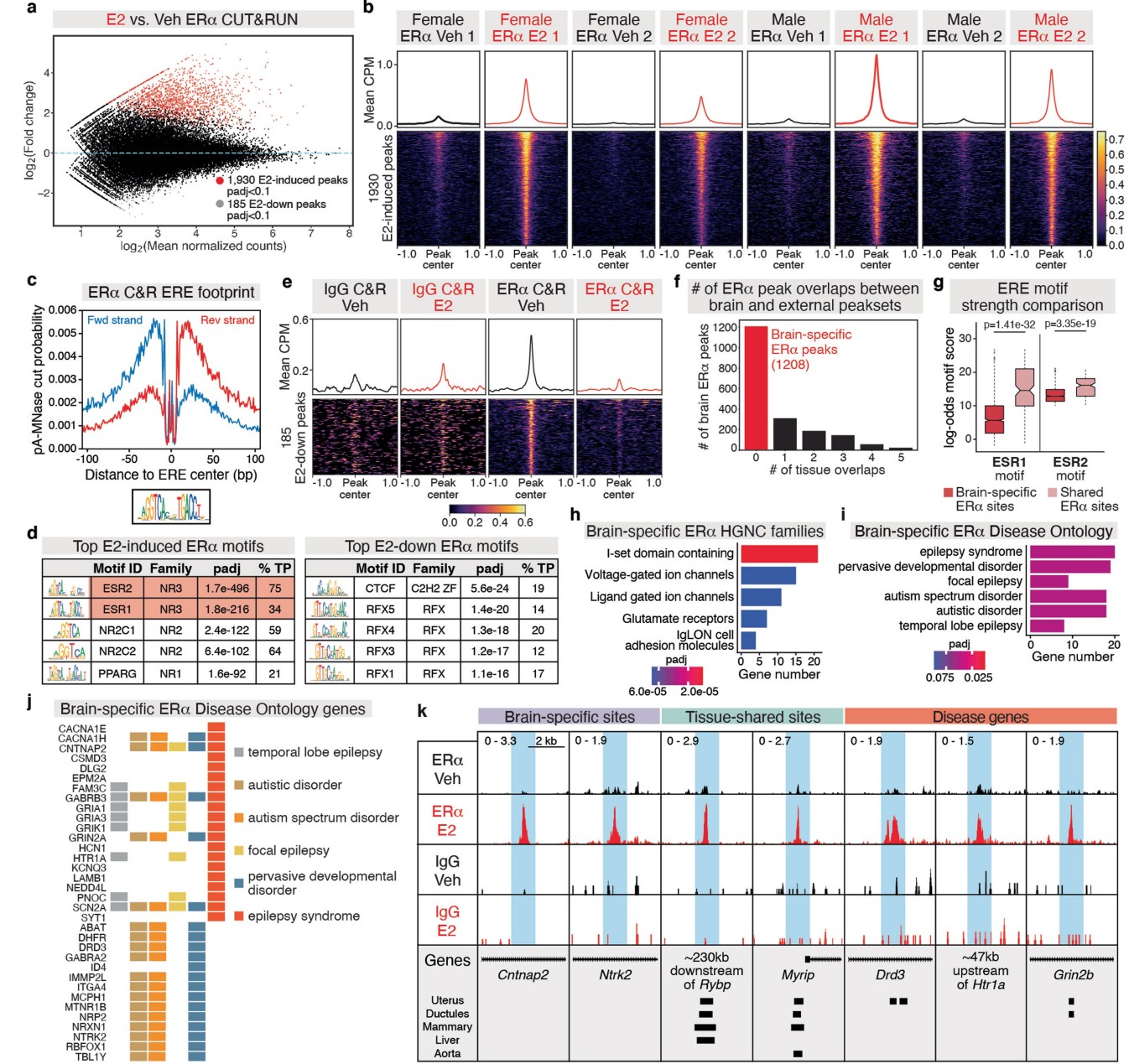

**Extended Data Fig. 2 | Additional analysis of adult brain ERα CUT&RUN dataset. a**, MA plot of differential ERα CUT&RUN peaks (DiffBind, edgeR, padj < 0.1) in adult mouse brain. red dots=E2-induced peaks, grey dots=E2-down peaks. **b**, Heatmap of mean brain ERα CUT&RUN CPM ±1Kb around 1930 E2-induced ERα CUT&RUN peaks (see also Fig. 1b) for individual replicates (n = 2 per condition). **c**, ESR1 motif footprint in ERα peaks (CUT&RUNTools). **d**, Top enriched motifs (AME) in (left) E2-induced ERα peaks and (right) E2-down ERα peaks. **e**, Heatmap of mean brain IgG and ERα CUT&RUN CPM ±1Kb around 185 E2-down ERα peaks. **f**, Number of overlaps between E2-induced ERα peaks and 7 external ERα ChIP-seq peaksets: intersected peaks of uterus 1 and uterus 2, intersected peaks of liver 1 and liver 2, aorta, efferent ductules, and mammary gland. Red indicates brain-specific ERα peaks. **g**, Log-odds motif scores (FIMO) for the ESR1 motif (MA0112.3, left)

and ESR2 motif (MA0258.2, right) in brain-specific (red) and shared (pink) ERα peaks. Boxplot center=median, box boundaries=1st and 3rd quartile, whiskers=1.5*IQR from boundaries. n = 1304 brain-specific ESR1, 139 shared ESR1, 1276 brain-specific ESR2, 157 shared ESR2. p-values from two-sided, Wilcoxon rank-sum test. **h**, Top Hugo Gene Nomenclature Committee (HGNC) gene families (clusterProfiler, padj < 0.1) enriched within brain-specific ERα peak-associated genes. **i**, Top Disease Ontology terms associated with genes nearest to brain-specific ERα peaks (DOSE, padj < 0.1). **j**, Brain-specific ERα peak-associated genes within each enriched Disease Ontology (DO) term (clusterProfiler, padj < 0.1), colored by term **k**, Example brain-specific (*Cntnap2*, *Ntrk2*), shared (*Rybp*, *Myrip*), and disease-associated (*Drd3*, *Htr1a*, *Grin2b*) ERα peaks.

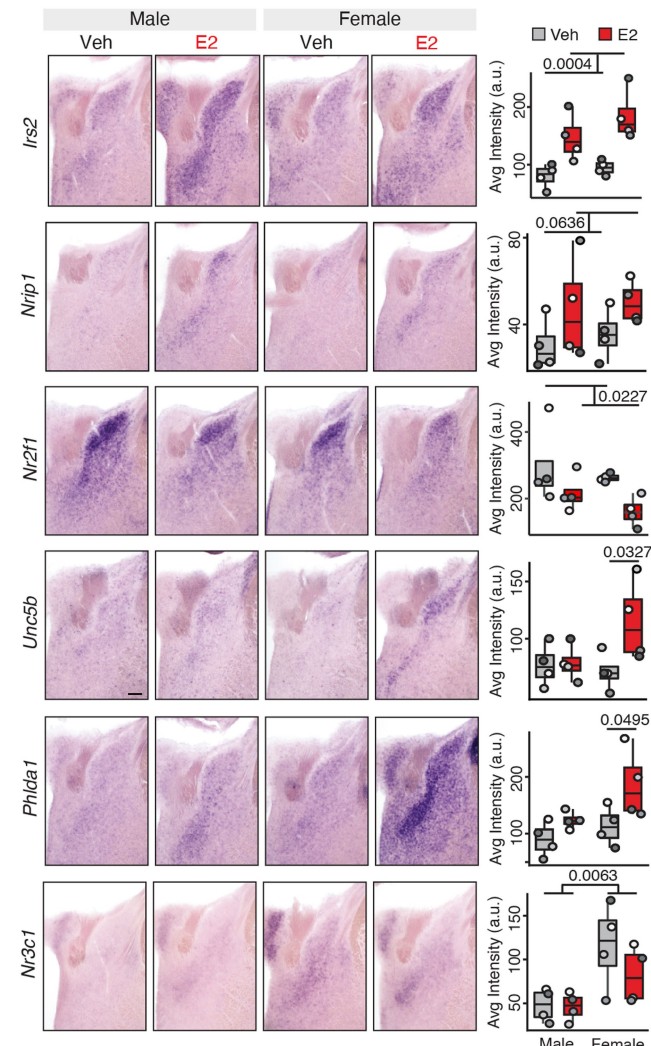

**Extended Data Fig. 3 | Additional BNSTp ISH validation.** In situ hybridization (ISH) validation of additional BNSTp E2-regulated genes identified by RNA-seq (see also Fig. 1e, f). Boxplot center=median, box boundaries=1st and 3rd quartile, whiskers=1.5*IQR from boundaries. p-value from 2-way ANOVA test, n = 4, scale = 200 um.

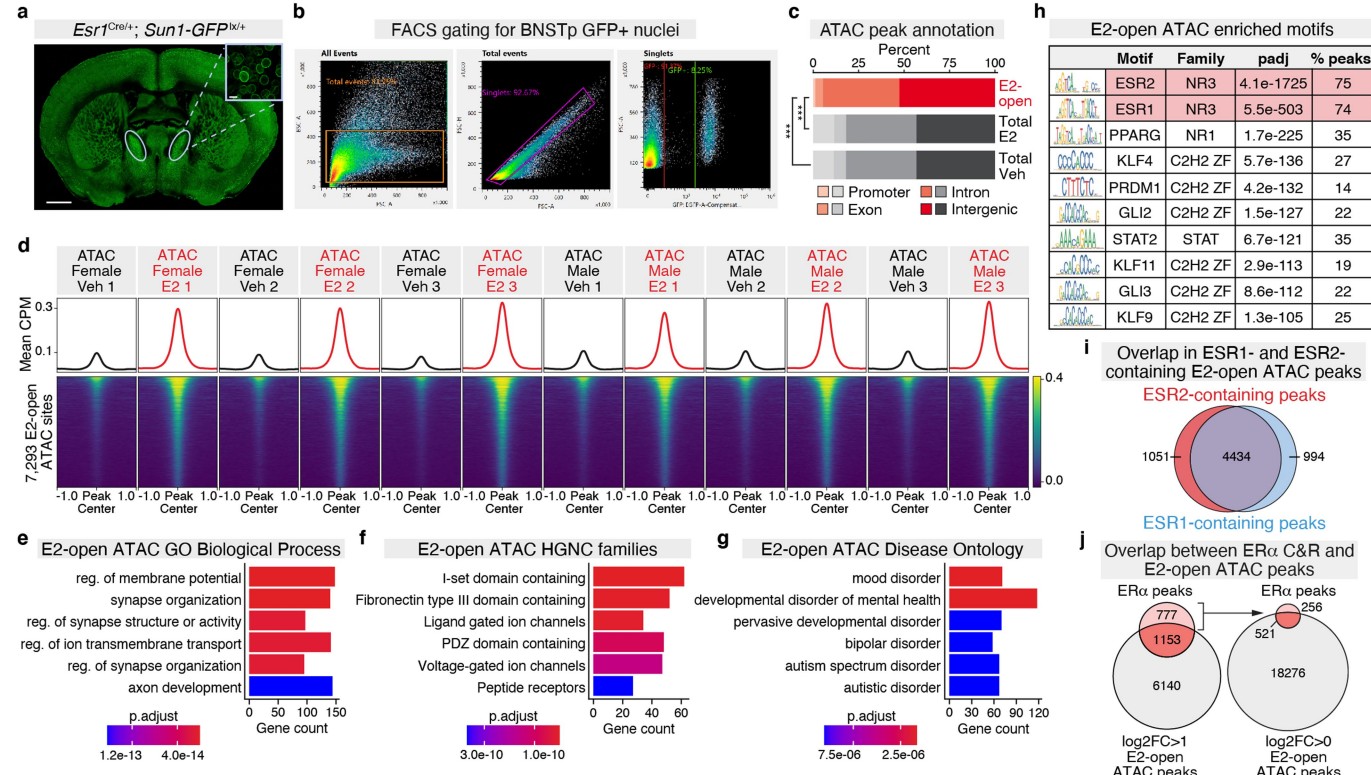

**Extended Data Fig. 4 | Additional analysis of adult BNSTp ATAC-seq. a**, GFP immunofluorescence staining in an adult male *Esr1*<sup>Cre/+</sup>; *Sun-GFP*<sup>lx/+</sup> mouse, scale = 1mm. Inset shows Sun1-GFP signal at nuclear membrane, scale = 10um. **b**, Fluorescence-activated cell sorting (FACS) gating strategy for isolating BNSTp GFP+ nuclei for ATAC-seq. **c**, Proportion of E2-open ATAC peaks (red), total E2 ATAC peaks (black), and total Veh ATAC peaks (black) annotated to promoters (±1Kb around TSS), exons, introns, and intergenic regions. E2-open ATAC peaks have a significantly lower proportion of `peaks annotated to gene promoters than total vehicle (11% vs 1%, Fisher's Exact Test, p = 4.6 x 10<sup>−260</sup>) and total E2 (11% vs 1%, Fisher's Exact Test, p = 4.3 x 10<sup>−267</sup>) peaks. ***p < 0.001. **d**, Heatmap of mean BNSTp *Esr1*+ ATAC CPM ±1Kb around 7293 E2-open ATAC

peaks for individual female and male replicates (n = 3 per condition) (see also Fig. 1g). **e**–**g**, Top Gene Ontology (GO) Biological Process terms (**e**), HGNC gene families (**f**), and DO terms (**g**) enriched within E2-open ATAC peak-associated genes (clusterProfiler, padj < 0.1). **h**, Top motifs enriched in E2-open ATAC-seq peaks (AME). % of peaks containing motifs determined with FIMO. **i**, Overlap in E2-open ATAC peaks containing the ESR1 motif (blue) and the ESR2 motif (red), identified using FIMO. The majority of peaks (4434/6479) containing either motif are the same. **j**, (left) Overlap between brain ERα CUT&RUN peaks and E2-open ATAC peaks (log2FC > 1). (right) Overlap between remaining 777 brain ERα CUT&RUN peaks and log2FC > 0 E2-open ATAC peaks.

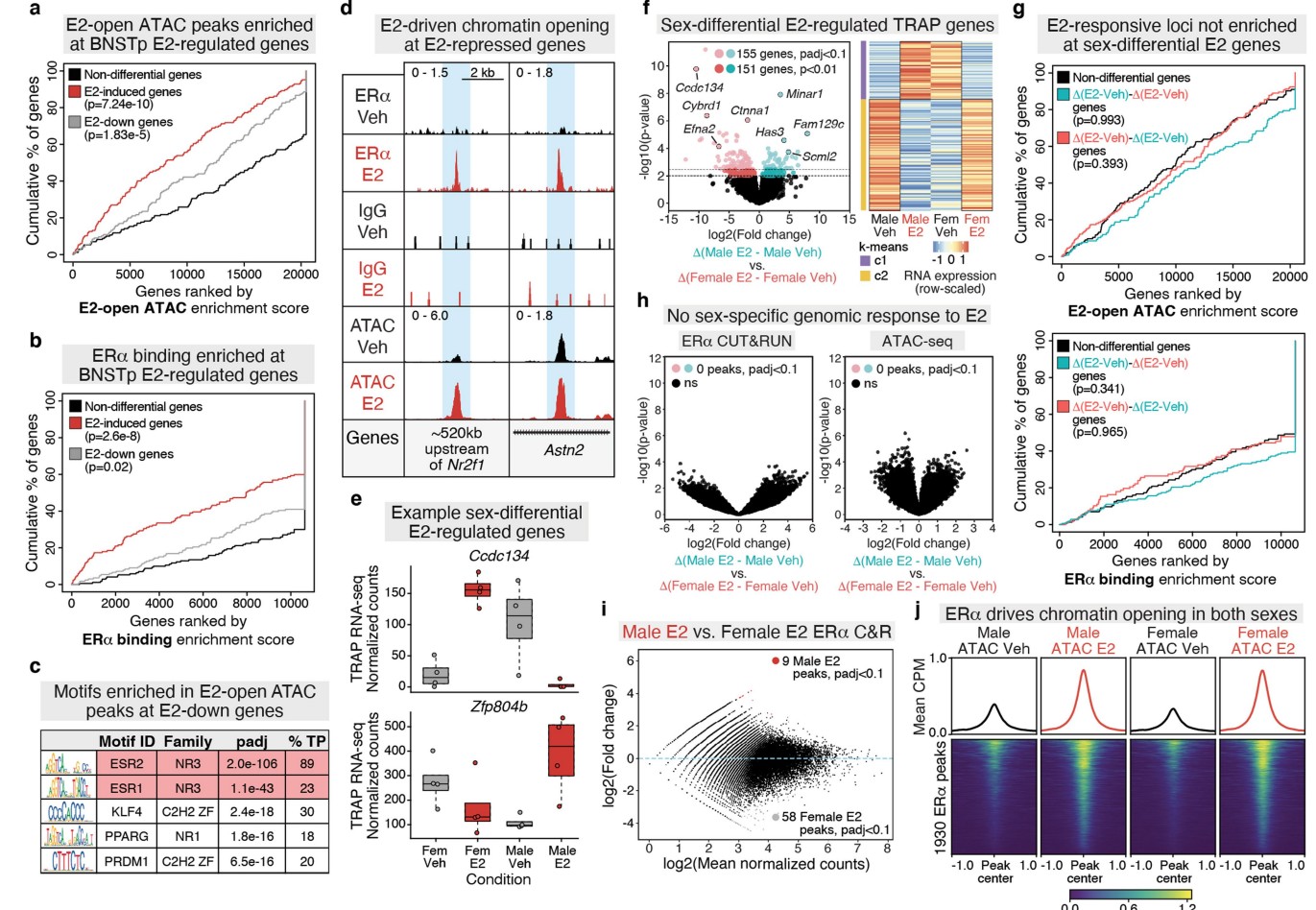

**Extended Data Fig. 5 | Integration of adult RNA-seq, ATAC-seq, and CUT&RUN datasets. a–b**, BETA enrichment of E2-open ATAC peaks (**a**) and brain ERα CUT&RUN peaks (**b**) at E2-induced and E2-down genes identified by RNA-seq (DESeq2, p < 0.01) relative to a background of non-differential, expressed genes. **c**, Top enriched motifs (AME) in E2-open ATAC peaks ±350Kb around E2-down genes (identified with BETA). **d**, Example E2-open ATAC peaks/ERα peaks at E2-repressed genes, *Nr2f1* and *Astn2*. **e**, Normalized counts for example genes (*Ccdc134*, *Zfp804b*) with a sex-dependent response to E2 treatment. Boxplot center=median, box boundaries=1st and 3rd quartile, whiskers=1.5*IQR from boundaries. n = 4. **f**, (Left) Volcano plot of sex-dependent, E2-responsive genes; light blue and red dots (DESeq2,

padj < 0.1), dark blue and red dots (DESeq2, padj < 0.1). (Right) Mean, normalized expression of sex-dependent, E2-responsive genes (DESeq2, padj < 0.1), grouped by k-means clustering. **g**, Lack of significant enrichment of E2-open ATAC peaks (top) and ERα peaks (bottom) at sex-dependent, E2-reponsive genes relative to a background of non-differential, expressed genes (BETA). **h**, Volcano plots of sex-dependent, E2-responsive ERα CUT&RUN peaks (edgeR, padj < 0.1) (left) and ATAC peaks (edgeR, padj < 0.1). **i**, MA plot of male E2 vs. female E2 ERα CUT&RUN peaks (DiffBind, edgeR, padj < 0.1); red dots=male E2-biased peaks, grey dots=female E2-biased peaks. **j**, Heatmap of mean ATAC CPM, split by sex and treatment, ±1Kb around E2-induced ERα peaks.

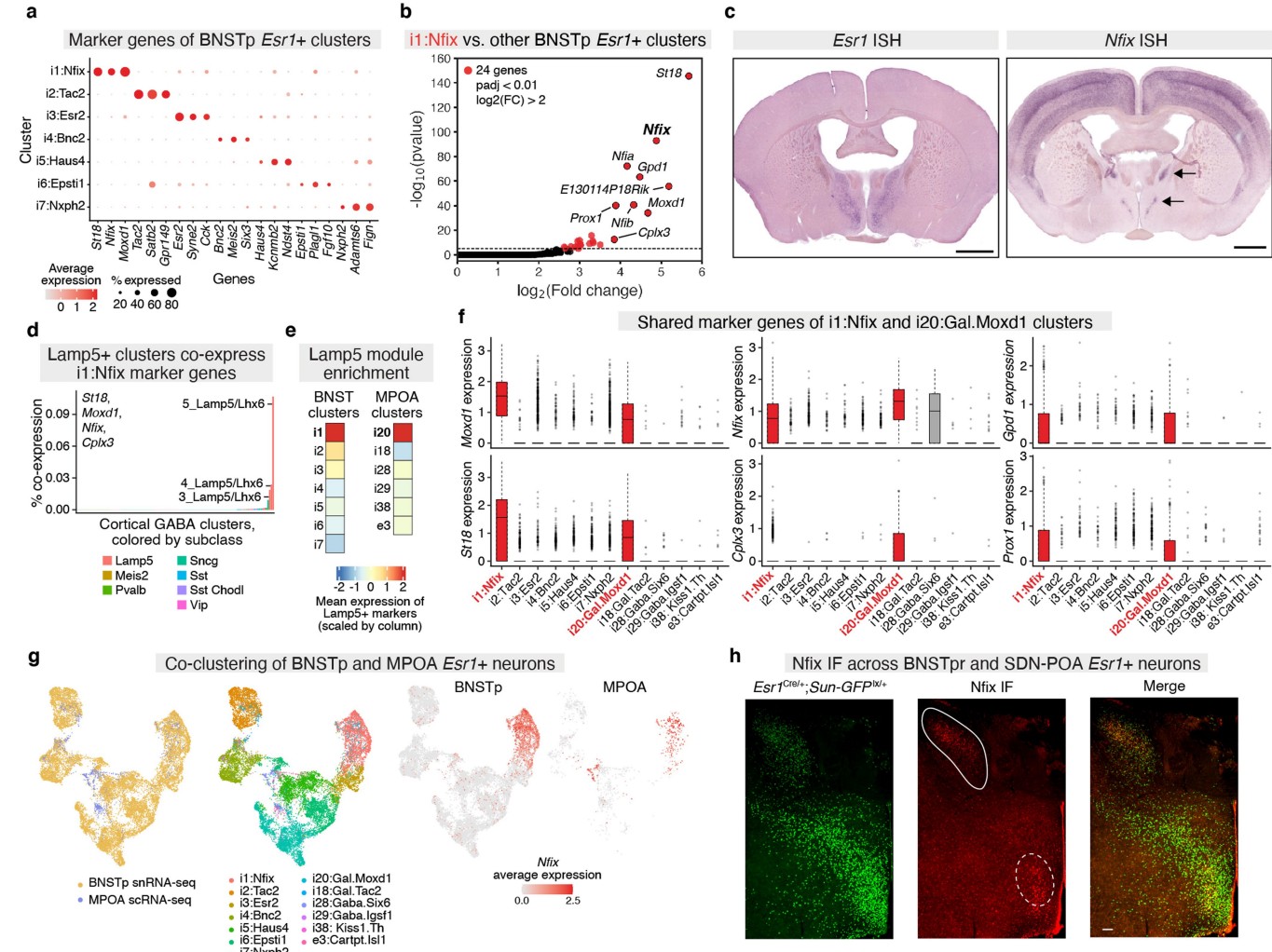

**Extended Data Fig. 6 | Characterization of a shared BNSTp/MPOA transcriptomic cluster. a**, Dotplot of top marker genes for each adult BNSTp *Esr1*+ GABAergic cluster (Wilcoxon rank-sum test, padj < 0.05).
**b**, Differentially-expressed genes between the i1:Nfix cluster and the other six BNSTp *Esr1*+ inhibitory neuron clusters (DESeq2, log2FC > 2, padj < 0.01). **c**, ISH of adult gonadectomized, Veh-treated male (*Esr1*) and adult male (*Nfix*) mouse. Arrows denote *Nfix* ISH staining in BNSTp (dorsal) and POA (ventral). Scale = 1 mm. **d**, Co-expression of top i1:Nfix marker genes (*St18*, *Moxd1*, *Nfix*, *Cplx3*) in individual BICCN cortical and hippocampal scRNA-seq GABAergic clusters, colored by subclass. Co-expression defined as % of cells per cluster with non-zero counts for all 4 marker genes. **e**, Mean expression of *Lamp5*+ subclass marker genes (Wilcoxon rank-sum test, avg_log_FC > 0.75, <40%

expression in non-*Lamp5*+ neurons, padj < 0.05) in BNSTp (left) and MPOA (right) *Esr1*+ clusters, scaled across clusters within each brain region.
**f**, Normalized expression of top marker genes (*Moxd1*, *St18*, *Nfix*, *Cplx3*, *Gpd1*, *Prox1*) shared between i1:Nfix and i20:Gal.Moxd1 (labeled in red). Boxplot center=median, box boundaries=1st and 3rd quartile, whiskers=1.5*IQR from boundaries. n = 297 i20:Gal.Moxd1 cells, 2459 i1:Nfix cells. **g**, UMAP visualization of integrated BNSTp and MPOA *Esr1*+ clusters, demonstrating shared *Nfix* expression across datasets (see also Fig. 2e). **h**, GFP (left) and Nfix (middle) immunofluorescence staining in an adult male *Esr1*[Cre/+];*Sun-GFP*[lx/+] mouse. Solid white circle indicates BNSTp; dotted white circle indicates SDN-POA. Scale = 100 um.

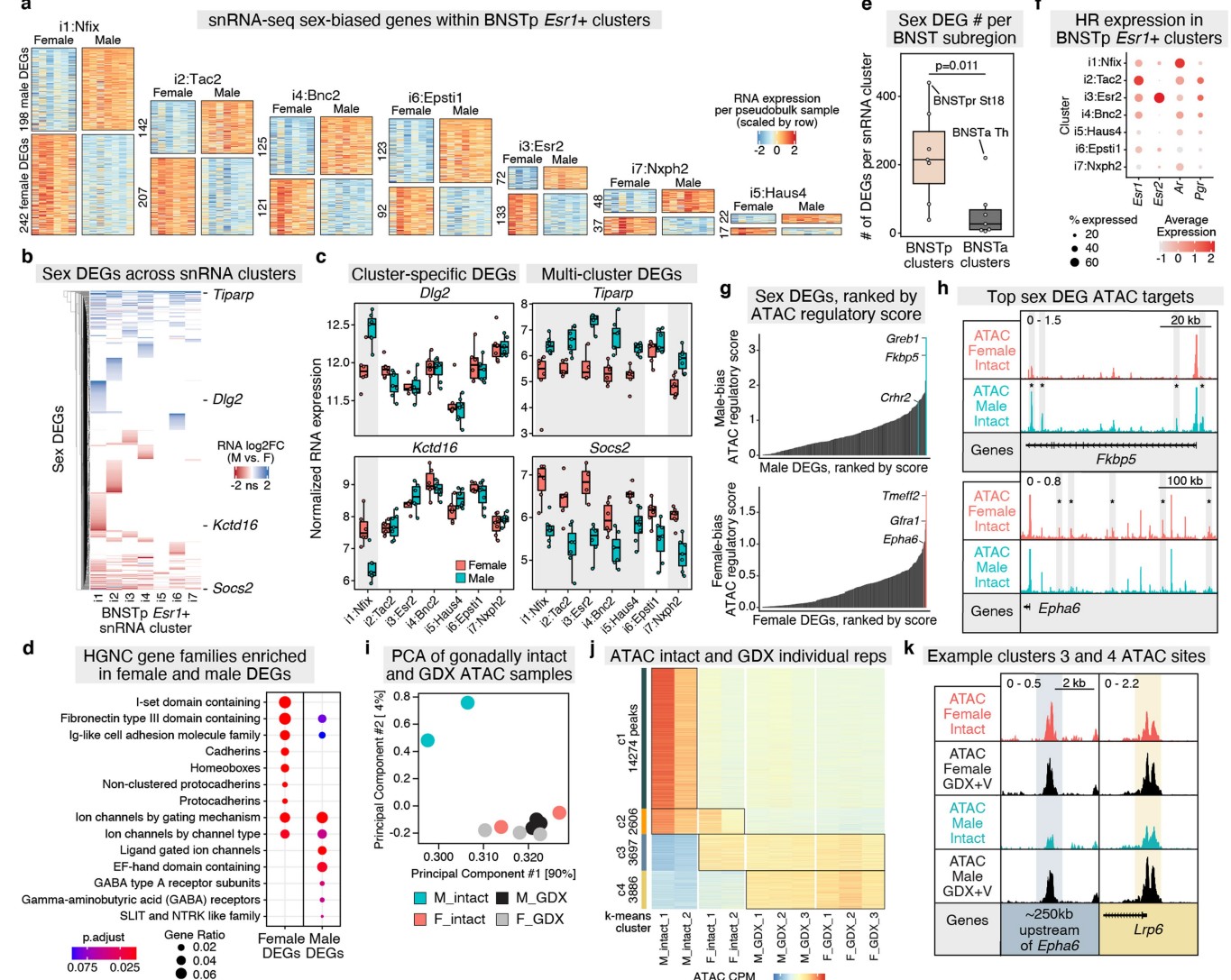

**Extended Data Fig. 7 | Additional analysis of BNSTp sex DEGs and gonadally intact ATAC-seq. a**, Normalized, pseudo-bulk expression of sex DEGs identified within each BNSTp *Esr1+* cluster (DESeq2, padj < 0.1). Each heatmap column corresponds to a pseudo-bulk sample (gene counts aggregated across cells in sample). **b**, Hierarchical clustering of log2FC values for sex DEGs called as significant in at least one BNSTp *Esr1+* cluster. Sex DEGs with non-significant differential expression colored in white. **c**, Example sex DEGs with significant differential expression in a single *Esr1+* cluster (*Dlg2*, *Kctd16*) and in multiple *Esr1+* clusters (*Tiparp*, *Socs2*). Boxplot center=median, box boundaries=1st and 3rd quartile, whiskers=1.5*IQR from boundaries. n = 4-7 female pseudo-bulk replicates, 6-8 male pseudo-bulk replicates. **d**, Top HGNC gene families (clusterProfiler, padj < 0.1) enriched within female-biased DEGs (left) and male-biased DEGs (right) relative to non-differential, expressed genes. **e**, Number of sex DEGs per cluster in *Esr1+* clusters annotated to the BNST posterior (BNSTp, n = 7 clusters) and anterior (BNSTa, n = 7 clusters)

subregions. Boxplot center=median, box boundaries=1st and 3rd quartile, whiskers=1.5*IQR from boundaries. p-value from two-sided, Wilcoxon rank-sum test. **f**, Dotplot of sex hormone receptor (HR) expression across BNSTp *Esr1+* clusters. **g**, Barplots of (top) male-biased DEGs ranked by male-biased ATAC peak regulatory potential score and (bottom) female-biased DEGs ranked by female-biased ATAC peak regulatory potential score. Higher score indicates higher density of sex-biased ATAC peaks around the TSS of sex DEGs. **h**, Example sex DEGs (*Fkbp5*, *Epha6*) with high density of sex-biased ATAC peaks. *sex-biased ATAC peak. **i**, Principal component analysis (PCA) of gonadally intact and gonadectomized (GDX), Veh-treated ATAC CPM values within the consensus peak matrix. **j**, Heatmap of ATAC CPM for gonadally intact (n = 2 per condition) and GDX, Veh-treated ATAC samples (n = 3 per condition) at differential peaks (edgeR, glmQLFTest, padj < 0.01), grouped by k-means clustering (see also Fig. 2i). **k**, Example differential ATAC peaks in k-means clusters c3 (left) and c4 (right).

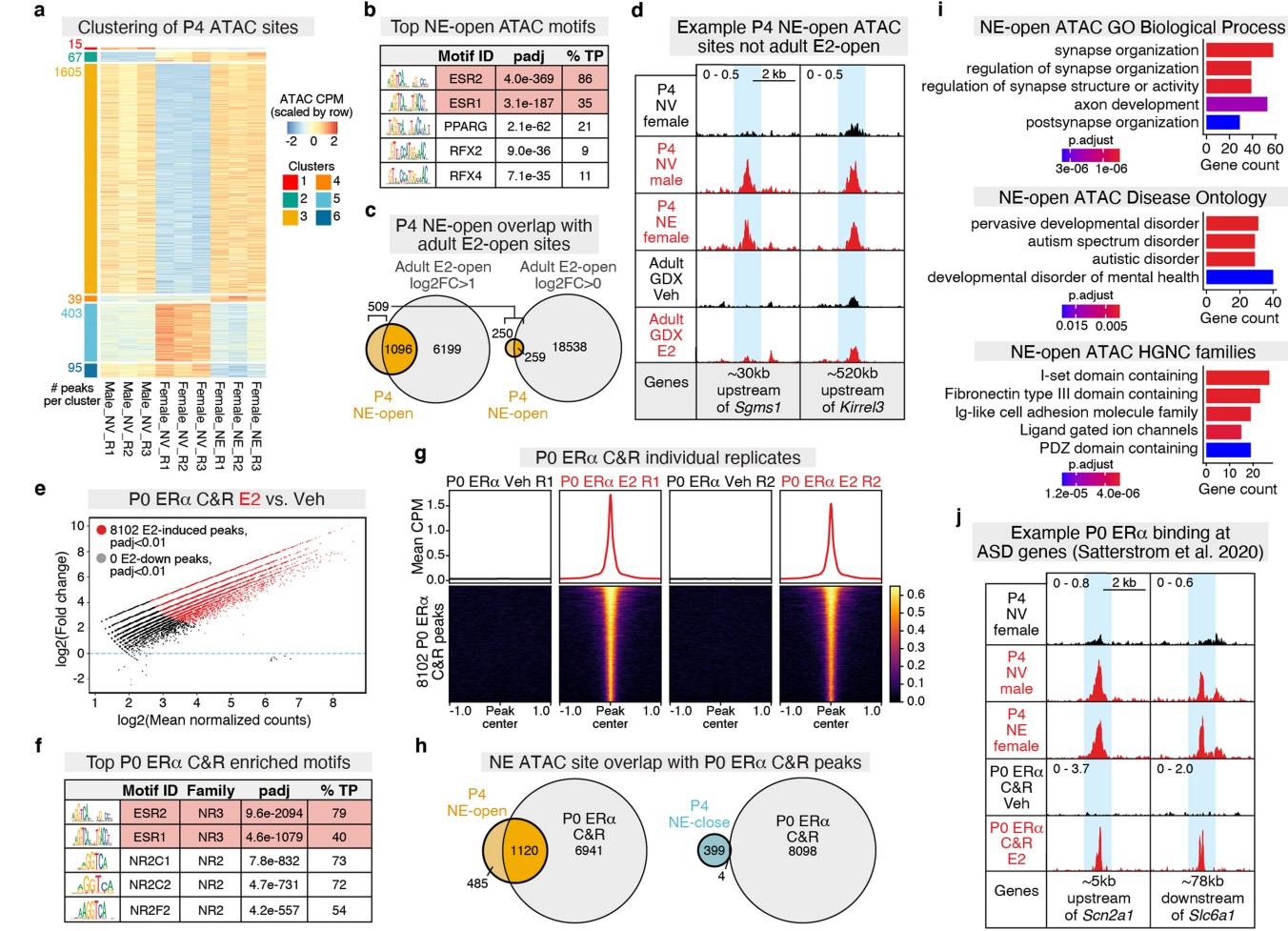

**Extended Data Fig. 8 | P4 ATAC-seq and P0 ERα CUT&RUN analysis.**
**a**, Heatmap of mean ATAC CPM for P4 NV male, NV female, and NE female individual replicates (n = 3 per condition) at differential peaks (edgeR, glmQLFTest, padj < 0.1), grouped by hierarchical clustering (cutree, k = 6). Clusters c3 and c5 correspond to NE-open and NE-close sites, respectively, shown in Fig. 4a. **b**, Top enriched motifs (AME) in NE-open ATAC peaks. **c**, (left) Overlap between P4 NE-open ATAC peaks and adult E2-open ATAC peaks (log2FC > 1). (right) Overlap between remaining 509 P4 NE-open ATAC peaks and log2FC > 0 E2-open ATAC peaks. **d**, Example P4 NE-open ATAC peaks not detected as E2-induced in adult E2-open ATAC peakset. **e**, MA plot of P0 female E2 vs. female Veh ERα CUT&RUN peaks (DiffBind, DESeq2, padj < 0.01); red dots=E2-induced peaks, grey dots=E2-down peaks. **f**, Top enriched motifs (AME) in P0 E2-induced ERα peaks. **g**, Heatmap of mean P0 ERα CUT&RUN CPM ±1Kb around 8102 E2-induced ERα peaks for individual replicates (n = 2 per condition). **h**, (left) Overlap between P4 NE-open ATAC peaks and P0 E2-induced ERα peaks. (right) Overlap between P4 NE-close ATAC peaks and P0 E2-induced ERα peaks. **i**, (top) Top GO Biological Process terms (clusterProfiler, padj < 0.1), (middle) DO terms (clusterProfiler, padj < 0.1), and (bottom) HGNC gene families (clusterProfiler, padj < 0.1) enriched within P4 NE-open peak-associated genes. **j**, Example P0 ERα peaks overlapping P4 NE-open peaks at high-confidence ASD candidate genes, *Scn2a1* and *Slc6a1*.

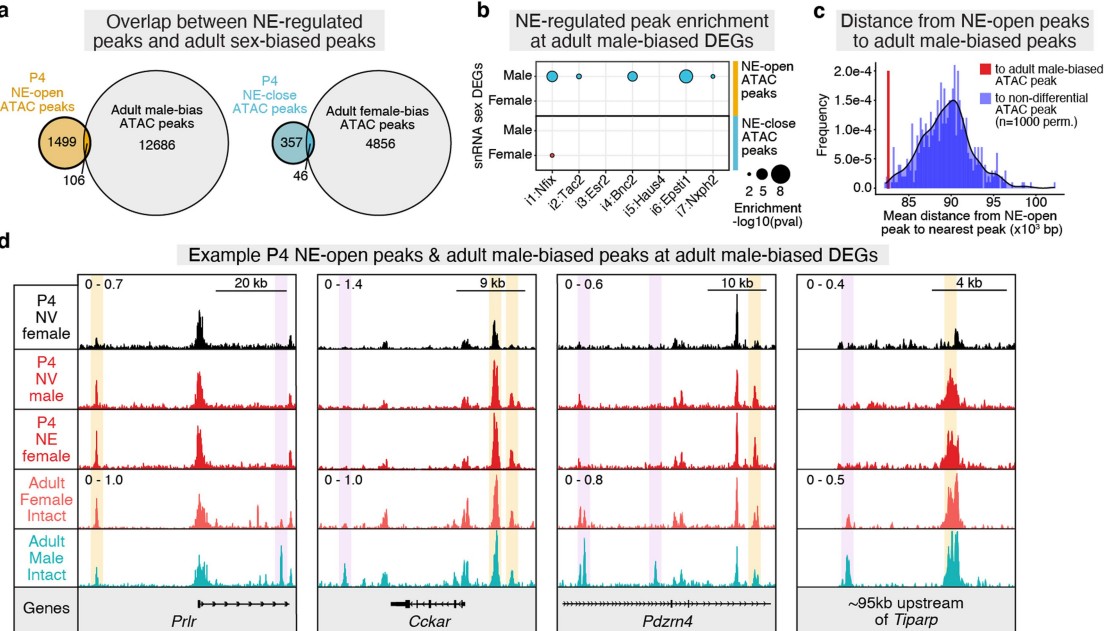

**a** Overlap between NE-regulated peaks and adult sex-biased peaks

**b** NE-regulated peak enrichment at adult male-biased DEGs

**c** Distance from NE-open peaks to adult male-biased peaks

**d** Example P4 NE-open peaks & adult male-biased peaks at adult male-biased DEGs

**Extended Data Fig. 9 | Comparison of P4 and adult *Esr1*+ ATAC-seq. a**, (left) Overlap between P4 NE-open ATAC peaks and gonadally intact adult male-biased ATAC peaks. (right) Overlap between P4 NE-close ATAC peaks and gonadally intact adult female-biased ATAC peaks. **b**, Dotplot of BETA enrichment p-values for P4 NE-open ATAC peaks (top) and NE-close ATAC peaks (bottom) at adult BNST snRNA-seq sex DEGs relative to a background of non-differential, expressed genes (see also Fig. 2h). **c**, Histogram of mean distance between P4 NE-open peaks and nearest gonadally intact adult

male-biased ATAC peak (red line) vs. nearest chromosome-matched, non-differential adult ATAC peak (n = 1000 permutations) (blue histogram). Mean distance between P4 NE-open peaks and adult male-biased peaks is significantly smaller than the expected distribution (Permutation test, p = 0.007). **d**, Example adult male-biased DEGs (*Prlr*, *Cckar*, *Pdzrn4*, *Tiparp*) with neighboring P4 NE-open (highlighted in yellow) and adult male-biased ATAC peaks (highlighted in purple).

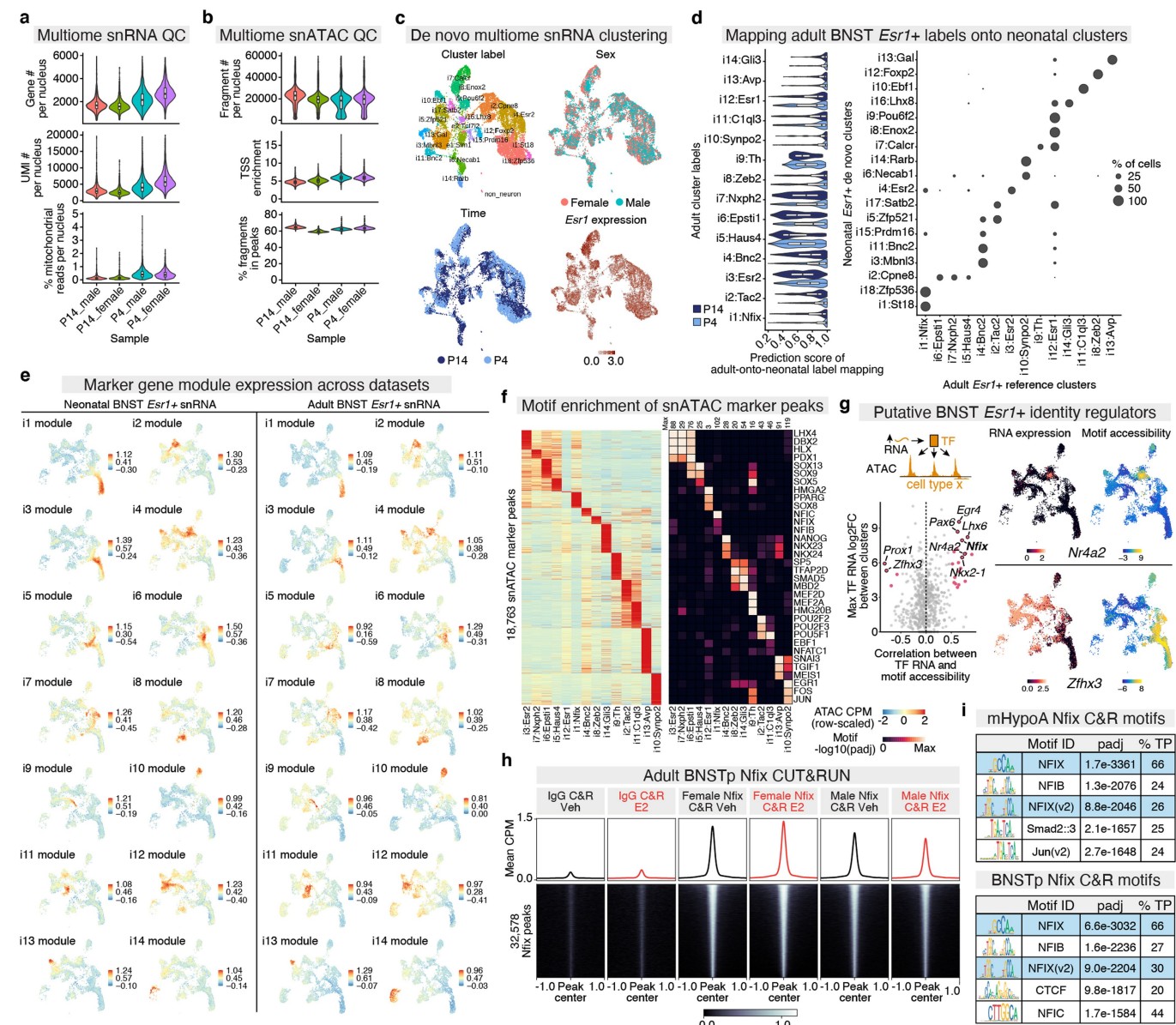

**Extended Data Fig. 10 | Additional analysis of neonatal BNST *Esr1*+ single-nucleus multiome dataset. a–b**, RNA (**a**) and ATAC (**b**) quality control (QC) metrics for neonatal (P4, P14) single-nucleus multiome experiments, split by timepoint and sex. Boxplot center=median, box boundaries=1st and 3rd quartile, whiskers=minimum and maximum values. n = 4265 P14_male, 3148 P14_female, 3128 P4_male, 4295 P4_female. **c**, UMAPs of *de novo* clustering of neonatal multiome snRNA data, colored by cluster identity (top left), sex (top right), timepoint (bottom left), and *Esr1* expression (bottom right). **d**, (left) Prediction scores of adult-to-neonatal label transfer for each adult BNST *Esr1*+ reference cluster, split by timepoint. Boxplot center=median, box boundaries=1st and 3rd quartile, whiskers=minimum and maximum values. n = 14836 cells. (right) % of nuclei in each neonatal *de novo* cluster that mapped to each adult BNST *Esr1*+ cluster. **e**, UMAPs of neonatal marker gene module

expression in neonatal dataset (left) and adult dataset (right). **f**, (left) Heatmap of pseudo-bulk ATAC CPM at 18783 marker peaks for neonatal multiome clusters. (right) Top three motifs enriched in marker peaks for each multiome cluster. **g**, (left) Correlation analysis of TF expression and motif accessibility across cells. Putative identity regulator TFs colored in pink. (right) TF RNA expression, activity score, and motif deviation UMAPs of example putative BNST *Esr1*+ neuron identity regulators, *Nr4a2* and *Zfhx3* (see also Fig. 3c, d). **h**, Heatmap of mean cortical IgG and BNSTp Nfix CUT&RUN CPM ±1Kb around 32,578 consensus Nfix peaks. **i**, Top motifs enriched (AME) in (top) 30,825 mHypoA cell Nfix CUT&RUN peaks (MACS2, q < 0.01) and in (bottom) 32,578 consensus BNSTp Nfix CUT&RUN peaks (MACS2, q < 0.01; peaks intersected across treatment and sex). %TP=% of peaks called as positive for the indicated motif.

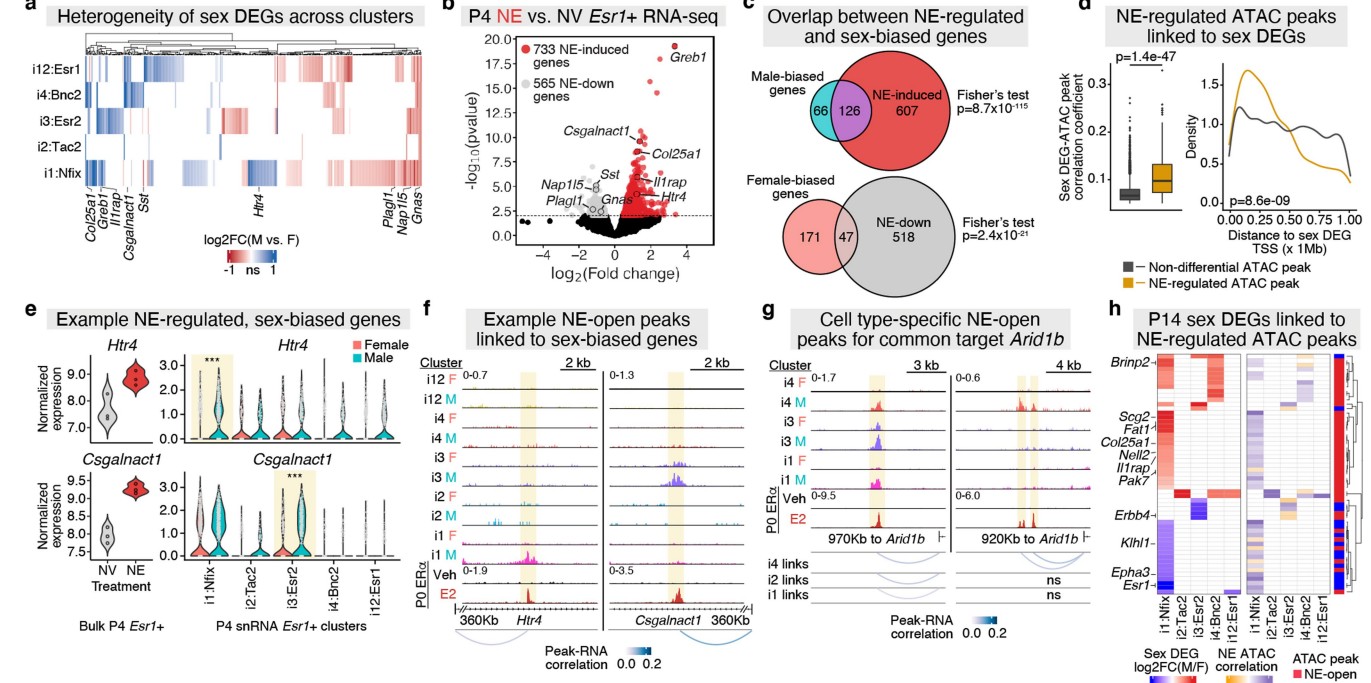

**Extended Data Fig. 11 | Sex differences in single-nucleus multiome dataset.**
**a**, Hierarchical clustering of log2FC values for P4 sex DEGs detected in *Esr1+*
inhibitory neuron clusters (see also Fig. 3e). Sex DEGs with non-significant
differential expression colored in white. **b**, Neonatal E2 (NE) vs. neonatal
vehicle (NV) female nuclear RNA-seq on P4 BNST *Esr1+* cells; grey, red dots
(DESeq2, padj < 0.1). **c**, (top) Overlap between NE-induced genes and P4
multiome male-biased genes. (bottom) Overlap between NE-downregulated
genes and female-biased genes. **d**, (left) Pearson's correlation coefficient
values for non-differential (grey) and NE-regulated (gold) ATAC peaks that
correlate with P4 sex DEG expression. Boxplot center=median, box
boundaries=1st and 3rd quartile, whiskers=1.5*IQR from boundaries. n = 5169
non-differential, 244 NE-regulated. p-value from two-sided, Wilcoxon

rank-sum test. (right) Distance between non-differential (grey) and
NE-regulated (gold) ATAC peaks to P4 sex DEG transcription start sites (TSS).
p-value from Kolmogorov-Smirnov test. **e**, Example P4 sex-biased genes that
are also NE-regulated, *Htr4* (top) and *Csgalnact1* (bottom). (left) n = 3, (right)
n = 887 i1:Nfix female cells, 676 i1:Nfix male cells, 404 i3:Esr2 female cells, 550
i3:Esr2 male cells. **f**, Tracks for NE-open ATAC peaks that correlate with
NE-regulated, sex-biased targets, *Htr4* and *Csgalnact1*. **g**, Different NE-open
ATAC peaks across i1:Nfix, i3:Esr2, and i4:Bnc2 neurons correlated with a
common male-biased target, *Arid1b*. **h**, Heatmaps indicating (left) RNA log2FC
of P14 sex DEGs and (right) Pearson's correlation coefficient of NE-open (red)
and -close (blue) ATAC peaks linked to sex DEGs within each cluster.
Non-significant genes and correlation values colored in white.

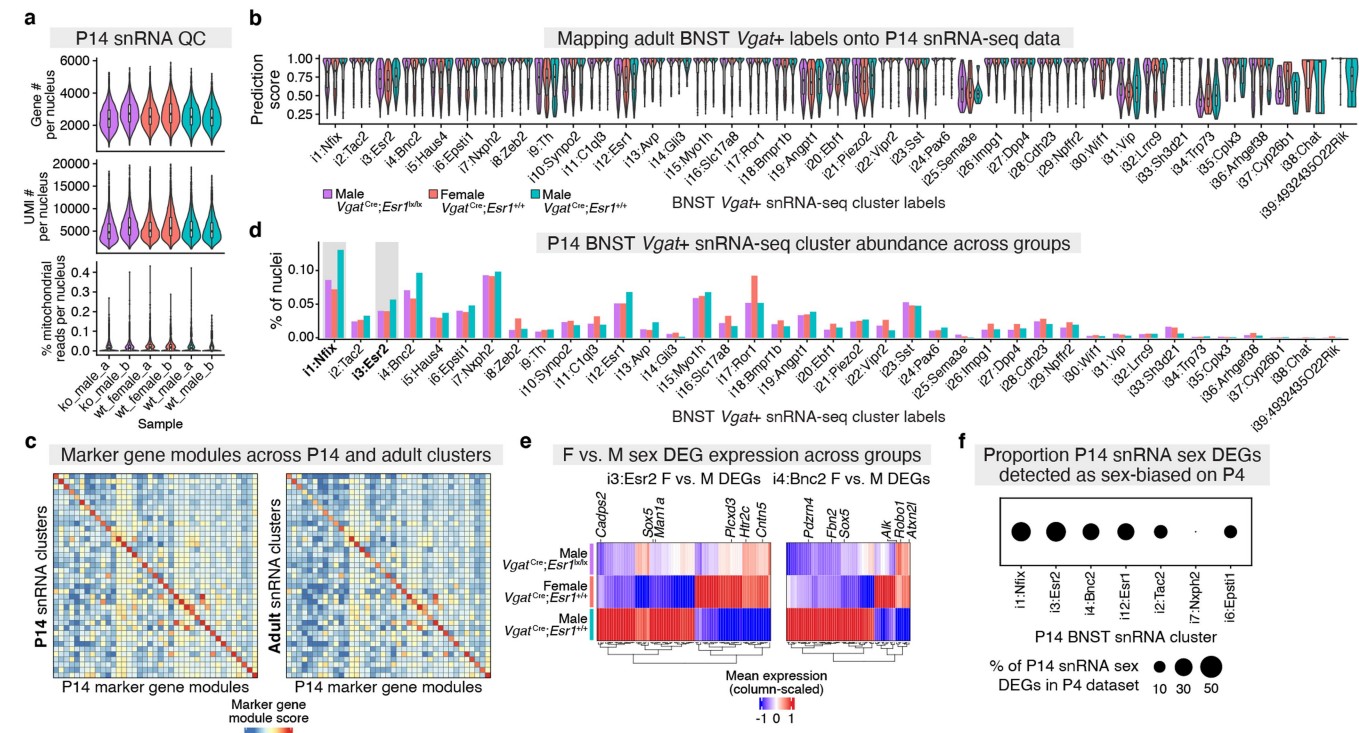

**Extended Data Fig. 12 | Additional analysis of P14 BNST *Vgat*+ snRNA-seq dataset. a**, RNA QC metrics for P14 BNST *Vgat*+ snRNA-seq experiment, split by sample. a and b refer to technical replicates. Boxplot center=median, box boundaries=1st and 3rd quartile, whiskers=minimum and maximum values. n = 6355 ko_male_a, 5614 ko_male_b, 7184 wt_female_a, 6367 wt_female_b, 6881 wt_male_a, 6561 wt_male_b. **b**, Prediction scores of adult-to-P14 label transfer for each adult BNST *Vgat*+ reference cluster, split by group. Boxplot center=median, box boundaries=1st and 3rd quartile, whiskers=minimum and maximum values. n = 38962 cells. **c**, Heatmap of P14 marker gene mean module score in P14 BNST *Vgat*+ clusters (left) and adult BNST *Vgat*+ clusters (right). **d**, Proportion of total P14 *Vgat*+ nuclei in each *Vgat*+ cluster, separated by group. Adult male-biased *Esr1*+ clusters i1:Nfix and i3:Esr2 are indicated in grey. **e**, Heatmap of mean expression of sex DEGs in (left) i3:Esr2 and (right) i4:Bnc2 clusters, scaled across experimental groups (see also Fig. 4b). **f**, % of sex DEGs in P14 *Vgat*+/*Esr1*+ clusters with that are also detected as sex-biased on P4 in corresponding multiome clusters.

**Extended Data Table 1 | Riboprobe sequences used for *in situ* hybridization**

| Gene | RefSeqID | Range |
|------|----------|-------|
| Brinp2 | NM_207583.2 | 2221-3115 |
| Esr1 | NM_001302532.1 | 1263-2243 |
| Enah | NM_001083120.2 | 2597-3067 |
| Irs2 | NM_001081212.1 | 1479-2249 |
| Nfix | NM_001081981.3 | 477-1825 |
| Nr2f1 | NM_010151.2 | 642-1408 |
| Nr3c1 | NM_008173.3 | 416-1385 |
| Nrip1 | NM_173440.2 | 3395-3975 |
| Phlda1 | NM_009344.3 | 731-1366 |
| Rcn1 | NM_009037.2 | 711-1373 |
| Tle3 | NM_001083927.1 | 1980-2982 |
| Unc5b | NM_029770.2 | 2318-3161 |

List of transcript sequences used as riboprobes for *in situ* hybridization experiments.1

# nature research

# Reporting Summary

Nature Research wishes to improve the reproducibility of the work that we publish. This form provides structure for consistency and transparency in reporting. For further information on Nature Research policies, see our Editorial Policies and the Editorial Policy Checklist.

## Statistics

For all statistical analyses, confirm that the following items are present in the figure legend, table legend, main text, or Methods section.

| n/a | Confirmed | |
|---|---|---|
| ☐ | ☒ | The exact sample size (*n*) for each experimental group/condition, given as a discrete number and unit of measurement |
| ☐ | ☒ | A statement on whether measurements were taken from distinct samples or whether the same sample was measured repeatedly |
| ☐ | ☒ | The statistical test(s) used AND whether they are one- or two-sided *Only common tests should be described solely by name; describe more complex techniques in the Methods section.* |
| ☐ | ☒ | A description of all covariates tested |
| ☐ | ☒ | A description of any assumptions or corrections, such as tests of normality and adjustment for multiple comparisons |
| ☐ | ☒ | A full description of the statistical parameters including central tendency (e.g. means) or other basic estimates (e.g. regression coefficient) AND variation (e.g. standard deviation) or associated estimates of uncertainty (e.g. confidence intervals) |
| ☐ | ☒ | For null hypothesis testing, the test statistic (e.g. *F*, *t*, *r*) with confidence intervals, effect sizes, degrees of freedom and *P* value noted *Give P values as exact values whenever suitable.* |
| ☒ | ☐ | For Bayesian analysis, information on the choice of priors and Markov chain Monte Carlo settings |
| ☐ | ☒ | For hierarchical and complex designs, identification of the appropriate level for tests and full reporting of outcomes |
| ☒ | ☐ | Estimates of effect sizes (e.g. Cohen's *d*, Pearson's *r*), indicating how they were calculated |

*Our web collection on statistics for biologists contains articles on many of the points above.*

## Software and code

Policy information about availability of computer code

Data collection

> Adult and neonatal ATAC-seq: ATAC fastq files were processed using the ENCODE ATAC-seq pipeline (v1.7.0), with parameters: atac.auto_detect_adapter = TRUE, atac.multimapping = 0, atac.pval_thresh = 0.01. Bigwig tracks were generated from the filtered BAM files using deepTools (v3.5.0).
>
> Adult, neonatal, and cell line CUT&RUN: Adapters and low-quality basecalls were trimmed using cutadapt (v2.8). Paired-end reads were aligned to mm10 using bowtie2 (v2.3.4.2). Duplicate reads were removed with picard (v2.18.20). Reads were filtered using SAMtools (v1.11). Peaks were called using MACS2 (v2.2.7.1). Bigwig tracks were generated using deepTools (v3.5.0).
>
> Adult RiboTag RNA-seq: Adapters and low-quality basecalls were trimmed using fastx_toolkit (v0.0.13). Single-end reads were aligned to mm10 using STAR (v2.7.6). Reads overlapping gene exons were counted using Subread featureCounts (v2.0.1).
>
> Neonatal bulk nuclear RNA-seq: Adapters and low-quality basecalls were trimmed using cutadapt (v2.8). Single-end reads were aligned to mm10 using STAR (v2.7.6). Technical duplicate reads (identical molecular identifier and read) were removed using nudup.py (v2.3.3). Reads overlapping each gene (including introns) were counted using Subread featureCounts (v2.0.1).
>
> Single-nucleus multiome-seq: Raw sequencing data were processed using the Cell Ranger ARC pipeline (v2.0.0) with the cellranger-arc mm10 reference. Default parameters were used to align reads, count unique fragments or transcripts, and filter high-quality nuclei.
>
> Single-nucleus RNA-seq: Raw sequencing data were processed using the Cell Ranger pipeline (v6.0.0) with the refdata-gex-mm10-2020-A reference. Default parameters were used to align reads, count unique transcripts, and filter high-quality nuclei.
>
> Flow cytometry: Sony SH800S Cell Sorter Software (v2.1).

| Data analysis | The following packages were used to analyze and visualize data: deepTools (v3.5.0), CUT&RUNTools scripts (https://bitbucket.org/qzhudfci/cutruntools/src/master/), MEME (v5.1.1), BEDTools (v2.29.0), clusterProfiler (v3.10.1), Gviz (v.1.34.1), DESeq2 (v1.30.1), DiffBind (v2.10.0), Seurat (v4.0.3), SingleCellExperiment (v1.12.0), ComplexHeatmap (v2.11.1), MetaNeighbor (v1.10.0), ChIPseeker (v1.18.0), eulerr (v6.1.1), ggplot2 (v3.3.3), pheatmap (v1.0.12), BETA (v1.0.7), seaborn (v0.11.0), pandas (v1.1.4), edgeR (v3.32.1), R stats (v4.0.3), chromVAR (v1.12.0), ArchR (v1.0.1), Signac (v1.3.0), regioneR (v1.22.0), Fiji/ImageJ (v2.0.0). Custom scripts can be found at https://github.com/gegenhu/estrogen_gene_reg. |
|---|---|

For manuscripts utilizing custom algorithms or software that are central to the research but not yet described in published literature, software must be made available to editors and reviewers. We strongly encourage code deposition in a community repository (e.g. GitHub). See the Nature Research guidelines for submitting code & software for further information.

# Data

Policy information about availability of data

All manuscripts must include a data availability statement. This statement should provide the following information, where applicable:
- Accession codes, unique identifiers, or web links for publicly available datasets
- A list of figures that have associated raw data
- A description of any restrictions on data availability

The data generated by this study are in GEO (GSE144718). Additional datasets were analyzed in this study: MCF7 ERa ChIP-seq (Franco et al. 2015, GSE59530), mouse liver 1 ERa ChIP-seq (Gertz et al. 2013, GSE49993), mouse liver 2 ERa ChIP-seq (Gordon et al. 2014, GSE52351), mouse uterus 1 ERa ChIP-seq (Hewitt et al. 2012, GSE36455), mouse uterus 2 ERa ChIP-seq (Gertz et al. 2013, GSE49993), mouse aorta ERa ChIP-seq (Gordon et al. 2014, GSE52351), mouse efferent ductules ERa ChIP-seq (Yao et al. 2017, Supplementary Info), mouse mammary gland ERa ChIP-seq (Palaniappan et al. 2019, GSE130032), mouse liver ATAC-seq (Cusanovich et al. 2018, GSE111586), BNST snRNA-seq (Welch et al. 2019, GSE126836), MPOA scRNA-seq (Moffitt et al. 2018, GSE113576), Allen Brain Institute Cell Type Database (Yao et al. 2020, https://portal.brain-map.org/atlases-and-data/rnaseq/mouse-whole-cortex-and-hippocampus-10x).

# Field-specific reporting

Please select the one below that is the best fit for your research. If you are not sure, read the appropriate sections before making your selection.

☒ Life sciences ☐ Behavioural & social sciences ☐ Ecological, evolutionary & environmental sciences

For a reference copy of the document with all sections, see nature.com/documents/nr-reporting-summary-flat.pdf

# Life sciences study design

All studies must disclose on these points even when the disclosure is negative.

| Sample size | Sample sizes were determined by the following factors: 1) number of replicates previously used for ATAC-seq, RNA-seq, CUT&RUN, snRNA-seq, and multiome in the literature (Stroud et al., 2020, Neuron; Allaway et al., 2021; Nature; Di Bella et al., 2021, Nature), 2) cost of sequencing and animal maintenance, and 3) expected variability between samples for each assay after collecting preliminary data. For neonatal single-cell experiments, the target number of cells per sample was determined by prior estimates of cell type abundance from the published adult BNST snRNA-seq dataset (Welch et al., 2019, Cell). |
|---|---|
| Data exclusions | No data were excluded. |
| Replication | The number of biological replicates per condition for each experiment is listed below. All attempts at replication were successful. |
| | MCF7 ERa CUT&RUN: n=2 |
| | Adult brain ERa CUT&RUN: n=2 |
| | Adult brain Nfix CUT&RUN: n=2 |
| | Adult gonadectomy+E2 ATAC-seq: n=3 |
| | Adult gonadally intact ATAC-seq: n=2 |
| | Adult RiboTag RNA-seq: n=4 |
| | Adult in situ hybridization: n=4 |
| | P14 Nfix immunofluorescent staining: n=6 |
| | Neonatal bulk nuclear RNA-seq: n=3 |
| | Neonatal ATAC-seq: n=3 |
| | Neonatal ERa CUT&RUN: n=2 |
| | |
| | For each sequencing experiment, brain tissue was pooled from 3-5 animals or 8-9 animals (adult RNA-seq only) per biological replicate to account for dissection variability. For the single-cell multiome experiments, brain tissue was pooled from 5 animals per condition prior to collection. For the P14 single-nucleus RNA-seq experiment, brain tissue was pooled from 3 animals per condition. |
| Randomization | For the adult treatment experiments, gonadectomized female and male mice were randomly assigned to vehicle and E2 groups. For the neonatal treatment experiments, females were randomly assigned to vehicle and E2 groups. For MCF-7 cell culture experiments, cells were randomly assigned to vehicle and E2 groups. |
| Blinding | Blinding was not performed for bioinformatics analysis, because knowledge of sample identity is required for designing statistical tests and visualizing data. Investigators were blinded to group allocation during data collection of bioinformatics experiments. For the ISH and IF experiments, the investigator was blinded during data collection and analysis. |

# Reporting for specific materials, systems and methods

We require information from authors about some types of materials, experimental systems and methods used in many studies. Here, indicate whether each material, system or method listed is relevant to your study. If you are not sure if a list item applies to your research, read the appropriate section before selecting a response.

## Materials & experimental systems

| n/a | Involved in the study |
|---|---|
| ☐ | ☒ Antibodies |
| ☐ | ☒ Eukaryotic cell lines |
| ☒ | ☐ Palaeontology and archaeology |
| ☐ | ☒ Animals and other organisms |
| ☒ | ☐ Human research participants |
| ☒ | ☐ Clinical data |
| ☒ | ☐ Dual use research of concern |

## Methods

| n/a | Involved in the study |
|---|---|
| ☐ | ☒ ChIP-seq |
| ☐ | ☒ Flow cytometry |
| ☒ | ☐ MRI-based neuroimaging |

## Antibodies

| | |
|---|---|
| Antibodies used | Human ERa CUT&RUN Ab #1: Santa Cruz sc-8002 (Lot 41718); Human ERa CUT&RUN Ab #2: Millipore Sigma 06-935 (Lot 2971020); Mouse ERa CUT&RUN Ab: Millipore Sigma 06-935 (Lot 2971020); Nfix CUT&RUN Ab: Abcam ab101341 (Lot GR3173994); Nfix IF Ab: Thermo Fisher PA5-30897 (Lot WA31716320); Guinea pig anti-rabbit IgG CUT&RUN Ab: Antibodies-Online ABIN101961 (Lot 42670); GFP IF Ab: Aves Labs GFP-1020 (Lot GFP697986); Anti-rabbit IgG-Cy3 Ab: Jackson Immuno 711-165-152; Anti-chicken IgG-488: Jackson Immuno 703-545-155. |
| Validation | All antibodies used for IF staining have been validated by the manufacturer for this purpose and have been used in prior publications (ERa IF: Wu et al. 2017; GFP IF: Mo et al. 2015; Nfix IF: Adam et al. 2020). The ERa antibody (Millipore Sigma 06-935) used for CUT&RUN was validated previously by IF staining in ERa knockout animals (Wu et al., 2017) and in this study through bioinformatic analysis, specifically 1) top enrichment of the ERE motif (Extended Data Fig. 1.1b, f), 2) similarity to prior ERa ChIP-seq data (Extended Data Fig. 1.1a-b, e-f), and 3) comparison between P0 brain Esr1+ and Esr1- cells (Fig. 3a). The Nfix antibody (Abcam ab101341) used for CUT&RUN was validated in this study in mHypoA cells and BNSTp by top enrichment of the NFI family motif (Extended Data Fig. 3.3i). |

## Eukaryotic cell lines

Policy information about cell lines

| | |
|---|---|
| Cell line source(s) | MCF-7 cells - ATCC; mHypoA cells - Cedarlane Laboratories |
| Authentication | The human MCF-7 cell line was validated by STR profiling and morphology. The mouse mHypoA cell line was validated by morphology. Both cell lines were used only for preliminary validation of primary antibodies for CUT&RUN. |
| Mycoplasma contamination | Cell lines were not tested for mycoplasma contamination. |
| Commonly misidentified lines (See ICLAC register) | No commonly misidentified lines were used. |

## Animals and other organisms

Policy information about studies involving animals; ARRIVE guidelines recommended for reporting animal research

| | |
|---|---|
| Laboratory animals | Esr1-Cre, Vgat-Cre, Rpl22-HA, Sun1-GFP, and C57Bl6/J wildtype mice were obtained from Jackson labs. Esr1-flox mice were received from Sohaib A. Khan.<br>Adult CUT&RUN experiments were performed on C57Bl6/J female and male mice at 8-12 weeks of age.<br>Adult RNA-seq and ISH experiments were performed on Esr1-Cre/Rpl22-HA female and male mice at 8-12 weeks of age.<br>Adult ATAC-seq experiments were performed on Esr1-Cre/Sun1-GFP female and male mice at 8-12 weeks of age.<br>Neonatal ATAC-seq, bulk RNA-seq (female only), IF staining, and multiome experiments were performed on Esr1-Cre/Sun1-GFP female and male mice at P4 and P14 (IF and multiome only).<br>Neonatal CUT&RUN was performed on Esr1-Cre/Sun1-GFP female mice at P0.<br>Neonatal snRNA-seq was performed on Vgat-Cre/Esr1-flox/Sun1-GFP female and male mice at P14. |
| Wild animals | None |
| Field-collected samples | None |
| Ethics oversight | All mouse experiments were performed under strict guidelines set forth by the CSHL Institutional Animal Care and Use Committee (IACUC). |

Note that full information on the approval of the study protocol must also be provided in the manuscript.

# ChIP-seq

## Data deposition

☒ Confirm that both raw and final processed data have been deposited in a public database such as GEO.

☒ Confirm that you have deposited or provided access to graph files (e.g. BED files) for the called peaks.

| | |
|---|---|
| Data access links<br>*May remain private before publication.* | CUT&RUN data are available in GEO (GSE144718). Reviewer access token is shapeowsfdwznot |
| Files in database submission | narrowPeak, bigwig, and raw fastq files are available for each sample. |
| Genome browser session<br>(e.g. UCSC) | Tracks can be visualized in a genome browser with the provided bigwig files. |

## Methodology

| | |
|---|---|
| Replicates | MCF-7 ERa CUT&RUN, n=2 per treatment and per antibody<br>Adult brain IgG CUT&RUN, n=1 per treatment<br>Adult brain ERa CUT&RUN, n=2 per treatment and sex<br>mHypoA Nfix CUT&RUN, n=1<br>Adult BNSTp Nfix CUT&RUN, n=2 per treatment and sex<br>Neonatal brain ERa CUT&RUN, n=2 per treatment<br>Neonatal brain IgG CUT&RUN, n=1 per treatment |
| Sequencing depth | Read numbers indicate 1) total number of PE sequencing reads and 2) number of PE reads after BAM processing.<br><br>MCF-7 ERa Ab #1 Veh 1: 17128542 total; 2629779 processed<br>MCF-7 ERa Ab #1 E2 1: 18385942 total; 4593339 processed<br>MCF-7 ERa Ab #1 Veh 2: 14951660 total; 1915579 processed<br>MCF-7 ERa Ab #1 E2 2: 26172689 total; 7248734 processed<br>MCF-7 ERa Ab #2 Veh 1: 40779817 total; 11701045 processed<br>MCF-7 ERa Ab #2 E2 1: 42427425 total; 12165932 processed<br>MCF-7 ERa Ab #2 Veh 2: 39253805 total; 7674110 processed<br>MCF-7 ERa Ab #2 E2 2: 37248955 total; 9848088 processed<br>Adult brain IgG Veh 1: 17535876 total; 2873358 processed<br>Adult brain IgG E2 1: 16698729 total; 2521465 processed<br>Adult brain ERa male Veh 1: 20533002 total; 3293050 processed<br>Adult brain ERa male E2 1: 15185294 total; 2638242 processed<br>Adult brain ERa female Veh 1: 32037063 total; 5923281 processed<br>Adult brain ERa female E2 1: 31370254 total; 6464871 processed<br>Adult brain ERa male Veh 2: 23405701 total; 4078205 processed<br>Adult brain ERa male E2 2: 26406065 total; 4251286 processed<br>Adult brain ERa female Veh 2: 43085293 total; 7658066 processed<br>Adult brain ERa female E2 2: 41292689 total; 6021754 processed<br>P0 brain IgG Veh 1: 31842043 total; 9424613 processed<br>P0 brain IgG E2 1: 28725177 total; 8714349 processed<br>P0 brain ERa female Esr1+ Veh 1: 40857568 total; 8107696 processed<br>P0 brain ERa female Esr1+ E2 1: 39814592 total; 7438993 processed<br>P0 brain ERa female Esr1+ Veh 2: 39430462 total; 8574318 processed<br>P0 brain ERa female Esr1+ E2 2: 36778406 total; 5667825 processed<br>P0 brain ERa female Esr1- E2 1: 29611541 total; 8061354 processed<br>mHypoA Nfix: 10797210 total; 2778488 processed<br>Adult BNSTp Nfix male Veh 1: 23589255 total; 4300252 processed<br>Adult BNSTp Nfix male E2 1: 25583254 total; 5000254 processed<br>Adult BNSTp Nfix female Veh 1: 17477908 total; 4489675 processed<br>Adult BNSTp Nfix female E2 1: 19349717 total; 3705371 processed<br>Adult BNSTp Nfix male Veh 2: 22677910 total; 3543642 processed<br>Adult BNSTp Nfix male E2 2: 23858110 total; 3026763 processed<br>Adult BNSTp Nfix female Veh 2: 17843258 total; 4988121 processed<br>Adult BNSTp Nfix female E2 2: 21971104 total; 3185951 processed |
| Antibodies | Human ERa CUT&RUN Ab #1: Santa Cruz sc-8002 (Lot 41718); Human ERa CUT&RUN Ab #2: Millipore Sigma 06-935 (Lot 2971020); Mouse ERa CUT&RUN Ab: Millipore Sigma 06-935 (Lot 2971020); Nfix CUT&RUN Ab: Abcam ab101341 (Lot GR3173994); Guinea pig anti-rabbit IgG CUT&RUN Ab: Antibodies-Online ABIN101961 (Lot 42670). |
| Peak calling parameters | Peaks were called using MACS2 callpeak. Differential peaks were called using DiffBind, with specific parameters indicated in the Methods section. |
| Data quality | Only differential peaks that reached statistical significance were used for downstream analysis. For Nfix CUT&RUN, only peaks that were present across both sexes and treatment conditions were considered to be consensus peaks. For brain CUT&RUN experiments, target peaks overlapping peaks called in the IgG control were filtered out. Total peak numbers for each sample: MCF-7 ERa Ab #1 Veh 1: 610, MCF-7 ERa Ab #1 E2 1: 9506, MCF-7 ERa Ab #1 Veh 2: 642, MCF-7 ERa Ab #1 E2 2: 16153, MCF-7 ERa Ab #2 Veh 1: 11748, |

MCF-7 ERa Ab #2 E2 1: 9331, MCF-7 ERa Ab #2 Veh 2: 4417, MCF-7 ERa Ab #2 E2 2: 32817, Adult brain ERa male Veh 1: 9719, Adult brain ERa male E2 1: 10143, Adult brain ERa female Veh 1: 5520, Adult brain ERa female E2 1: 4965, Adult brain ERa male Veh 2: 15900, Adult brain ERa male E2 2: 13533, Adult brain ERa female Veh 2: 2395, Adult brain ERa female E2 2: 3754, P0 brain ERa female Esr1+ Veh 1: 4, P0 brain ERa female Esr1+ E2 1: 10924, P0 brain ERa female Esr1+ Veh 2: 7, P0 brain ERa female Esr1+ E2 2: 9098, mHypoA Nfix: 30825, Adult BNSTp Nfix male Veh 1: 24801, Adult BNSTp Nfix male E2 1: 39528, Adult BNSTp Nfix female Veh 1: 49151, Adult BNSTp Nfix female E2 1: 63111, Adult BNSTp Nfix male Veh 2: 69995, Adult BNSTp Nfix male E2 2: 21358, Adult BNSTp Nfix female Veh 2: 62390, Adult BNSTp Nfix female E2 2: 63019.

| Software | The following packages were used to analyze and visualize CUT&RUN data: deepTools (v3.5.0), CUT&RUNTools scripts (https://bitbucket.org/qzhudfci/cutruntools/src/master/), MEME (v5.1.1), BEDTools (v2.29.0), clusterProfiler (v3.10.1), Gviz (v.1.34.1), DiffBind (v2.10.0), edgeR (v3.32.1), ChIPseeker (v1.18.0), eulerr (v6.1.1), ggplot2 (v3.3.3), BETA (v1.0.7), seaborn (v0.11.0), pandas (v1.1.4). |
|---|---|

# Flow Cytometry

## Plots

Confirm that:

☒ The axis labels state the marker and fluorochrome used (e.g. CD4-FITC).

☒ The axis scales are clearly visible. Include numbers along axes only for bottom left plot of group (a 'group' is an analysis of identical markers).

☒ All plots are contour plots with outliers or pseudocolor plots.

☒ A numerical value for number of cells or percentage (with statistics) is provided.

## Methodology

| Sample preparation | Esr1Cre/+; Sun1-GFPlx/+ mice (4 pooled per condition) were deeply anesthetized with ketamine/dexmedetomidine. 500-µm sections spanning the BNSTp were collected in a mouse brain matrix (Kent Scientific) on ice. The BNSTp was microdissected and collected in 1 ml of cold supplemented homogenization buffer (250 mM sucrose, 25 mM KCl, 5 mM MgCl2, 120 mM tricine-KOH, pH 7.8), containing 1 mM DTT, 0.15 mM spermine, 0.5 mM spermidine, and 1X EDTA-free PIC (Sigma Aldrich 11873580001). The tissue was dounce-homogenized 15x in a 1 ml glass tissue grinder (Wheaton) with a loose pestle. 0.3% IGEPAL CA-630 was added, and the suspension was homogenized 5x with a tight pestle. The homogenate was filtered through a 40-µm strainer then centrifuged at 500 x g for 15 min at 4oC. The pellet was resuspended in 0.5 ml homogenization buffer containing 1 mM DTT, 0.15 mM spermine, 0.5 mM spermidine, and 1X EDTA-free PIC. 30,000 GFP+ nuclei were collected into cold ATAC-RSB (10 mM Tris-HCl pH 7.5, 10 mM NaCl, 3 mM MgCl2) using the Sony SH800S Cell Sorter (purity mode) with a 100-µm sorting chip. |
|---|---|
| Instrument | Sony SH800S Cell Sorter |
| Software | Sony SH800S Cell Sorter Software |
| Cell population abundance | Esr1-Cre/Sun1-GFP+ population comprised ~10-20% of singlet nuclei. Vgat-Cre/Sun1-GFP+ population comprised ~40-60% of singlet nuclei. |
| Gating strategy | 1) Nuclei were first gated on the BSC-A x FSC-A plot.<br>2) Singlet nuclei were gated on the linear axis of the FSC-H x FSC-A plot.<br>3) GFP+ singlet nuclei were gated on the FSC-A x GFP:EGFP-A plot.<br>Post-sort purity was assessed on a fluorescence microscope. |

☒ Tick this box to confirm that a figure exemplifying the gating strategy is provided in the Supplementary Information.

