## [Peer Review File · Nature]

Manuscript Title: Epigenomic organization and activation of brain sex differences

Reviewer Comments & Author Rebuttals

Reviewer Reports on the Initial Version:

Referee #1 (Remarks to the Author):

The authors of this manuscript provide comprehensive data on the genomic regulation of transcription by estrogen receptor alpha (ER α) in a brain structure that controls sexually dimorphic behaviors, the posterior bed nucleus of the stria terminalis (BNSTp). They identify genomic sites of ER α binding by 'CUT&RUN,' and characterize transcriptional programs in ER α + BNSTp neurons using TRAP, in tissue from both hormone-treated and vehicle-treated controls. Their data suggest that hormone-induced binding of ER α causes local chromatin opening, and that chromatin is open in both genes that are positively regulated by ER α , which is constitutively bound, and genes that are negatively regulated by hormone, where the receptor is not bound. Using analysis of a previously published single-nucleus RNAseq (snRNAseq) dataset from a different laboratory, they show that sex differences in gene expression in BNSTp occur almost exclusively in neurons that express ER α . They also show that changes in gene expression caused by neonatal estrogen administration in females encompass genes expressed in adult male BNSTp cell types, based on analysis of the snRNAseq dataset. Based on other published scRNAseq studies of GABAergic neurons, they impute an identity to BNSTp GABAergic cell types as similar to that of a 'neuroglia form' Lamp5+ cortico-hippocampal GABAergic type derived from the CGE. Finally, they identify the transcription factor Nfix, a member of the NFI family of transcription factors, as involved in ER α -mediated transcription in a subset of BNSTp cells, most likely via binding to open chromatin boundary sites.

The data provided in this paper provide a comprehensive look at the genomic landscape of ER α -mediated regulation in neural cells. This will provide valuable information for follow-up studies of how estrogen influences gene expression in the brain, and in that respect they offer a useful resource to the community. They also provide some new insights into the relationship between ER α -mediated gene regulation and male biases in gene expression, at least in BNSTp. While it is interesting that sex differences in gene expression in BNSTp occur mainly in ER α + cells, this finding is based on reanalysis of others' snRNAseq data, rather than on the primary dataset in this paper. Overall, the majority of the conclusions in this paper are consistent with, but extend to a more granular level of detail, previous studies; one may say there are relatively few surprises.

From the conceptual standpoint, the two major questions here are 1) how does transient neonatal exposure to estrogen induce lasting sex differences in the adult brain; and 2) how does estrogen activate neuron-specific vs. non-neuronal target genes in brain vs. peripheral tissue. On these two points, the data in the paper do not push the needle of our understanding very much further. The answer to the first question is primarily explained by the effects of estrogen to promote selective cell survival in the male brain. The alternative view, that estrogen binding in neonates somehow

establishes a stable pattern of open chromatin organization that persists into adulthood, is perhaps suggested by the experiment of Fig. 3d. However, as detailed below (specific comments), this experiment has serious flaws; moreover, it does not exclude that the persistence of male-specific gene expression into adulthood is simply due to the persistence of ER α expression into adulthood, rather than to stable open chromatin domains. On the second point, the idea that differences in targets of ER α transcriptional regulation between different tissues reflects differential chromatin accessibility in those targets has been around for decades. While the chromatin accessibility data presented here are consistent with that view, they do not distinguish cause from effect. In general, many of the conclusions here are based on correlations, and it is difficult to disentangle direct from indirect effects within this correlative network without some kind of formal, quantitative multiple regression analysis.

Specific Points:

1. Why was CUT&RUN used instead of ChIP-seq for the brain tissue? Explain in main text.
2. Different antibodies were used for ChIP-seq/CUT&RUN in breast cancer cells, vs. in brain tissue. Why? What controls were done to ensure that both antibodies are equally specific for ER α , in both the ChIP-seq and CUT&RUN procedures?
3. “Estrogen-induced gene regulation in the BNSTp was not shared across all ER α + brain regions, demonstrating regional specificity...” No evidence to support this statement is presented, nor any literature cited.
4. Expt of Fig. 3d: It would be useful to compare genes that are down-regulated in neo-natal males treated with an ER α antagonist, rather than just genes that are up-regulated in neonatal females treated with an ER α agonist.
5. Why are the data in Fig. 3d “consistent with the hypothesis that early life estrogen signaling organizes a lasting male-typical transcriptional program”? If the authors want to show this, they should perform the analysis of Fig. 3d on cell types from adult females exposed to neonatal estrogen, not on data from adult males drawn from a previously published study. Why are some of the neonatally induced transcriptional changes maintained and others lost in adulthood?
6. Assignment of BNSTp ER α cell types to cortical/hippocampal IN classes: I’m not sure if this is valid, since it uses two different datasets (both produced by other labs) derived using potentially different methods of scRNAseq.
7. “32% of ER α peaks containing the NFI motif” – it would be useful to see the numbers for a different class of NHR peaks. Maybe NFI motifs are present in 32% of all NHR peaks.
8. What percentage of BNSTp ER α + neurons co-express Nfix? The authors refer to a “subset,” but do not state the size of that subset.
9. “ER α and Nfix co-regulate a distinct module of estrogen-responsive and male-biased genes in the BNSTp.” This may be semantics, but to my mind “co-regulation” implies some sort of cooperativity. Here it seems more likely that ER α opens chromatin, and that Nfix is bound at open-chromatin boundaries. It may look like a specific “co-regulation” just because Nfix expression is enriched in BNSTp.
10. Evidence for CGE derivation of ER α /Nfix-expressing inhibitory neurons is just circumstantial and based on shared gene expression, which does not predict shared developmental ancestry. Fate-mapping studies are required to establish this point.

Referee #2 (Remarks to the Author):

As the authors point out, little is known about the direct gene targets of estradiol in the brain, despite the importance of this hormone in the establishment of sex differences and the control of physiology and behavior. The authors use a suite of the most modern molecular techniques to address this and related questions. Without a doubt, this is the most in-depth analysis of estrogen-regulated gene expression in the brain to date, with many interesting findings along the way. I am amazed that the authors were able to accomplish what they did, using highly heterogeneous and hard-to-work-with brain tissue.

A number of the findings in the first half of the paper are confirmatory. For example, the authors find that the preponderance of estrogen-regulated genes have an ERE, and that at the large majority of direct ER α targets, expression is up- (rather than down-) regulated by estradiol. These conclusions will not surprise anyone, and have been reached over the years by others using older techniques. Other findings (mostly in the second half of the paper) are truly new. For example, the provision of a comprehensive list of ER binding sites in the brain is novel and provides molecular candidates that may underlie effects of estradiol both normally and in disease, and the large number of genes epigenetically regulated by ER binding (based on chromatin accessibility) is surprising and, as far as I know, new.

There are three general areas of concern, however, that dampened my enthusiasm.

1. The paper is difficult to read.

For someone willing to do the work, there is much to be mined from this paper. However, even for someone intensely interested in the topic, getting through the paper is hard work. It is dense both because of the amount and type of data being presented, and also because the writing doesn't do enough to help the reader along.

2. The emphasis on "neurodevelopment" seems misplaced (or unconvincing)

The authors use adult gonadectomies followed by acute estradiol treatment, to conclude that estradiol induces a "robust neurodevelopmental gene regulatory program." The emphasis on "neurodevelopmental" in several places in the paper is puzzling, since the findings are based on experiments performed in adults, and many of the genes identified perform roles such as the regulation of membrane potential and synapse organization, which are important throughout the life of a neuron.

The only experiment performed during development treated newborn females with estradiol or vehicle at birth and examined gene expression on P4. The authors find that 33% of the estradiol-induced, ER target genes are also among those with male-biased expression in adults and conclude that this is "consistent with the hypothesis that early-life estrogen signaling organizes a lasting, male-typical transcriptional program" "...that likely specify and maintain sex differences in neuronal firing and synaptic patterning." I'm not convinced that the authors have shown this. Wouldn't an equally likely explanation be simply that some genes are regulated by estradiol both in newborns and also later in life (i.e., nothing is "set up" by the early estradiol)? It also is surprising that there were not newborn males in this experiment, to support conclusions about the start of a male-typical

transcriptional program.

3. Confusion about Sexual Differentiation

One major thrust of this paper regards sexual differentiation and, in fact, the Introduction starts with a description of sexual differentiation during perinatal life. However, the findings regarding sexual differentiation seem somewhat contradictory. The authors present data supporting the conclusion that, “both sexes possess similar capacity to mount a transcriptional response to estrogen” and elsewhere, “... ER α binding ... also lacks sex-specific recruitment.” At the same time, however, the paper goes on to identify what the authors call a “male-biased gene expression signature” (Abstract and elsewhere). Either there is, or is not, sexual differentiation of gene expression, yet the manuscript seems to have it both ways.

I suspect that some of the confusion comes from the fact that animals were gonadectomized and given acute estradiol in the studies performed here, but were gonadally intact in the single-nucleus RNA-seq dataset by Welch et al (2019) that the authors re-analyze and compare to some of their data. The reader is not told about this difference, however. I think the authors try to address this in the Discussion, but too late to avoid confusion in the mind of the reader. I wonder also if the authors are missing an opportunity to address the potentially non-physiological nature of the gonadectomy-acute-estradiol model (a paradigm that allows one to synchronize estradiol exposure but is not a situation a mouse would ever encounter). In other words, to the extent that there is overlap between the male-biased genes (Welch et al.) and the ER binding sites (current study), it may support the physiological relevance of the authors’ findings.

Related to points 2 & 3: Given how the authors start the paper (“how transient exposure to estrogen organizes lasting sex differences remains a central question,”) would it have been more fruitful to look at ER binding sites in Male, Female, and Female + E2 newborns (around the time of the testosterone surge)? Alternatively, the authors could change their Introduction so it is less focused on sexual differentiation, and more on what estradiol is doing in adults, since the bulk of the data is in adults.

Minor comments/corrections

Male/Female labels seem to be missing on the quantification in Figure 2b.

The Figure legends often don’t give the reader enough information to know what they are looking at.

In what sense is postnatal day 4 “the peak of BNSTp sexual differentiation?”

Is it correct that “all BNST neurons are GABAergic” (second paragraph of Discussion), or just that many of them are?

Referee #3 (Remarks to the Author):

This manuscript presents an in depth-analysis of transcriptional regulation by estrogen receptor - alpha (Esr1) in a distinct region of the mouse brain known to be functionally important to sex differences in social behaviors. Much of what they discover regarding ER-alpha interaction with the genome is unique to the brain and thus, not surprisingly, are mostly relevant to parameters associated with neural functioning. The work appears to be expertly done and is among the most exciting and progressive on the important topic of biological origins of sex differences in the brain. The weakness is in presentation in terms of making the significance of the findings accessible to the reader, even the experts in the field. Specific comments below.

1) The strengths of the report include the sophisticated approaches providing an unprecedented level of information regarding hormonal modulation of transcription in males versus females in a brain region known to be “programmed” or “organized” by steroids early in life. However there is some muddling of the message that at times makes it unclear what are the take aways. For instance, the authors state in the Abstract that “estrogen induces a robust neurodevelopmental gene regulatory program in both sexes” – but they do not discuss that the fundamental principle of sexual differentiation of the rodent brain is that females are not naturally exposed to estrogens. It has been known for 50 years that females can be masculinized by treating them with exogenous estradiol as neonates. Are the authors trying to say that the same transcriptional profile is induced in females as occurs naturally in males during development and hence explaining how sexual differentiation occurs? Or, are they saying something quite different, that transcriptional regulation does NOT explain sexual differentiation of behavior. It is difficult to tell.

2) Related to this, I am confused by the header “Estradiol induces a neurodevelopmental gene regulatory program in the adult BNSTp” – it is not clear at what age animals were treated, did both sexes receive the same treatment and why this is a neurodevelopmental program? Synaptic organization does not imply development given what we now know about synaptic plasticity. The authors appear to conclude that there is no impact of developmental programming by estradiol on adult estradiol responding. This may be true at the transcriptional level but it certainly is not true at the behavioral level. This section also includes some over interpretations as are present throughout, such as the idea that because the glucocorticoid receptor is expressed at higher levels in females this might explain sex differences in stress responding. Other references to relevance of the current report to disease are also over extensions.

3) The next section speaks to male specific transcriptional changes which involve ER-alpha binding, so at first glance appears to contradict the above. It was not clear to this reviewer if the authors are arguing these DEGs are unrelated to the hormone, estradiol, but are a function of its receptor, ER-alpha? They go on to hypothesize that certain genes more highly expressed in males are the result of a “persistent transcriptional signature of neonatal estradiol”, yet they emphasize that both male and female brains respond to estradiol the same, so which is it? The point being, if there is a clear conclusion here, it is not easy for the reader to grasp.

4) The last section concludes that a subset of ER-alpha expressing inhibitory neurons “define sexual dimorphism in the BNST” but this is a bridge too far beyond the data. There is no evidence here that the sex difference discovered plays any role in either the morphology of the BNST or the behaviors it regulates. It is a possibility, but right now is speculation. Steps to consider for making the causal

connection are morphometric phenotyping of the *Esr1*-*Nfix* expressing neurons in term of dendritic complexity, synaptic density and afferent and efferent inputs. Ultimately manipulating those cells specifically via optogenetic or chemical means to modulate sex differences in behavior, would fully close the loop.

Minor points

- 1) It is important to get the nomenclature correct when referring to steroid hormones. Here the authors tend to use the term “estrogen” and “estradiol” interchangeably or, at times, incorrectly. Estrogens are a class of compounds that include estradiol, estriol and estrone. Thus one cannot treat with “estrogen” only with one of the 3 steroids mentioned above. The lead in sentence of the abstract is incorrect as there is no “endogenous form” of estrogen. Wording should be changed throughout to reflect that estrogen is a category and estradiol is a hormone.
- 2) Related to the above, the authors treat mice with estradiol-benzoate, a modified version of 17-beta-estradiol that prolongs it's action, however the hormonal effects are that of estradiol with the benzoate moiety serving no signaling role. The authors use EB to refer to estradiol benzoate, which is customary for neuroendocrinologist, but it may be better to replace with E2, the chemical name of 17-beta-estradiol, for a more general audience.
- 3) Lastly, in the Abstract that authors state that estradiol helps establish sex differences in the vertebrate brain but this is an over generalization, particularly in that there is no definitive evidence that estrogens are important to primate brain sexual differentiation, including humans.

Referee #4 (Remarks to the Author):

Summary: In this work, the authors seek to address the interesting question of how sex-dependent gene expression programs are established in hypothalamic circuitry via the estrogen receptor alpha ($ER\alpha$) transcription factor. The authors combine genomic methods including CUT&RUN and ATAC-seq to map binding of $ER\alpha$ and loci whose accessibility is responsive to estradiol exposure in vivo. The authors also perform a re-analysis of previously published single cell RNA-seq data to show that $ER\alpha$ binding sites are enriched at genes whose expression is male-biased, suggesting that $ER\alpha$ could regulate the establishment of male-biased gene expression. The authors identify *Nfix* as a transcription factor that potentially coordinates with $ER\alpha$ in a subset of inhibitory neurons. Overall, the work presented is technically well done and will provide an important resource for the field of sex-biased differences in neuronal gene expression. However, there are a number of points that should be addressed before publication. Most notably, the authors should attempt to test their claims using direct perturbations of $ER\alpha$ and *Nfix* function as outlined below.

Major Points: The authors use bioinformatic approaches to arrive at a potential model of how *Nfix* and $ER\alpha$ might collaborate to specify male-specific gene expression. A main criticism of the paper, however, is that they provide no experimental perturbation to test this model. For example, can the authors genetically remove *Nfix* or $ER\alpha$ during the critical period (for example by injecting Cre-expressing viruses into conditional KO animals) and show that loss of these factors has a specific effect on male-biased genes in brain regions such as the BNSTp?

While this may be a challenging experiment, it would greatly improve the paper if the authors could attempt to map endogenous binding of ER α in males vs. females during the perinatal period. It may be that the authors are missing interesting key differences in ER α -dependent gene regulation by profiling with a strong estradiol stimulus at adult ages. This would also help to clarify the finding that despite ER α 's capacity to bind similar sets of loci in estradiol-stimulated adult females and males, these binding sites are enriched within genes with male-biased expression.

Figure 1 and Extended Data

1. The experimental design of ER α binding should be better explained in the text for a general audience. For example, given that the authors are interested in factors that establish sex-specific differences, it is surprising they chose to perform the experiments in the adult rather than during the perinatal critical period. The authors should clearly state the age at which these experiments are conducted and the rationale. In addition, the choice to use gonadectomized females and males needs to be explicitly clarified for a non-specialist audience. Do these animals develop normal hypothalamic circuitry (i.e. connectivity and cell number/distribution) such that the regions the authors are assaying in these mice are comparable to mice that have not undergone such treatment?
2. How do the levels of estradiol introduced subcutaneously in the treatment paradigm compare to physiological levels of estrogen? Is there a way to examine ER α binding in response to physiological levels of estrogen? For example, can the authors profile ER α binding in female mice upon entry to proestrus when circulating levels of estrogen are high? Alternatively, can the authors titrate the levels of estradiol to which mice are exposed to more closely mimic physiological levels?
3. The observation that ER α binds similarly in males and females (Extended Data Fig. 1.2) should be presented in Figure 1 as it is a major finding given how the authors explain the goals of their study in the introduction.
4. In Extended Data 1.2D, how do the authors explain a lack of ER α binding motifs at the EB-downregulated peaks? Are these bona fide ER α peaks? What is happening to evict ER α from these sites? Without an ERE motif, it seems possible that these peaks might be spurious or a reflection of background chromatin accessibility at these sites. Accordingly, I am not sure that these peaks represent "coordination of looping at chromatin boundaries."
5. In Extended data 1.3C, the authors should also compare ERE strength for the ESR2 motif, as this is more highly enriched than the ESR1 motif in their data.
6. I would like to see more characterization of the ER α antibody used in this study. Does the antibody recognize ER beta? Given how central the CUT&RUN results are to this study, the authors would also do well to perform a validation experiment in KO control animals rather than relying solely on IgG controls.

Figure 2 and Extended Data

1. Can the authors provide evidence that *Esr1Cre/+; Rpl22HA/+* and *Esr1Cre/+;Sun1-GFP/+* are marking the same sets of cells in males and females? This is important for interpreting sex differences in gene expression presented in this figure.
2. In Figure 2A, why is padj used for only one class?
3. In Extended Data 2.2B, the authors should provide quantification of the signal to more readily indicate which transcripts show differential regulation in regions other than BNSTp.

Figure 3 and Extended Data

1. In Figure 3B, the authors should clarify whether this clustering is performed in female or male mice or in both clusters. In this respect, it would be beneficial to include a UMAP plot that indicates the sex of the clustered samples.
2. The authors should also tone down their claim that “Across the 40 snRNA-seq clusters, nearly all differentially expressed genes (DEGs) were detected in ER α -expressing clusters (Fig. 3a), revealing ER α expression is predictive of sex differences in BNST gene expression.” Indeed the authors themselves point out that “other sex hormone receptors, including AR, ER β , and progesterone receptor (PR), were detected in BNSTp ER α + neurons with varying degrees of cluster specificity, and may also contribute to sex differences in gene expression (Extended Data Fig. 3.1E)”. If the authors were to plot the degree of sex-bias in gene expression in AR+, Er β +, and progesterone receptor (PR+) clusters, would it look different from Esr1+ expressing clusters?
3. In Extended data 3.1B, the panels are reversed in the figure legend. The authors should also clarify what the different columns in the expression heatmap panel represent.
4. For the neonatal EB exposure experiment in Extended data 3.2, how long does the male-gene bias persist after early life exposure to estradiol in females? It is unclear whether the bias in expression truly persists into adulthood given that samples are collected four days after exposure. The authors should perform the neonatal EB exposure and perform RNA-seq at an adult timepoint.
5. What is the rationale for performing the neonatal exposure in female mice and not male mice? Again, these experimental design choices need to be made very clear to a non-specialist audience. The authors state that 33% of neonatal EB-induced genes retain male-biased expression in adult. Reciprocally, the authors should analyze what percentage of male-biased genes are captured in the neonatal EB-induced gene list as this might give a sense of the relative importance of EB signaling in overall specification of a male-biased gene program.

Figure 4 and Extended Data

1. The choice to focus on inhibitory neuron classes for analysis of the BNSTp is not entirely clear to a non-specialist. Please provide further rationale in the text.
2. In Extended data 4.1D, what is the enrichment of Nfix binding sites at female-biased genes? While the St18 cluster does have a larger number of nuclei detected in males compared to females (Extended data 4.1A), there were a similar number of female- and male-biased genes detected in this cluster (Figure 3A).
3. Nfix is only expressed in a subset of ER α cells as suggested in Figure 4D. The overlap should be quantified.
4. The authors identify ER α sites are both pre-bound by Nfix and also sites where ER α recruits Nfix. What is the significance of these classes? How do these classes of sites differ at a) male-biased genes b) BNSTp EB-induced genes (defined in Figure 2) and c) neonatal EB-induced genes?
5. The authors’ analysis and discussion of the role of Nfix in CGE-specification seems tangential in this study and distracts from the main interest in sex specific gene expression programs (Fig. 4g, Extended Data Fig. 4.1).
6. Given that Nfix expression is expressed in only a subset of ER α + cells and binds at only 25% of ER α binding sites, how are male-biased genes being specified in additional ER α + clusters? While interesting, the reason to focus on Nfix in this study is not entirely clear. It would be interesting for the authors to attempt to identify the cooperating TF for ER alpha for all male biased clusters and if this is not possible or clear, to state why this might be the case.

7. The authors should tone down their claim that “Together, these results demonstrate that male-typical behaviors are largely regulated by a population of CGE-derived ER α /Nfix+ inhibitory neurons spanning the BNSTpr and SDN-POA” as they have neither perturbed Nfix function nor examined its impact on male-typical behavior in this study.

Author Rebuttals to Initial Comments:

We thank the Reviewers for their constructive comments, which we have carefully addressed. In our revision, we focused on a major conceptual point noted by all four Reviewers: Does early life estradiol signaling exert lasting effects on neuronal gene expression into adulthood? Indeed, this question is what motivated our initial characterization of ER α genomic binding in the brain. We are delighted to report our new findings, which we believe provide a comprehensive answer to this long-standing question. With the inclusion of extensive new data, the manuscript has become quite dense, therefore we have opted to expand to five figures to improve readability. Below, we summarize the major additions for the revision, and then provide a point-by-point response to each comment.

Additional experiments now included in the manuscript:

- 1) **ATAC-seq in P4 BNST ER α + neurons from males and females treated with vehicle on P0, as well as from females treated with estradiol on P0.** These findings complement our prior P4 RNA-seq and show that endogenous male-biased chromatin accessibility is largely recapitulated by P0 estradiol treatment of females, demonstrating the physiological relevance of our treatment paradigm.
- 2) **CUT&RUN for ER α in P0 mice.** These data show that ER α binding at birth is concordant with endogenous male-biased chromatin accessibility at P4. Together with the P4 ATAC-seq data, we demonstrate that neonatal ER α genomic recruitment directs sustained chromatin opening in males for at least four days.
- 3) **Multiome single-nucleus RNA-seq/ATAC-seq of BNST ER α + neurons at P4 and P14.** This powerful approach permits association of chromatin accessibility and transcript abundance within individual cells to identify sex-biased enhancers for differentially-expressed genes. We map our male-biased chromatin loci to sex-biased genes within individual neuron types, and discover striking heterogeneity in the chromatin response to the neonatal surge. A subset of neonatal-estradiol-opened targets show persistent male-biased expression out to P14, revealing the longevity of the genomic response to this transient stimulus at birth. Integration of these findings with prior Nfix CUT&RUN data demonstrates that Nfix binding sites are most accessible within the i1:Nfix (formerly BNSTpr_St18) cluster, and that Nfix binds its own locus exclusively in this cluster, implicating it as a key regulator of neuronal identity in this male-biased cell type.
- 4) **Genetic deletion of Esr1 (ER α) in BNSTp.** snRNA-seq of BNSTp inhibitory neurons from P14 *Vgat^{Cre};Esr1^{lox/lox};Sun1-GFP^{lx}* mutant male animals and *Vgat^{Cre};Esr1^{+/+};Sun1-GFP^{lx}* control male and female animals demonstrates that loss of ER α in these specific neurons both feminizes the sustained male-typical gene expression program and the abundance of the two male-biased neuron types (i1:Nfix and i3:Esr2).
- 5) **ATAC-seq in BNSTp ER α neurons from adult gonadally-intact males and females.** These results identify sex-differential loci that associate with sex-differential genes previously described in the adult snRNA-seq data and resolve a question raised by all 4 Reviewers: are adult male-biased genes a signature of neonatal estradiol? While we find that a significant fraction of neonatal male-biased genes are also male-biased in adults, the accessible loci associated with these genes are unexpectedly largely distinct; early life male-biased loci are opened by estradiol while adult male-biased loci for the same genes are opened by testosterone, indicating sequential activation by these two hormones at different life stages.

Referee #1 (Remarks to the Author):

From the conceptual standpoint, the two major questions here are 1) how does transient neonatal exposure to estrogen induce lasting sex differences in the adult brain; and 2) how does estrogen activate neuron-specific vs. non-neuronal target genes in brain vs. peripheral tissue. On these two points, the data in the paper do not push the needle of our understanding very much further. The answer to the first question is primarily explained by the effects of estrogen to promote selective cell survival in the male brain. The alternative view, that estrogen binding in neonates somehow establishes a stable pattern of open chromatin organization that persists into adulthood, is perhaps suggested by the experiment of Fig. 3d. However, as detailed below (specific comments), this experiment has serious flaws; moreover, it does not exclude that the persistence of male-specific gene expression into adulthood is simply due to the persistence of ER α expression into adulthood, rather than to stable open chromatin domains. On the second point, the idea that differences in targets of ER α transcriptional regulation between different tissues reflects differential chromatin accessibility in those targets has been around for decades. While the chromatin accessibility data presented here are consistent with that view, they do not distinguish cause from effect. In general, many of the conclusions here are based on correlations, and it is difficult to disentangle direct from indirect effects within this correlative network without some kind of formal, quantitative multiple regression analysis.

Specific Points:

1. Why was CUT&RUN used instead of ChIP-seq for the brain tissue? Explain in main text.

CUT&RUN was used because it provides higher signal:noise than ChIP-seq, which enables low-input profiling of TF binding. The advantages of CUT&RUN over ChIP-seq are extensively documented in the literature¹⁻³, and CUT&RUN has recently been used to profile transcription factors (TFs) in brain^{4,5}. Therefore we feel the rationale for using CUT&RUN does not need to be re-stated in the main text.

2. Different antibodies were used for ChIP-seq/CUT&RUN in breast cancer cells, vs. in brain tissue. Why? What controls were done to ensure that both antibodies are equally specific for ER α , in both the ChIP-seq and CUT&RUN procedures?

The MCF-7 cell culture experiment was the first ER α CUT&RUN experiment performed in our lab; therefore we used a mouse monoclonal antibody (sc-8002) recommended by Santa Cruz Biotechnology as the replacement for their discontinued rabbit polyclonal sc-542 antibody, which was the gold standard for ER α ChIP-seq in MCF-7 cells^{6,7}. However, sc-8002 does not work effectively in mouse tissue. We have now repeated the MCF-7 CUT&RUN experiment using the same antibody (EMD Millipore 06-935) that was employed in the brain CUT&RUN experiments. Our lab has previously validated this antibody by IF staining in *Vgat*^{Cre};*Esr1*^{lox/lox} animals and confirmation of its specificity for ChIP-seq has also been published^{8,9}. The findings from the two antibodies are concordant, and both experiments are now included in Extended Data 1.1. For both antibodies, antibody specificity is validated by: 1) the presence of the estrogen response element (ERE) as the top enriched motif in E2-induced ER α binding sites, 2) an MNase footprint surrounding the ERE in E2-induced binding sites, which indicates physical TF occupancy on chromatin¹, and 3) high concordance with a published MCF-7 ER α ChIP-seq dataset collected using a different ER α antibody (produced by W. Lee Kraus lab)¹⁰ (Extended Data 1.1). Moreover, the MCF-7 cell line does not express *Esr2*/ER β ¹¹; therefore these ER α

CUT&RUN sites cannot be attributed to cross-ER epitope recognition. Finally, the EMD Millipore 06-935 antibody targets an epitope within 15 amino acids from the C-terminal region, which should have little homology with ER β .

3. “Estrogen-induced gene regulation in the BNSTp was not shared across all ER α + brain regions, demonstrating regional specificity...” No evidence to support this statement is presented, nor any literature cited.

Originally, ISH images supporting this statement were located in Extended Data Fig. 2.2b (although they were not directly referenced in that sentence). However, given the additional focus on the BNSTp throughout the review process, which includes identifying factors that control physiological sex differences in BNSTp gene regulation, we feel that making a conclusive statement about estradiol regulation of gene expression across brain regions is beyond the scope of this paper. For that reason, we have decided to remove original Extended Data 2.2b and the corresponding text (see also response to reviewer #4, figure #2, point #3).

4. Expt of Fig. 3d: It would be useful to compare genes that are down-regulated in neo-natal males treated with an ER α antagonist, rather than just genes that are up-regulated in neonatal females treated with an ER α agonist.

We agree that this experiment is useful, in that it would demonstrate the necessity of neonatal ER α activation for the observed male-biased transcriptional program. However, we feel that treating neonatal males with an ER α antagonist would be insufficient to block this transcriptional program, as the testosterone surge emerges immediately after birth and subsides within ~4-6 hr¹², while ER α binds extensively to the genome within minutes of ligand exposure¹³. Moreover, there is ample evidence that ER α modulators, such as tamoxifen and raloxifene, can act as agonists depending on the tissue or cell type¹⁴ and are only recently being investigated for their transcriptional effects in brain¹⁵, whereas the ER α degrader ICI-182780 (fulvestrant) has been reported to not cross the blood-brain barrier¹⁶. As an alternative strategy to demonstrate ER α necessity in this transcriptional program, and also in response to reviewer #4 point #1, we have performed single-nucleus RNA-seq (snRNA-seq) on the BNST of male and female *Vgat*^{Cre};*Esr1*^{+/+};*Sun1-GFP*^x and male *Vgat*^{Cre};*Esr1*^{lox/lox};*Sun1-GFP*^x animals two weeks after the neonatal surge has subsided (P14). This approach revealed that the extensive sex differences in gene expression in the BNSTp are lost in ER α KO males (Fig. 4h), thus validating the requirement of ER α in sustained sex differences in gene expression.

5. Why are the data in Fig. 3d “consistent with the hypothesis that early life estrogen signaling organizes a lasting male-typical transcriptional program”? If the authors want to show this, they should perform the analysis of Fig. 3d on cell types from adult females exposed to neonatal estrogen, not on data from adult males drawn from a previously published study. Why are some of the neonatally induced transcriptional changes maintained and others lost in adulthood?

Thank you for raising this important point. We agree with the limitations of directly comparing P4 neonatal estradiol-treated females with adult male data (see also response to reviewer #2, point 2). We have collected additional data during the review process that improves our understanding of the persistence of gene regulatory changes arising due to the neonatal surge, as well as mechanisms controlling adult sex differences, which we outline below:

1. Bulk ATAC-seq of BNST *Esr1*⁺ neurons collected from neonatal vehicle-treated (NV) males, NV females, and neonatal estradiol-treated (NE) females on postnatal day (P4) revealed 1605 chromatin loci that are persistently open (NE-open) as a result of the neonatal surge (ie, higher in both NV males and NE females compared to NV females; Fig. 4a, Extended Data Fig. 4.1a). 403 chromatin loci were also persistently closed (NE-close) as a result of the neonatal surge, including at the *Esr1* locus, indicating neonatal estradiol both masculinizes and de-feminizes chromatin accessibility.
2. Single-cell multiome (snATAC+snRNA-seq) of BNST *Esr1*⁺ neurons from P4 females and males revealed extensive sex differences in gene expression across BNST *Esr1*⁺ inhibitory neuron types (Fig. 4f-g). Leveraging the innovation of multiome data, we characterized specific NE-regulated ATAC loci as putative enhancers of these sex DEGs (Fig. 4f), via identification of peak-gene pairs with significant correlation between accessibility and expression¹⁷.
3. Single-cell multiome (snATAC+snRNA-seq) of BNST *Esr1*⁺ neurons from P14 females and males revealed a subset of DEGs identified on P4, and their correlated NE-regulated ATAC loci, persist for up to 2 weeks following the neonatal surge (Fig. 4g). As these NE-regulated loci predominantly contain P0 ER α binding sites detected by CUT&RUN (Fig. 4a), we can conclude that these long-lasting gene regulatory changes are driven by neonatal ER α activation.
4. snRNA-seq of P14 male and female *Vgat*^{Cre};*Esr1*^{+/+};*Sun1-GFP*^{lx} and male *Vgat*^{Cre};*Esr1*^{lox/lox};*Sun1-GFP*^{lx} animals revealed these sustained sex DEGs require ER α (Fig. 5a-b, Extended Fig. 5.1, Supplementary Table 11).
5. Bulk ATAC-seq of BNST *Esr1*⁺ neurons from gonadally-intact adult (8-12 weeks-old) females and males revealed extensive sex differences in the chromatin landscape (~18,000 loci) that are not present in our previous GDX bulk ATAC-seq data (Fig. 3h), indicating adult sex differences in the chromatin landscape require adult gonadal hormone. Consistent with this finding, we detected minimal overlap between these adult sex-biased ATAC loci and our NE-regulated ATAC loci. Motif enrichment analysis of these adult male-biased peaks revealed the androgen receptor (AR) as the principal driver of these sites (Fig. 3i). These results indicate that adult sex differences in gene regulation are *primarily driven by adult gonadal hormones*. However, we speculate that NE-regulated loci/genes that persist until P14 likely serve to influence the response to adult hormones, ensuring appropriate display of male-typical behaviors.

Given this additional evidence, we stand by our initial claim that neonatal activation of ER α coordinates a lasting male-biased transcriptional program. However, we now acknowledge that puberty represents an additional critical window, during which time adult hormones dramatically increase sex differences in the chromatin accessibility of BNST *Esr1*⁺ neurons. Applying the same mechanistic approach to puberty that we've applied to understanding neonatal sex differences is, we feel, beyond the scope of this manuscript. The question of why certain transcriptional changes persist until P14 while others do not is indeed critical moving forward. However, we believe addressing this question will require additional insights into the epigenomic modifications (DNA methylation, histone PTMs), 3D chromatin architecture, and ER α binding partners in neonatal BNST *Esr1*⁺ neurons, which we aim to study in our future work.

6. Assignment of BNSTpr ERalpha cell types to cortical/hippocampal IN classes: I'm not sure if this is valid, since it uses two different datasets (both produced by other labs) derived using potentially different methods of scRNAseq.

On the contrary, integration of scRNA-seq datasets across sequencing technologies is now ubiquitous in the field of single-cell analysis^{18–22}. As progress is made toward the completion of large-scale reference cell atlases, such as the Human Cell Atlas and the Brain Initiative Cell Census Network (BICCN), the expectation is that these references will be used to aid in the interpretation of future scRNA-seq datasets generated from the same tissue. The tool that we have used, MetaNeighbor, has been extensively adopted by the BICCN to define consensus clusters across multiple scRNA-seq platforms (10X Genomics v2, 10X Genomics v3, SMART-seq)²³ and data modalities (scRNA-seq, scATAC-seq, scBS-seq)²⁴. It has recently been used to identify conserved cell types across 7 different vertebrate and invertebrate species, using independent samples collected from different labs²⁵. MetaNeighbor ranks cells in a query dataset by their similarity in gene co-expression to cells in an independent reference dataset^{24,26}. Because the core transcriptional program defining a cell's identity is robust to sequencing methodology, MetaNeighbor can score the similarity of clusters across datasets irrespective of technical variation.

We now provide additional evidence supporting our observation that ER α /Nfix+ neurons resemble *Lamp5*+ interneurons: 1) *Lamp5*+ neurons, specifically *Lamp5/Lhx6*+ clusters, selectively co-express the top marker genes of ER α /Nfix+ neurons (*St18*, *Nfix*, *Moxd1*, *Cplx3*) (Extended Data Fig. 3.1d), and 2) among BNSTp *Esr1*+ clusters, ER α /Nfix+ neurons have enriched expression of *Lamp5*+ marker genes (Extended Data Fig. 3.1e).

7. “32% of ER α peaks containing the NFI motif” – it would be useful to see the numbers for a different class of NHR peaks. Maybe NFI motifs are present in 32% of all NHR peaks.

We agree with the reviewer's suggestion that the interaction between NFI and NHR TFs is likely a general phenomenon. Co-binding of an NFI TF and the glucocorticoid receptor (GR) at the mouse mammary tumor virus (MMTV) promoter was first described *in vitro* over 30 years ago by Gordon Hager's group²⁷. The mechanism of NFI and GR co-binding was further elucidated in subsequent studies, which revealed bi-modal recruitment of GR and NFIs to chromatin in response to dexamethasone treatment^{28,29}. Work from Miguel Beato's group also revealed cooperativity between progesterone receptor (PR) and NFIs at the MMTV promoter^{30,31}. Additionally, more recent genome-wide sequencing approaches have revealed extensive co-binding between androgen receptor (AR) and Nfib in the LNCaP prostate cancer cell line^{32,33}, as well as enrichment of the NFI motif in GR binding sites in male rat hippocampus (around ~50-55% of GR binding sites containing the NFI half-site)³⁴. Given this existing knowledge on the co-binding of NFI and NHR TFs, we think little new information will be generated by re-plotting the % of NFI motifs in NHR ChIP-seq datasets. We note that our Nfix CUT&RUN data are the only genomic binding data for a NFI factor in the brain. We have reframed our analysis of Nfix binding to emphasize its role in defining the accessible chromatin landscape in the i1:Nfix neuron type (see response to point 9 below).

8. What percentage of BNSTp ER α + neurons co-express Nfix? The authors refer to a “subset,” but do not state the size of that subset.

We have now quantified the male-bias in BNSTp ER α /Nfix+ neurons by Nfix IF staining in *Esr1*^{Cre/+}; *ROSA26*^{CAG-Sun1-sfGFP-Myc/+} animals (Fig. 3c). We have included the % of BNSTp ER α + neurons co-labeled with Nfix in the main text:

We confirmed that ER α /Nfix+ neurons, which represent ~30% of the BNSTp *Esr1*+ population, are male-biased using immunofluorescent staining (Fig. 3c, Extended Data Fig. 3.1c).

9. “ER α and Nfix co-regulate a distinct module of estrogen-responsive and male-biased genes in the BNSTpr.” This may be semantics, but to my mind “co-regulation” implies some sort of cooperativity. Here it seems more likely that ER α opens chromatin, and that Nfix is bound at open-chromatin boundaries. It may look like a specific “co-regulation” just because Nfix expression is enriched in BNSTp.

We have reduced our focus on Nfix throughout the manuscript, as additional data collected during the review process have revealed that while ER α and Nfix co-regulate estradiol-responsive genes, Nfix is not the primary factor contributing to physiological sex differences in gene regulation. Rather, we find robust evidence that Nfix defines the identity of a male-biased BNSTp inhibitory neuron type (Fig. 3b-c, Fig. 4c), based on the observation that 1) the Nfix motif and Nfix CUT&RUN binding sites have enriched accessibility in the i1:Nfix (formerly labeled as BNSTpr_St18) transcriptomic cluster and 2) Nfix binds putative enhancers at the *Nfix* locus in i1:Nfix neurons, consistent with an auto-regulatory mechanism (Fig. 4c). Finally, the number of i1:Nfix cells is feminized upon genetic deletion of ER α /*Esr1* from inhibitory neurons (Extended Data Fig. 5.1d).

10. Evidence for CGE derivation of ER α /Nfix-expressing inhibitory neurons is just circumstantial and based on shared gene expression, which does not predict shared developmental ancestry. Fate-mapping studies are required to establish this point.

We appreciate this comment. We have removed the statement that BNSTp ER α /Nfix-expressing inhibitory neurons are conclusively a CGE-derived population, as we agree that comprehensive fate-mapping is required to demonstrate this point.

We stand by our initial observation that ER α /Nfix-expressing inhibitory neurons resemble CGE-derived neurons, on the basis that *Nfix* is selectively expressed in the embryonic CGE and, among adult inhibitory neurons, is selectively expressed in those deriving from the CGE^{35,36}.

Referee #2 (Remarks to the Author):

As the authors point out, little is known about the direct gene targets of estradiol in the brain, despite the importance of this hormone in the establishment of sex differences and the control of physiology and behavior. The authors use a suite of the most modern molecular techniques to address this and related questions. Without a doubt, this is the most in-depth analysis of estrogen-regulated gene expression in the brain to date, with many interesting findings along the way. I am amazed that the authors were able to accomplish what they did, using highly heterogeneous and hard-to-work-with brain tissue.

A number of the findings in the first half of the paper are confirmatory. For example, the authors find that the preponderance of estrogen-regulated genes have an ERE, and that at the large majority of direct ER α targets, expression is up- (rather than down-) regulated by estradiol. These conclusions will not surprise anyone, and have been reached over the years by others using older techniques. Other findings (mostly in the second half of the paper) are truly new. For example, the provision of a comprehensive list of ER binding sites in the brain

is novel and provides molecular candidates that may underlie effects of estradiol both normally and in disease, and the large number of genes epigenetically regulated by ER binding (based on chromatin accessibility) is surprising and, as far as I know, new.

There are three general areas of concern, however, that dampened my enthusiasm.

1. The paper is difficult to read.

For someone willing to do the work, there is much to be mined from this paper. However, even for someone intensely interested in the topic, getting through the paper is hard work. It is dense both because of the amount and type of data being presented, and also because the writing doesn't do enough to help the reader along.

We appreciate the reviewer's comment and have made systematic changes to the main text to improve readability for a broader audience.

2. The emphasis on "neurodevelopment" seems misplaced (or unconvincing)

The authors use adult gonadectomies followed by acute estradiol treatment, to conclude that estradiol induces a "robust neurodevelopmental gene regulatory program." The emphasis on "neurodevelopmental" in several places in the paper is puzzling, since the findings are based on experiments performed in adults, and many of the genes identified perform roles such as the regulation of membrane potential and synapse organization, which are important throughout the life of a neuron.

We agree that using the term "neurodevelopment" to describe adult estradiol gene targets is inaccurate. "Neurodevelopment" was chosen as a singular term to capture the diversity of synaptic plasticity, cell adhesion molecule, and axonogenesis genes identified in our adult genomic datasets. However, as the reviewer correctly points out, many of these genes are not restricted to a neurodevelopmental function. We have now replaced "neurodevelopment" with "neuronal connectivity" in the abstract, Fig. 2 results section title, and main text. We feel that "neuronal connectivity" spans the molecular function of estradiol gene targets and offers explanatory power to the many studies demonstrating the effects of estradiol on neuron morphology, dendritic spines, and axonogenesis/axon regeneration.

The only experiment performed during development treated newborn females with estradiol or vehicle at birth and examined gene expression on P4. The authors find that 33% of the estradiol-induced, ER target genes are also among those with male-biased expression in adults and conclude that this is "consistent with the hypothesis that early-life estrogen signaling organizes a lasting, male-typical transcriptional program" "...that likely specify and maintain sex differences in neuronal firing and synaptic patterning." I'm not convinced that the authors have shown this. Wouldn't an equally likely explanation be simply that some genes are regulated by estradiol both in newborns and also later in life (i.e., nothing is "set up" by the early estradiol)? It also is surprising that there were not newborn males in this experiment, to support conclusions about the start of a male-typical transcriptional program.

To address this point, and point #3, we have performed several experiments to both characterize gene regulation during the neonatal critical period, and to understand how adult ER α target genes are invoked in gonadally intact animals. We performed ATAC-seq in gonadally intact animals and subsequently mapped adult male-biased ATAC loci to adult male-biased genes (explained in more detail for point #3). We discovered AR is

the top enriched motif in male-biased ATAC peaks (Fig. 3i). Moreover, we have performed ATAC-seq in P4 animals (explained in more detail for response to “Related to points 2 & 3”) and identified male-biased ATAC peaks that are induced by the neonatal surge. We find that P4 male-biased ATAC peaks are enriched for the ERE and have only minor overlap with adult male-biased ARE-containing ATAC peaks. Therefore, while certain neonatal E2 target *genes* overlap with adult male-biased *genes* (as shown in original Fig. 3d), the corresponding accessible genomic *loci* at these genes are largely distinct (Extended Data Fig. 4.2). Hence these data suggest ER α and AR regulate an overlapping set of genes at different life stages. For this reason, we have removed the statement “consistent with the hypothesis that early-life estrogen signaling organizes a lasting, male-typical transcriptional program” to avoid confusion.

Regarding the inclusion of newborn males, we have now performed single-nucleus multiome (RNA+ATAC) sequencing, simultaneously capturing both RNA and chromatin accessibility from the same cells, on P4 and P14 female and male BNSTp *Esr1*+ cells to 1) identify physiological sex differences in gene expression at single-cell resolution and 2) determine the extent to which these transcripts maintain sex-biased expression across time. We have also performed ER α CUT&RUN in P0 animals, described in more detail below (see response to “Related to points 2 & 3”).

3. Confusion about Sexual Differentiation

One major thrust of this paper regards sexual differentiation and, in fact, the Introduction starts with a description of sexual differentiation during perinatal life. However, the findings regarding sexual differentiation seem somewhat contradictory. The authors present data supporting the conclusion that, “both sexes possess similar capacity to mount a transcriptional response to estrogen” and elsewhere, “... ER α binding ... also lacks sex-specific recruitment.” At the same time, however, the paper goes on to identify what the authors call a “male-biased gene expression signature” (Abstract and elsewhere). **Either there is, or is not, sexual differentiation of gene expression, yet the manuscript seems to have it both ways.**

I suspect that some of the confusion comes from the fact that animals were gonadectomized and given acute estradiol in the studies performed here, but were gonadally intact in the single-nucleus RNA-seq dataset by Welch et al (2019) that the authors re-analyze and compare to some of their data. The reader is not told about this difference, however. I think the authors try to address this in the Discussion, but too late to avoid confusion in the mind of the reader. I wonder also if the authors are missing an opportunity to address the potentially non-physiological nature of the gonadectomy-acute-estradiol model (a paradigm that allows one to synchronize estradiol exposure but is not a situation a mouse would ever encounter). In other words, to the extent that there is overlap between the male-biased genes (Welch et al.) and the ER binding sites (current study), it may support the physiological relevance of the authors’ findings.

We appreciate this point. We agree that the presentation of these results is confusing and have completed additional experiments, as well as clarified the results section, to mitigate this confusion. Given the non-physiological nature of the GDX+estradiol paradigm, we decided to profile physiological sex differences in chromatin accessibility in adult BNSTp *Esr1*+ neurons and compare to both the adult single-nucleus RNA-seq (snRNA-seq) data as well as our GDX ATAC-seq data. This approach revealed extensive sex differences in ATAC peaks that have matching enrichment at sex-biased genes in the snRNA-seq data (Fig. 3h), meaning *female-biased peaks putatively control female-biased genes, and male-biased peaks control male-biased genes*. Given our observation that both sexes possess a similar capacity to mount a genomic response to

estradiol, we hypothesized that GDX largely ablates sex differences in adult BNSTp *Esr1*+ neurons, thus enabling a similar response to estradiol treatment. In support of this hypothesis, we find that sex-biased ATAC peaks are largely abolished following GDX (Fig. 3h-i). Specifically, female GDX causes minimal sex-specific changes in chromatin accessibility, while male GDX both robustly closes male-biased sites and opens female-biased sites (Fig. 3i, Extended Data Fig. 3.2i-k). This novel finding *indicates adult sex differences in chromatin loci are largely dependent on male gonadal hormones*.

Related to points 2 & 3: Given how the authors start the paper ("how transient exposure to estrogen organizes lasting sex differences remains a central question,") would it have been more fruitful to look at ER binding sites in Male, Female, and Female + E2 newborns (around the time of the testosterone surge)? Alternatively, the authors could change their Introduction so it is less focused on sexual differentiation, and more on what estradiol is doing in adults, since the bulk of the data is in adults.

We agree with this point. Instead of changing our introduction, we decided to collect more genomic data from neonatal animals, as this was unanimously requested by the reviewers. We performed ATAC-seq on BNST *Esr1*+ neurons from P4 females and males (Fig. 4a), matching the timing of our bulk neonatal RNA-seq experiment. Females were treated with vehicle (NV) or estradiol benzoate (NE) on the day of birth to identify the extent to which sexually dimorphic chromatin regions are dependent on neonatal estradiol. This approach revealed 1605 chromatin loci with increased accessibility in NV males and NE females compared to NV females (NE-open) and only a handful of regions, primarily located on chromosome Y, with increased accessibility in NV males alone (Fig. 4a, Supplementary Table 6).

We attempted to profile physiological ER α binding sites in P0 males with several different approaches (see response to reviewer #4, major point #2); however, none proved successful, which we believe is due to the transient nature of the neonatal testosterone surge. Instead, to validate that neonatal E2/male-biased ATAC loci are indeed driven by ER α , we performed P0 ER α CUT&RUN on FACS-purified *Esr1*+ nuclei harvested from females after 4 hr treatment with vehicle or E2. This approach revealed strong overlap between P0 ER α binding sites and male-biased loci – ~70% of NE-open loci contain ER α binding sites – (Fig. 4a, Extended Data Fig. 4.1), thus validating that chromatin accessibility induction is driven by neonatal ER α activation.

Minor comments/corrections

Male/Female labels seem to be missing on the quantification in Figure 2b.

We have added these labels to the ISH quantification plots in Fig. 2b.

The Figure legends often don't give the reader enough information to know what they are looking at.

We have provided additional details to the figure legends, wherever possible, while still following the formatting requirements of the journal.

In what sense is postnatal day 4 "the peak of BNSTp sexual differentiation?"

We have modified the text to provide additional information regarding the timing of BNSTp sexual differentiation. Postnatal day 4 (P4) represents the onset of male-biased BNSTp cell survival and axon outgrowth^{37,38}.

To identify this program, we performed ATAC-seq on BNSTp *Esr1*+ neurons collected four days after the neonatal hormone surge (P4), which corresponds to the onset of male-biased BNSTp cell survival and axonogenesis^{44,45}.

Is it correct that “all BNST neurons are GABAergic” (second paragraph of Discussion), or just that many of them are?

All neurons annotated to the BNSTp in the snRNA-seq dataset are indeed GABAergic, based on the expression of *Gad1/Gad2* and lack of *Slc17a6*¹⁹. We have also previously demonstrated this by comparing ER α IF staining in P0 *Vgat*^{Cre};*Esr1*^{lox/lox} and *Vglut2*^{Cre};*Esr1*^{lox/lox} animals⁸.

Referee #3 (Remarks to the Author):

This manuscript presents an in depth-analysis of transcriptional regulation by estrogen receptor -alpha (*Esr1*) in a distinct region of the mouse brain known to be functionally important to sex differences in social behaviors. Much of what they discover regarding ER-alpha interaction with the genome is unique to the brain and thus, not surprisingly, are mostly relevant to parameters associated with neural functioning. The work appears to be expertly done and is among the most exciting and progressive on the important topic of biological origins of sex differences in the brain. The weakness is in presentation in terms of making the significance of the findings accessible to the reader, even the experts in the field. Specific comments below.

1) The strengths of the report include the sophisticated approaches providing an unprecedented level of information regarding hormonal modulation of transcription in males versus females in a brain region known to be “programmed” or “organized” by steroids early in life. However there is some muddling of the message that at times makes it unclear what are the take aways. For instance, the authors state in the Abstract that “estrogen induces a robust neurodevelopmental gene regulatory program in both sexes” – but they do not discuss that the fundamental principle of sexual differentiation of the rodent brain is that females are not naturally exposed to estrogens. It has been known for 50 years that females can be masculinized by treating them with exogenous estradiol as neonates. Are the authors trying to say that the same transcriptional profile is induced in females as occurs naturally in males during development and hence explaining how sexual differentiation occurs? Or, are they saying something quite different, that transcriptional regulation does NOT explain sexual differentiation of behavior. It is difficult to tell.

Thank you for raising this point. Reviewer #2, point #2 also raised concern about the inclusion of “neurodevelopmental gene regulatory program” in the Abstract, and we have provided a detailed solution. We agree that this terminology is confusing. In short, we are saying the former point – that the same transcriptional profile can be induced in adult females as occurs naturally during development in males, hence explaining how sexual differentiation occurs. We have now included additional chromatin data from neonatal animals (see

responses to reviewer 2, points #2-3) solidifying this conclusion, as the majority of neonatal male-biased chromatin loci (~85%) are induced by estradiol in the adult BNSTp of both sexes.

2) Related to this, I am confused by the header “Estradiol induces a neurodevelopmental gene regulatory program in the adult BNSTp” – it is not clear at what age animals were treated, did both sexes receive the same treatment and why this is a neurodevelopmental program? Synaptic organization does not imply development given what we now know about synaptic plasticity. The authors appear to conclude that there is no impact of developmental programming by estradiol on adult estradiol responding. This may be true at the transcriptional level but it certainly is not true at the behavioral level. This section also includes some over interpretations as are present throughout, such as the idea that because the glucocorticoid receptor is expressed at higher levels in females this might explain sex differences in stress responding. Other references to relevance of the current report to disease are also over extensions.

We appreciate this point. Reviewer #2, point #2 also raised concern about the use of the term “neurodevelopmental” to describe adult estradiol gene targets, which we have responded to. In short, we have replaced “neurodevelopmental” with “neuronal connectivity” for conveying the function of estradiol target genes.

We agree that behavioral responses to adult E2 are sex-specific, due largely to irreversible organization of BNSTp circuitry during the neonatal critical window. We have modified this sentence to specifically refer to the shared genomic response to E2:

In contrast, across ER α CUT&RUN and ATAC-seq modalities, we observed negligible sex differences and sex-specific E2-regulated changes (Fig. 2f, 2h, Extended Data Fig. 2.3g, Supplementary Table 3), demonstrating that females and males can mount a shared genomic response to exogenous estradiol independently of differential exposure to hormones throughout the lifespan.

We have toned down our discussion of neuropsychiatric and neurodevelopmental disease in the Introduction and Discussion. However, we believe our finding that neural ER α target genes are implicated in these diseases deserves mention. The relevance of estrogens to mood, anxiety, and psychotic disorders has been known for over 30 years^{39–41}. As the onset of these disorders often coincides with windows of hormonal fluctuation (e.g., puberty, pregnancy, and menopause), many have speculated that hormone levels interact with genetic and environmental factors to influence disease risk and severity. Randomized, double-blind clinical studies have further shown estradiol can alleviate symptoms of schizophrenia and depression in women^{42–45}. More recently, estrogens have been linked to male-biased neurodevelopmental disorders, primarily autism spectrum disorder (ASD)⁴⁶. Estrogens are elevated in the amniotic fluid of autistic boys, and compared to other steroid hormones, estradiol is the strongest predictor of autism likelihood⁴⁶. Estradiol also interacts with large-effect ASD-associated genes in 2D and 3D *in vitro* models of human brain development⁴⁷.

Our work extends these findings by demonstrating, for the first time, that estradiol regulates genes implicated in anxiety, depression, and ASD via ER α genomic recruitment in the brain. Moreover, we have now identified that many of the ASD ER α target genes, including *Arid1b*, *Scn2a1*, *Nrxn1*, and *Grin2b*, are regulated by the male-specific neonatal hormone surge, demonstrating their neurodevelopmental relevance (Extended Data Fig. 4.1, Supplementary Table 6).

3) The next section speaks to male specific transcriptional changes which involve ER-alpha binding, so at first glance appears to contradict the above. It was not clear to this reviewer if the authors are arguing these DEGs are unrelated to the hormone, estradiol, but are a function of its receptor, ER-alpha? They go on to hypothesize that certain genes more highly expressed in males are the result of a “persistent transcriptional signature of neonatal estradiol”, yet they emphasize that both male and female brains respond to estradiol the same, so which is it? The point being, if there is a clear conclusion here, it is not easy for the reader to grasp.

Reviewer #2, point #3 as well as reviewer #4, figure #3, point #4 raised a similar concern about the unclear conclusions regarding a “persistent transcriptional signature of neonatal estradiol”. We have collected additional data from neonatal and adult animals during the review process, as well as modified the text, to clarify this point. We have identified that while ER α turns on many of the same *genes* in the neonatal brain that have male-biased expression in adulthood, only a small proportion of ER α binding sites have male-biased accessibility both neonatally and in adulthood (~10%). Rather, in the adult brain, we detected ~12,800 male-biased ATAC sites in gonadally intact animals that putatively regulate male-biased genes and primarily contain the motif for androgen receptor (AR) (Fig. 3h-i). Therefore, our results indicate ER α and AR regulate an overlapping set of genes following birth and puberty, respectively, via recruitment to distinct binding sites (Extended Data Fig. 4.2).

4) The last section concludes that a subset of ER-alpha expressing inhibitory neurons “define sexual dimorphism in the BNST” but this is a bridge too far beyond the data. There is no evidence here that the sex difference discovered plays any role in either the morphology of the BNST or the behaviors it regulates. It is a possibility, but right now is speculation. Steps to consider for making the causal connection are morphometric phenotyping of the *Esr1-Nfix* expressing neurons in term of dendritic complexity, synaptic density and afferent and efferent inputs. Ultimately manipulating those cells specifically via optogenetic or chemical means to modulate sex differences in behavior, would fully close the loop.

We agree with this point. Our intention was to demonstrate that male-biased ER α /*Nfix*⁺ inhibitory neurons have a shared transcriptomic identity with SDN-POA *Moxd1*⁺ neurons that are selectively activated during three different male-typical behaviors: mounting/intromission, inter-male aggression, and pup-directed aggression⁴⁸. We do not intend to claim that these are the *only* neurons responsible for sexual dimorphism in the brain, nor make claims about their morphology or activity, which is currently infeasible due to a lack of *Cre*- and *Flp*-lines enabling specific targeting of these cells. We have amended the text so that it now reads:

Moreover, two of the top marker genes for *i1:Nfix* neurons, *Moxd1* and *Cplx3* (Extended Data Fig. 3.1b, f-g), were previously identified as markers of a male-biased SDN-POA neuron type (*i20:Gal/Moxd1*) that is selectively activated during male-typical mating, inter-male aggression, and parenting behaviors⁴². Beyond these two marker genes, we found that *i1:Nfix* and *i20:Gal/Moxd1* neuron types have a shared transcriptomic identity, in line with *Nfix* immunofluorescence across both the BNSTp and SDN-POA (Fig. 3e, Extended Data Fig. 3.1h). Together, these results define male-biased neurons in the BNSTp and reveal a shared *Lamp5*⁺ neurogliaform identity between BNSTp ER α /*Nfix*⁺ inhibitory neurons and SDN-POA neurons that are engaged during male-typical behaviors.

Minor points

1) It is important to get the nomenclature correct when referring to steroid hormones. Here the authors tend to use the term “estrogen” and “estradiol” interchangeably or, at times, incorrectly. Estrogens are a class of compounds that include estradiol, estriol and estrone. Thus one cannot treat with “estrogen” only with one of the 3 steroids mentioned above. The lead in sentence of the abstract is incorrect as there is no “endogenous form” of estrogen. Wording should be changed throughout to reflect that estrogen is a category and estradiol is a hormone.

We appreciate the reviewer’s comment on the proper nomenclature. We have replaced “estrogen” with “estradiol” throughout the text.

2) Related to the above, the authors treat mice with estradiol-benzoate, a modified version of 17-beta-estradiol that prolongs it’s action, however the hormonal effects are that of estradiol with the benzoate moiety serving no signaling role. The authors use EB to refer to estradiol benzoate, which is customary for neuroendocrinologist, but it may be better to replace with E2, the chemical name of 17-beta-estradiol, for a more general audience.

We also appreciate the reviewer’s comment on using a more suitable abbreviation for estradiol benzoate. We have replaced “EB” with “E2” throughout the text and in all figure panels.

3) Lastly, in the Abstract that authors state that estradiol helps establish sex differences in the vertebrate brain but this is an over generalization, particularly in that there is no definitive evidence that estrogens are important to primate brain sexual differentiation, including humans.

We agree that there is little definitive evidence showing estrogens control primate brain sexual differentiation; overall, there is little information on the specific mechanisms of brain sexual differentiation in primates. For instance, it remains unknown which brain regions and neuron types express *Ar*, *Esr1*, and *Cyp19a1* in the developing primate brain. However, estrogens have been shown to establish brain sex differences across a variety of other vertebrate species throughout the animal kingdom, including birds ⁴⁹ and ferrets ⁵⁰. Therefore, we have toned down the first sentence of the Abstract so that it now reads:

Estradiol establishes neural sex differences in many vertebrates¹⁻³ and modulates mood, behavior, and energy balance in adulthood⁴⁻⁹.

Referee #4 (Remarks to the Author):

Summary: In this work, the authors seek to address the interesting question of how sex-dependent gene expression programs are established in hypothalamic circuitry via the estrogen receptor alpha (ER) transcription factor. The authors combine genomic methods including CUT&RUN and ATAC-seq to map binding of ER and loci whose accessibility is responsive to estradiol exposure in vivo. The authors also perform a re-analysis of previously published single cell RNA-seq data to show that ER binding sites are enriched at genes whose expression is male-biased, suggesting that ER could regulate the establishment of male-biased gene expression. The authors identify Nfix as a transcription factor that potentially coordinates with ER in a subset of inhibitory neurons. Overall, the work presented is technically well done and will provide an important

resource for the field of sex-biased differences in neuronal gene expression. However, there are a number of points that should be addressed before publication. Most notably, the authors should attempt to test their claims using direct perturbations of ER and Nfix function as outlined below.

Major Points: The authors use bioinformatic approaches to arrive at a potential model of how Nfix and ER might collaborate to specify male-specific gene expression. A main criticism of the paper, however, is that they provide no experimental perturbation to test this model. For example, can the authors genetically remove Nfix or ER during the critical period (for example by injecting Cre-expressing viruses into conditional KO animals) and show that loss of these factors has a specific effect on male-biased genes in brain regions such as the BNSTp?

We agree that the use of an experimental perturbation is critical for the paper. Upon collection of additional genomic data (see additional points), we decided to focus on the requirement of ER α in specifying a sustained male-specific transcriptional program that initiates following the neonatal surge. Because testosterone production occurs immediately following birth on P0¹², we considered it unlikely that delivery of a Cre-expressing virus into the BNST of P0 *Esr1*^{lox/lox} animals would act quickly enough to remove ER α protein prior to hormonal activation. Instead, we decided to generate *Vgat*-Cre conditional deletion males (*Vgat*^{Cre};*Esr1*^{lox/lox}) as well as cross in the *Sun1-GFP* nuclear reporter line to selectively profile *Vgat*⁺ nuclei harboring the deletion. To that end, we have performed single-nucleus RNA-seq (snRNA-seq) on P14 BNST inhibitory neurons collected from *Vgat*^{Cre};*Esr1*^{+/+};*Sun1-GFP*^x female and male and *Vgat*^{Cre};*Esr1*^{lox/lox};*Sun1-GFP*^x male animals. This approach revealed loss of persistent male-biased genes, which we demonstrate are largely induced by neonatal ER α activation, across BNST inhibitory neuron clusters in the absence of ER α protein (Fig. 5a-b).

While this may be a challenging experiment, it would greatly improve the paper if the authors could attempt to map endogenous binding of ER in males vs. females during the perinatal period. It may be that the authors are missing interesting key differences in ER-dependent gene regulation by profiling with a strong estradiol stimulus at adult ages. This would also help to clarify the finding that despite ER's capacity to bind similar sets of loci in estradiol-stimulated adult females and males, these binding sites are enriched within genes with male-biased expression.

We agree that capturing the endogenous binding of ER α during the perinatal period is of high importance. We performed several experiments to resolve endogenous ER α binding in P0 males. The following conditions were tested: 1) 500,000 bulk nuclei from P0 BNST/MPOA/MeA, 2) 1 x 10⁶ bulk nuclei from P0 BNST/MPOA/MeA, 3) ~175,000 *Esr1*⁺ nuclei (collected via FACS) from P0 BNST/MPOA/MeA. None of these approaches was able to resolve more than a few robust binding sites (see table below). As mentioned in our response to reviewer #1, point #4, the testosterone surge emerges shortly after birth and subsides within ~4-6 hr¹², and the persistence of neurally-derived estradiol is unclear. *In vitro* the dwell time of ER α on chromatin following ligand treatment is primarily on the order of milliseconds⁷. Therefore, despite our best efforts to capture endogenous ER α binding, which involved systematic monitoring of breeding cages every 6 hours in order to receive a single litter containing a minimum of 5 male neonates to pool from, this approach proved unsuccessful.

Male P0 ER α CUT&RUN conditions	MACS2 peak number
500,000 bulk nuclei	9
1x10 ⁶ bulk nuclei	19
175,000 FACS-sorted Esr1 ⁺ nuclei rep 1	6
175,000 FACS-sorted Esr1 ⁺ nuclei rep 2	8

However, given that our adult ATAC-seq data revealed ER α -dependent chromatin opening can persist following ER α binding, we hypothesized that neonatal ER α activation coordinates a sustained genomic response detectable at the level of chromatin accessibility. Therefore, we performed ATAC-seq on BNSTp *Esr1*⁺ nuclei from P4 females and males (Fig. 4a), matching the timing of our previous bulk neonatal RNA-seq experiment. Females were treated with vehicle (NV) or estradiol benzoate (NE) on the day of birth to identify the extent to which sexually dimorphic chromatin regions are dependent on neonatal estradiol. This approach revealed 1605 chromatin loci with increased accessibility in NV males and NE females compared to NV females and only 15 sites (7 of which on chromosome Y), with increased accessibility in NV males alone (Fig. 4a, Supplementary Table 6). The ERE was the top enriched motif in these 1605 loci, and ~85% of sites overlapped sites that were opened by E2 in adult BNSTp *Esr1*⁺ cells (Extended Data Fig. 4.1b-c). Therefore, we have both captured the chromatin loci targeted by the neonatal surge and provided additional physiological relevance to our adult treatment paradigm.

To validate that neonatal E2/male-biased loci are indeed driven by ER α , we also performed P0 ER α CUT&RUN on FACS-purified *Esr1*⁺ nuclei harvested from females after 4 hr treatment with vehicle or E2. This approach revealed strong overlap between P4 NE-open loci and P0 ER α binding sites (Fig. 4a), thus validating the sites are driven by neonatal ER α activation.

Figure 1 and Extended Data

1. The experimental design of ER binding should be better explained in the text for a general audience. For example, given that the authors are interested in factors that establish sex-specific differences, it is surprising they chose to perform the experiments in the adult rather than during the perinatal critical period. The authors should clearly state the age at which these experiments are conducted and the rationale. In addition, the choice to use gonadectomized females and males needs to be explicitly clarified for a non-specialist audience. Do these animals develop normal hypothalamic circuitry (i.e. connectivity and cell number/distribution) such that the regions the authors are assaying in these mice are comparable to mice that have not undergone such treatment?

We appreciate the request for additional information regarding our approach. Our interest is in both characterizing the genomic response to estradiol and understanding the factors that establish sex differences in the brain. As there have been no previous studies examining estradiol-regulated genomic mechanisms in the brain with contemporary sequencing approaches, we first sought to understand the effects of estradiol in the adult brain. In order to control for circulating hormone levels, we opted to use the adult (8-10-weeks old) gonadectomy (GDX) + E2 replacement paradigm, as this approach is standard for modeling sex-typical behaviors (i.e., male-typical mounting and intromission, female-typical lordosis) in the neuroendocrinology field

⁵¹. Moreover, adult GDX+E2 replacement is an *in vivo* parallel to the *in vitro* culture conditions (48 hr in hormone-free media + acute estradiol treatment) that cancer biologists have used for two decades to examine ER α genomic binding in MCF-7 breast cancer cells ⁵². Thus our paradigm, both conceptually and experimentally, bridges a long-standing gap between these two fields. We have provided more information in the beginning of the text to explain these points:

To determine the genomic targets of ER α in the brain, we used a hormone starvation and replacement paradigm that elicits sex-typical behaviors² and replicates the media conditions required to detect ER α genomic binding in cell lines¹⁸. We gonadectomized (GDX) adult females and males then treated animals with estradiol benzoate (E2) or corn oil vehicle (Veh) following three weeks of hormone starvation. Four hours after treatment, we profiled ER α genomic targets within interconnected limbic circuitry in which ER α regulates sex-typical behaviors: the posterior bed nucleus of the stria terminalis (BNSTp), medial pre-optic area (MPOA), and posterior medial amygdala (MeAp)^{13,15,19}(Fig. 1a). We used the low-input TF profiling method CUT&RUN, which we first optimized in MCF-7 breast cancer cells by comparing with prior ChIP-seq data (Extended Data Fig. 1.1).

As GDX is performed post-puberty, long after the BNSTp neonatal critical period has concluded ^{37,38}, the overall circuitry is considered to be unaffected. There are known changes in gene expression ⁵³ and dendritic spines ^{54,55} following GDX; however, we don't consider these changes as invalidation of the hormone-controlled GDX paradigm, since they are reinstated in GDX animals with hormone treatment. Moreover, as mentioned in our other points, we have now discovered that the GDX+E2 paradigm recapitulates normal physiology in two different ways: 1) mimicking the effect of the male-specific neonatal surge and 2) capturing loci that are maintained by adult gonadal hormones in both sexes.

2. How do the levels of estradiol introduced subcutaneously in the treatment paradigm compare to physiological levels of estrogen? Is there a way to examine ER binding in response to physiological levels of estrogen? For example, can the authors profile ER binding in female mice upon entry to proestrus when circulating levels of estrogen are high? Alternatively, can the authors titrate the levels of estradiol to which mice are exposed to more closely mimic physiological levels?

After performing ATAC-seq on BNSTp *Esr1+* neurons from gonadally intact animals and comparing to GDX ATAC profiles, we found that sites decreasing in accessibility in both sexes following GDX primarily overlap E2-open ATAC sites, whereas sites decreasing specifically in males do not (Fig. 3i-j). This finding reveals that physiological estradiol, produced peripherally by ovaries in females or locally via aromatase in males, maintains chromatin accessibility in the brains of both sexes.

Given this finding, we attempted to profile the endogenous binding of ER α in gonadally intact females and males. However, this approach did not reveal robust binding events (see below), which we attributed to the transient dwell time of ER α on chromatin in the absence of strong ligand exposure.

3. The observation that ER binds similarly in males and females (Extended Data Fig. 1.2) should be presented in Figure 1 as it is a major finding given how the authors explain the goals of their study in the introduction.

We appreciate the reviewer's request. We have now split the ATAC-seq heatmap in Fig. 2f by sex to demonstrate that E2 induces similar ER α binding and chromatin opening in both sexes. Moreover, we have extended our previous analysis of sex-specific E2 changes by directly testing for an interaction between sex and treatment in our edgeR GLM design, as has recently been recommended in the field⁵⁶. This approach validated our initial findings, as there are 0 peaks across either assay with a significantly differential response to E2 between sexes (edgeR, padj<0.1). We have added volcano plots displaying this result as a panel in Fig. 2h (replacing our original Extended Data Fig. 1.2f and Extended Data Fig. 2.3j-k) and added the results from these statistical tests to Supplementary Tables 1 and 3.

Surprisingly, when we tested for an interaction between sex and treatment in our RiboTag data, we detected 155 genes (padj<0.1) with a sex-differential response to E2 (Fig. 2g). E2-open ATAC sites and ER α binding sites are not significantly enriched at these genes relative to non-differential, expressed genes (Extended Data Fig. 2.3f). As the RiboTag method selectively captures ribosome-bound mRNA, it is possible that these genes are regulated by estradiol at the translational level, consistent with the recent discovery that ER α can act as an RNA-binding protein⁵⁷.

4. In Extended Data 1.2D, how do the authors explain a lack of ER binding motifs at the EB-downregulated peaks? Are these bona fide ER peaks? What is happening to evict ER from these sites? Without an ERE motif, it seems possible that these peaks might be spurious or a reflection of background chromatin accessibility at these sites. Accordingly, I am not sure that these peaks represent "coordination of looping at chromatin boundaries."

We considered the possibility that these sites may reflect spurious pA-MNase cleavage events. However, given that they 1) are statistically differential between the vehicle and E2 treatment conditions, 2) do not overlap background peaks called by MACS2 in either of our vehicle and E2 IgG CUT&RUN controls, and 3) do

not overlap E2-regulated ATAC loci (see below), we cannot exclude them as spurious or reflective of a change in the background accessibility.

While ER α does primarily bind to EREs via its DNA-binding domain, one of the canonical ER α genomic mechanisms is “tethering”, or binding indirectly to chromatin via interactions with other DNA-bound TFs⁵⁸. A number of TFs, spanning different protein families, have been reported to tether ER α ^{59,60}. Moreover, ligand-independent ER α binding at non-ERE containing peaks has previously been reported on a genome-wide scale¹³. Therefore, the lack of EREs does not necessarily indicate a lack of specific ER α binding. There are a number of possible mechanisms explaining why ER α signal could *decrease* at a subset of loci following ligand treatment. For instance, it is possible that E2 itself causes unliganded ER α to change its protein conformation, such that it is released from other TFs and subsequently binds ERE-containing sites. It is also possible that E2 causes additional cofactors to bind ER α that block the epitope from antibody recognition, as has been previously reported for CHIP⁶¹. We agree that stating the peaks represent “coordination of looping at chromatin boundaries” is a step too far and have removed the statement.

5. In Extended data 1.3C, the authors should also compare ERE strength for the ESR2 motif, as this is more highly enriched than the ESR1 motif in their data.

We have included a comparison of ERE strength for the ESR2 motif as well (now shown in Extended Data Fig. 1.2f). In general, ER α and ER β have an identical DNA-binding domain and bind the same motif *in vitro*⁶²; we believe ESR2 is called as having a lower adjusted p-value in the motif enrichment analysis because of its motif position weight matrix (PWM) in the JASPAR database, which is more tolerant of base mismatches than the ESR1 PWM.

6. I would like to see more characterization of the ER antibody used in this study. Does the antibody recognize ER beta? Given how central the CUT&RUN results are to this study, the authors would also do well to perform a validation experiment in KO control animals rather than relying solely on IgG controls.

We appreciate the reviewer’s request for additional characterization of the ER α antibody used in this study. We have previously validated antibody specificity by performing ER α immunofluorescence (IF) staining of P0

BNST in *Vgat^{Cre};Esr1^{lox/lox}* and *Vgat^{Cre};Esr1^{lox/+}* animals, which revealed complete loss of staining in BNSTp inhibitory neurons in KO animals ⁸.

Moreover, we have now performed ER α CUT&RUN in MCF-7 breast cancer cells, which do not express *Esr2* ¹¹, using the same antibody that we used for *in vivo* CUT&RUN. This experiment revealed concordant binding between different ER α antibodies (Santa Cruz sc-8002 and EMD Millipore 06-935) and different methodologies (CUT&RUN vs. ChIP-seq). Reviewer 1 also asked about antibody specificity, please see our response to their point #2 for additional details.

Finally, as we also performed ER α CUT&RUN in *Esr1+* nuclei harvested from P0 *Esr1^{Cre/+};Sun1-GFP^{lox/+}* animals via FACS, we were able to perform ER α CUT&RUN in *Esr1-* nuclei from E2-treated animals. This approach revealed an overall lack of ER α binding events in *Esr1-* nuclei (Fig. 4a), further validating antibody specificity.

Figure 2 and Extended Data

1. Can the authors provide evidence that *Esr1^{Cre/+}; Rpl22HA/+* and *Esr1^{Cre/+};Sun1-GFP* are marking the same sets of cells in males and females? This is important for interpreting sex differences in gene expression presented in this figure.

Our analysis of the adult BNST snRNA-seq dataset ¹⁹ indicates both sexes contain the same populations of BNSTp *Esr1+* neurons, in that no populations were exclusively detected in one sex and not the other. We have now provided a UMAP plot colored by sex in Fig. 3a to demonstrate this point (also in response to Figure 3, Point #1). While we have not systematically examined BNSTp cell populations in *Esr1^{Cre/+}; Rpl22^{HA/+}* animals, we have now performed single-nucleus multiome (RNA+ATAC) sequencing in P4 and P14 BNSTp *Esr1^{Cre/+};Sun1-GFP* females and males, which detected the same BNSTp *Esr1+* cell populations that were previously detected in the adult BNST snRNA-seq dataset (Fig. 4b). Importantly, while we do not detect sex-specific cell populations, there is a significant male-bias in the proportion of two out of seven BNSTp *Esr1+* cell populations (i1:Nfix, i3:Esr2) (original Extended Data Fig. 4.1a, now Fig. 3b) that is also detectable by IF staining (quantified in Fig. 3c), with males having ~1.5-2x more BNSTp ER α /Nfix+ cells than females.

To determine whether a male-bias in these two cell populations is responsible for the observed male-biased genes in the bulk BNSTp *Esr1+* RiboTag dataset, we compared the expression levels of male-biased RiboTag genes across BNSTp *Esr1+* snRNA-seq clusters. We did not detect a significant difference in the expression of RiboTag male-biased DEGs across BNSTp *Esr1+* snRNA-seq clusters (one-way ANOVA, $p = 0.318$) or between male-biased i1:Nfix and i3:Esr2 clusters and non-male-biased BNSTp clusters (i1:Nfix vs. non-male-biased clusters, Mann-Whitney U test, $p = 0.222$; i3:Esr2 vs. non-male-biased clusters, Mann-Whitney U test, $p = 0.320$), indicating that male-biased genes in the bulk RiboTag data are generally not attributable to male-biased populations of BNSTp *Esr1+* cells (see figure below).

2. In Figure 2A, why is padj used for only one class?

Two different p-value thresholds are annotated for the RiboTag experiment, as we identified several genes in the $p < 0.01$ category with significant estradiol-upregulation by in situ hybridization (ISH), such as *Tle3* and *Enah*. Moreover, estradiol-regulated ER α binding sites and ATAC loci were significantly enriched at genes in the $p < 0.01$ category relative to non-differential, expressed genes (Extended Data Fig. 2.3a-b), including *Tle3*, *Enah*, *Col25a1*, *Rbm20*, *Arfgef2*, *Shank2*, and many others (Supplementary Table 1, 3).

3. In Extended Data 2.2B, the authors should provide quantification of the signal to more readily indicate which transcripts show differential regulation in regions other than BNSTp.

This point was also raised by Reviewer 1 (point 3). Our initial intent with this panel was to show that genes regulated by estradiol in BNSTp are not always present in other ER α + brain regions such as the VMHvl and MeA, which suggests specificity of neural ER α action. We now feel that making a conclusive statement about estradiol regulation of gene expression across brain regions is beyond the scope of this paper, given the additional focus on the BNSTp throughout the review process, which includes identifying factors that regulate sex differences in BNSTp gene regulation in adulthood and following the neonatal surge. For that reason, we have decided to remove the panels in Extended Data 2.2b and the corresponding text from the paper.

Figure 3 and Extended Data

1. In Figure 3B, the authors should clarify whether this clustering is performed in female or male mice or in both clusters. In this respect, it would be beneficial to include a UMAP plot that indicates the sex of the clustered samples.

For consistency, the cell labels in the adult BNST snRNA-seq dataset are the same as those used in the original publication¹⁹; therefore, we did not perform any additional unsupervised clustering of the dataset. The clustering in¹⁹ was performed on both female and male mice, and no sex-specific clusters were identified in the original publication. We have now included a UMAP plot indicating the sex of BNSTp *Esr1*+ cells in Fig. 3a.

2. The authors should also tone down their claim that “Across the 40 snRNA-seq clusters, nearly all differentially expressed genes (DEGs) were detected in ER-expressing clusters (Fig. 3a), revealing ER expression is predictive of sex differences in BNST gene expression.” Indeed the authors themselves point out that “other sex hormone receptors, including AR, ER β , and progesterone receptor (PR), were detected in

BNSTp ER+ neurons with varying degrees of cluster specificity, and may also contribute to sex differences in gene expression (Extended Data Fig. 3.1E)". If the authors were to plot the degree of sex-bias in gene expression in AR+, Erβ+, and progesterone receptor (PR+) clusters, would it look different from *Esr1*+ expressing clusters?

We appreciate the reviewer's comment, as it motivated us to adopt a more unbiased approach for examining the relationship between TF expression and sex DEGs. To identify which TFs are most predictive of sex DEG number, we computed the Pearson correlation coefficient between % TF expression (for all 1,721 annotated TFs in the mouse SCENIC database) and number of sex DEGs per cluster across all 40 BNST snRNA-seq clusters. We then ranked TFs by their correlation coefficient. This approach revealed that among all annotated TFs, *Esr1* is most predictive of sex DEG number, in that *Esr1* has the highest correlation coefficient (Fig. 3g). However, both *Ar* and *Pgr* are among the top 10 TFs, thus validating our original claim that these factors are relevant for sex differences. Following our profiling of chromatin accessibility in BNSTp *Esr1*+ neurons of gonadally intact animals (reviewer #2, point #3), we now know that male-biased genes appear to be largely driven by AR binding within *Esr1*+ neurons, whereas E2-responsive loci are maintained in an accessible state by gonadal hormones in both sexes. Given these additional analyses and data, we have modified our original statement:

Esr1 predicted the degree of sex-biased genes better than other TFs in the genome (Fig. 3g) – nearly all sex differences were detected in *Esr1*+ neurons, primarily within the BNSTp, consistent with their central role in the regulation of sex-typical behaviors (Extended Data Fig. 3.2e). Other gonadal steroid hormone receptors, including AR, ERβ, and progesterone receptor (PR), were detected in BNSTp *Esr1*+ neurons with varying degrees of expression (Extended Data Fig. 3.2f).

3. In Extended data 3.1B, the panels are reversed in the figure legend. The authors should also clarify what the different columns in the expression heatmap panel represent.

We have modified the panels in Extended Data Fig. 3.1b (now Extended Data Fig. 3.2a) to now show the expression of all sex DEGs detected in BNSTp *Esr1*+ clusters. Each column in these expression heatmaps represent individual snRNA-seq pseudo-bulk samples (reads aggregated across cells within each cluster per sample). We have now added this information to the Extended Data. Fig. 3.2 legend.

4. For the neonatal EB exposure experiment in Extended data 3.2, how long does the male-gene bias persist after early life exposure to estradiol in females? It is unclear whether the bias in expression truly persists into adulthood given that samples are collected four days after exposure. The authors should perform the neonatal EB exposure and perform RNA-seq at an adult timepoint.

We appreciate the reviewer's request for additional information regarding the persistence of male-biased gene expression following the neonatal hormone surge (see also response to reviewer #1, point #5 and reviewer #2, point #2). Rather than performing additional RNA-seq experiments on neonatal estradiol-treated animals, which we acknowledge is an artificial treatment paradigm designed to mimic a physiological process, we have instead examined the longevity of sex differences in gene expression by performing single-cell multiome sequencing of BNST *Esr1*+ neurons from females and males on P4 and P14. This approach revealed extensive sex DEGs across *Esr1*+ inhibitory neuron clusters on P4; a subset of which persisted until P14 (Fig. 4e, h, Extended Data Fig. 4.4h, Supplementary Table 10). Moreover, snRNA-seq of P14 BNST inhibitory

neurons isolated from male and female *Vgat^{Cre};Esr1^{+/+};Sun1-GFP^{lx}* animals revealed additional sex DEGs shared across P4 and P14, which we attribute to higher cell number and UMI capture in our snRNA-seq dataset than our multiome dataset (Fig. 5b, Extended Data Fig. 5.1, Supplementary Table 11). As these sex DEGs are not present when comparing ER α KO (*Vgat^{Cre};Esr1^{lox/lox};Sun1-GFP^{lx}*) males to control females, we conclude that the persistence of sex differences in neonatal BNST gene expression require ER α (Fig. 5b).

After collecting additional adult chromatin data, we have identified that these loci generally do not persist into adulthood (Extended Data Fig. 4.2a); rather the majority of sites are accessible in BNSTp *Esr1+* neurons across both sexes, which we believe is attributed to the production of estradiol in females during puberty (~P28-35). Instead, we have identified that adult gonadal hormones are the primary driver of adult sex differences in chromatin accessibility, as we detected ~18,000 sex-biased ATAC sites that become feminized in response to adult GDX (Fig. 3h-i). enrichment. We speculate that persistent gene regulatory changes observed at P14 may influence the chromatin response to hormones released during puberty, rather than the same loci being preserved as sex-biased into adulthood.

5. What is the rationale for performing the neonatal exposure in female mice and not male mice? Again, these experimental design choices need to be made very clear to a non-specialist audience. The authors state that 33% of neonatal EB-induced genes retain male-biased expression in adult. Reciprocally, the authors should analyze what percentage of male-biased genes are captured in the neonatal EB-induced gene list as this might give a sense of the relative importance of EB signaling in overall specification of a male-biased gene program.

Thank you for this point. Our rationale for performing the neonatal exposure experiment was because it is a standard approach in the field of neuroendocrinology to recapitulate the endogenous neural estradiol produced in newborn males from transient circulating testosterone. It is well-established that neonatal estradiol treatment of rodent females masculinizes the brain (e.g. cell number in BNSTp) and behavior (reproductive and territorial behaviors)^{63,64}. We now provide additional background and rationale for our neonatal hormone manipulations in the introduction paragraph and in the text related to Fig 4:

Estradiol is the master regulator of rodent brain sexual differentiation. In males, the testes become active shortly after birth, leading to a sharp rise in testosterone which subsides within hours¹². Circulating testosterone is converted by neural aromatase to 17 β -estradiol, which acts through ER α in discrete neuronal populations to specify sex differences in cell number and circuit connectivity^{1,3,13}, enabling the display of sex-typical behaviors in adulthood¹⁴⁻¹⁶.

In addition to cell number dimorphism, neonatal estradiol promotes axonogenesis and synapse formation^{3,43}. Sexual dimorphism in BNSTp wiring emerges during a two-week window following birth, long after neural estradiol has subsided. This raises the possibility that neonatal ER α activation drives a sustained male-specific gene regulatory program. However, genomic targets of the neonatal surge have not been described. To identify this program, we performed ATAC-seq on BNSTp *Esr1+* neurons collected four days after the neonatal hormone surge (P4), which corresponds to the onset of male-biased BNSTp cell survival and axonogenesis^{44,45}. Males were treated with vehicle on the day of birth, while females were either treated with vehicle or estradiol to determine the extent to which sex differences in chromatin state depend on neonatal hormone.

We have now computed the % of P4 male-biased genes (identified in our single-cell multiome data) that are driven by neonatal estradiol and have included this plot in Extended Data Fig. 4.4c.

Figure 4 and Extended Data

1. The choice to focus on inhibitory neuron classes for analysis of the BNSTp is not entirely clear to a non-specialist. Please provide further rationale in the text.

We appreciate the request for additional information regarding the focus on inhibitory neurons. All neurons in the BNSTp are GABAergic^{8,19} – hence our focus on assigning BNSTp *Esr1*+ snRNA-seq clusters to only inhibitory neuron classes. We have now stated this more clearly in the text:

BNSTp *Esr1*+ neurons are GABAergic¹⁹, but the transcriptomic identity of male-biased neurons, and whether they constitute sex-shared or sex-specific populations, remains unknown. To characterize these neurons, we re-analyzed a single-nucleus RNA-seq (snRNA-seq) dataset collected from the BNST of adult females and males³⁵. Among seven BNSTp *Esr1*+ transcriptomic neuron types, we found that inhibitory neurons expressing *Nfix* (i1:Nfix) and *Esr2* (i3:Esr2) are more abundant in males than females (Fig. 3a-b, Extended Data Fig. 3.1a-b). While an increased number of *Esr2*/ER β cells in the BNSTp of males is known³⁶, a role for *Nfix* has not been previously described.

2. In Extended data 4.1D, what is the enrichment of *Nfix* binding sites at female-biased genes? While the St18 cluster does have a larger number of nuclei detected in males compared to females (Extended data 4.1A), there were a similar number of female- and male-biased genes detected in this cluster (Figure 3A).

We did not detect significant enrichment of *Nfix* binding sites at adult BNST female-biased genes in any snRNA-seq cluster. The p-values are listed in the table below.

Cluster label	BETA K-S test p-value
BNSTpr_St18/i1:Nfix	0.838
BNSTp_Tac2/i2:Tac2	0.321
BNSTpr_Esr2/i3:Esr2	1
BNSTp_Bnc2/i4:Bnc2	1
BNSTp_Haus4/i5:Haus4	0.94
BNSTp_Epsti1/i6:Epsti1	0.973
BNSTp_Nxph2/i7:Nxph2	0.971

3. *Nfix* is only expressed in a subset of ER cells as suggested in Figure 4D. The overlap should be quantified.

We have now quantified the male-bias in BNSTp ER α /*Nfix*+ neurons by *Nfix* IF staining in *Esr1*^{Cre/+}; *ROSA26*^{CAG-Sun1-sfGFP-Myc/+} animals (Fig. 3c). We have included the % of BNSTp ER α + neurons co-labeled with *Nfix* in the main text:

We confirmed that ER α /*Nfix*+ neurons, which represent ~30% of the BNSTp *Esr1*+ population, are male-biased using immunofluorescent staining (Fig. 3c, Extended Data Fig. 3.1c).

4. The authors identify ER sites are both pre-bound by Nfix and also sites where ER recruits Nfix. What is the significance of these classes? How do these classes of sites differ at a) male-biased genes b) BNSTp EB-induced genes (defined in Figure 2) and c) neonatal EB-induced genes?

From a gene regulatory standpoint, these two classes represent different modes by which ER α interacts with chromatin. It has previously been shown that ER α , and other steroid hormone receptors, bind low-accessible chromatin regions when motif affinity is high (i.e., when the sequence perfectly matches the consensus motif)^{65,66}. Moreover, there is an inverse relationship between motif strength and recruitment of cooperative TFs^{65,66} – ER α can recruit its binding partner FOXA1 to chromatin when motif affinity is high, thus taking on the classical “pioneer factor” role.

In the original submission, we noticed this same phenomenon when comparing ER α /Nfix recruited to ER α /Nfix pre-bound sites:

ER α /Nfix recruited sites could be distinguished from ER α /Nfix pre-bound sites by lower baseline chromatin accessibility, higher accessibility induction, and stronger EREs (Extended Data Fig. 4.2). These results suggest ER α can recruit Nfix to low-accessible regions through high-affinity motif binding and subsequent chromatin remodeling.

We did examine whether these two classes of sites differed in their enrichment at male-biased genes, BNSTp E2-regulated genes, and neonatal E2-regulated genes (see table below). However, in each case, we did not detect a major difference in the enrichment of these two categories of sites at genes in each list. Therefore, our interpretation was that both modes of ER α /Nfix co-binding contribute to observed transcriptional changes.

DEG list	Nfix/ER α co-binding	BETA K-S test p-value
Adult E2-induced genes	Nfix pre-bound	0.0663
	Nfix co-recruited	0.173
Adult E2-downregulated genes	Nfix pre-bound	0.788
	Nfix co-recruited	0.974
Neonatal E2-induced genes	Nfix pre-bound	0.814
	Nfix co-recruited	0.741
Neonatal E2-downregulated genes	Nfix pre-bound	0.324
	Nfix co-recruited	0.956
i1:Nfix (BNSTpr_St18) male-biased genes	Nfix pre-bound	0.0345
	Nfix co-recruited	0.125
i1:Nfix (BNSTpr_St18) female-biased genes	Nfix pre-bound	0.985
	Nfix co-recruited	0.87

Related to points #5 and #6 below, we have decided to reduce our analysis of the Nfix CUT&RUN dataset in the paper, as we identified extensive sex differences in ATAC sites in gonadally intact animals (Fig. 3h-i) that lack robust overlap with Nfix binding events (~5% of male-biased loci and ~2% of female-biased loci overlap Nfix peaks). Therefore, upon collecting additional data, Nfix does not appear to be a principal regulator of physiological sex differences in gene regulation; rather, it appears most relevant as a cell-identity regulator for a population of BNSTp male-biased *Esr1+* inhibitory neurons (Fig. 3c, Fig. 4c).

5. The authors' analysis and discussion of the role of Nfix in CGE-specification seems tangential in this study and distracts from the main interest in sex specific gene expression programs (Fig. 4g, Extended Data Fig. 4.1).

We agree with this point and have removed our re-analysis of the BICCN snATAC data from the manuscript. We have also toned down our focus on the role of Nfix in CGE-specification. Upon collecting single-cell multiome data from neonatal BNST, and comparing to our Nfix CUT&RUN data, we have identified that Nfix is a putative regulator of the identity of a particular male-biased inhibitory cluster (Fig. 4c), in that 1) among all TFs in the CISBP motif database, *Nfix* has one of strongest correlation coefficient values between motif accessibility and TF expression across BNST *Esr1+* cells (Extended Data Fig. 4.3g), 2) the Nfix motif and Nfix CUT&RUN binding sites have enriched accessibility in the i1:Nfix (formerly labeled as BNSTpr_St18) population (Fig. 4c) and 3) Nfix binds putative enhancers at the *Nfix* locus in i1:Nfix neurons, consistent with an auto-regulatory mechanism (Fig. 4c).

6. Given that Nfix expression is expressed in only a subset of ER+ cells and binds at only 25% of ER binding sites, how are male-biased genes being specified in additional ER+ clusters?

While interesting, the reason to focus on Nfix in this study is not entirely clear. It would be interesting for the authors to attempt to identify the cooperating TF for ER alpha for all male biased clusters and if this is not possible or clear, to state why this might be the case.

The question of how male-biased genes are specified in adult *Esr1+* clusters is indeed critical. We initially profiled Nfix genomic binding in our adult GDX+E2 paradigm, based on our observations that the 1) NFI motif is significantly enriched within ER α binding sites and 2) Nfix is enriched within the BNSTp by ISH and IF. After performing single-cell analysis, we identified 1) Nfix is a marker of a specific male-biased BNSTp *Esr1+* neuron population (Fig. 3c, Extended Data Fig. 3.1 a-b) and 2) Nfix binding defines the chromatin accessibility landscape of this population (Fig. 4c).

The question naturally arose as to whether Nfix contributes to sex differences in gene expression within BNSTpr St18 cells (now re-labeled as i1:Nfix). While only 25% of ER α sites overlapped Nfix sites, these co-bound sites were significantly, and selectively, enriched at genes called as male-biased in these cells (original Extended Data Fig. 4.1d,e), indicating they are functionally relevant. However, as correctly pointed out, male-biased genes are detected across all BNSTp *Esr1+* populations (Supplementary Table 4). Moreover, there was no enrichment of co-bound sites at female-biased genes (see response to point #2). Therefore, Nfix is clearly not required for *all* sex DEGs, and the larger question of how hormones regulate sex differences in gene expression remained unanswered.

To identify the factors regulating sex-biased gene expression in BNSTp *Esr1*+ neurons with an unbiased approach, we performed ATAC-seq in BNSTp *Esr1*+ neurons from gonadally intact females and males. This revealed extensive sex differences in the chromatin landscape (Fig. 3h) with matching enrichment at sex-biased genes detected in *Esr1*+ clusters (i.e., female-biased peaks predominantly occur at female-biased genes, and male-biased peaks occur at male-biased genes). Moreover, by comparing to our GDX ATAC-seq dataset, we discovered that sex differences are largely ablated upon adult GDX (Fig. 3h), with the majority of sex differences being dependent on male rather than female gonadal hormones (Fig. 3i) (see also response to reviewer #2, point #3). Through motif analysis, we identified that male-biased chromatin loci that decrease in accessibility upon GDX are strongly enriched for the androgen response element (ARE) (Fig. 3i). Interestingly, loci that lose accessibility upon GDX in both sexes primarily overlap our E2-open loci and are enriched for the ERE (Fig. 3i), indicating estradiol maintains chromatin accessibility in the brains of both sexes.

We are also interested in the identity of TFs that may bind with ER α across BNST *Esr1*+ clusters, particularly following the neonatal surge (Fig. 4a). Upon performing single-cell multiome sequencing on P4 female and male BNST *Esr1*+ neurons (Fig. 4b), we identified considerable heterogeneity in neonatal male-biased chromatin loci across BNST *Esr1*+ clusters (Fig. 4d), suggesting that ER α binding site selection is strongly influenced by TFs and/or pre-existing chromatin states that differ across neuron types, rather than a singular TF across all *Esr1*+ neurons. In line with this point, we found that male-biased loci enriched in i1:Nfix neurons selectively overlap our previously identified Nfix binding sites (Fig. 4d).

7. The authors should tone down their claim that “Together, these results demonstrate that male-typical behaviors are largely regulated by a population of CGE-derived ER/Nfix+ inhibitory neurons spanning the BNSTpr and SDN-POA” as they have neither perturbed Nfix function nor examined its impact on male-typical behavior in this study.

We agree with this point (see also response to reviewer #3, point #4). We have toned down the conclusion from this section:

Moreover, two of the top marker genes for i1:Nfix neurons, *Moxd1* and *Cplx3* (Extended Data Fig. 3.1b, f-g), were previously identified as markers of a male-biased SDN-POA neuron type (i20:Gal/*Moxd1*) that is selectively activated during male-typical mating, inter-male aggression, and parenting behaviors⁴². Beyond these two marker genes, we found that i1:Nfix and i20:Gal/*Moxd1* neuron types have a shared transcriptomic identity, in line with Nfix immunofluorescence across both the BNSTp and SDN-POA (Fig. 3e, Extended Data Fig. 3.1h). Together, these results define male-biased neurons in the BNSTp and reveal a shared *Lamp5*+ neurogliaform identity between BNSTp ER α +Nfix+ inhibitory neurons and SDN-POA neurons that are engaged during male-typical behaviors.

References

1. Skene, P. J. & Henikoff, S. An efficient targeted nuclease strategy for high-resolution mapping of DNA binding sites. *Elife* **6**, (2017).
2. Skene, P. J., Henikoff, J. G. & Henikoff, S. Targeted in situ genome-wide profiling with high efficiency for low cell numbers. *Nat. Protoc.* **13**, 1006–1019 (2018).
3. Hainer, S. J., Bošković, A., McCannell, K. N., Rando, O. J. & Fazio, T. G. Profiling of Pluripotency Factors in Single Cells and Early Embryos. *Cell* **177**, 1319-1329.e11 (2019).
4. Stroud, H. *et al.* An Activity-Mediated Transition in Transcription in Early Postnatal Neurons. *Neuron* **107**, 874-890.e8 (2020).
5. Allaway, K. C. *et al.* Genetic and epigenetic coordination of cortical interneuron development. *Nature* **597**, 693–697 (2021).
6. Carroll, J. S. *et al.* Chromosome-wide mapping of estrogen receptor binding reveals long-range regulation requiring the forkhead protein FoxA1. *Cell* **122**, 33–43 (2005).
7. Swinstead, E. E. *et al.* Steroid Receptors Reprogram FoxA1 Occupancy through Dynamic Chromatin Transitions. *Cell* **165**, 593–605 (2016).
8. Wu, M. V. & Tollkuhn, J. Estrogen receptor alpha is required in GABAergic, but not glutamatergic, neurons to masculinize behavior. *Horm. Behav.* **95**, 3–12 (2017).
9. Glont, S.-E. *et al.* Identification of ChIP-seq and RIME grade antibodies for Estrogen Receptor alpha. *PLoS One* **14**, e0215340 (2019).
10. Franco, H. L., Nagari, A. & Kraus, W. L. TNF α signaling exposes latent estrogen receptor binding sites to alter the breast cancer cell transcriptome. *Mol. Cell* **58**, 21–34 (2015).
11. Nelson, A. W. *et al.* Comprehensive assessment of estrogen receptor beta antibodies in cancer cell line models and tissue reveals critical limitations in reagent specificity. *Mol. Cell. Endocrinol.* **440**, 138–150 (2017).

12. Clarkson, J. & Herbison, A. E. Hypothalamic control of the male neonatal testosterone surge. *Philos. Trans. R. Soc. Lond. B Biol. Sci.* **371**, 20150115 (2016).
13. Holding, A. N., Cullen, A. E. & Markowitz, F. Genome-wide Estrogen Receptor- α activation is sustained, not cyclical. *Elife* **7**, (2018).
14. Shang, Y. & Brown, M. Molecular determinants for the tissue specificity of SERMs. *Science* **295**, 2465–2468 (2002).
15. Zhang, Z. *et al.* Estrogen receptor alpha in the brain mediates tamoxifen-induced changes in physiology in mice. *Elife* **10**, (2021).
16. Wade, G. N., Blaustein, J. D., Gray, J. M. & Meredith, J. M. ICI 182,780: a pure antiestrogen that affects behaviors and energy balance in rats without acting in the brain. *Am. J. Physiol.* **265**, R1392-8 (1993).
17. Ma, S. *et al.* Chromatin Potential Identified by Shared Single-Cell Profiling of RNA and Chromatin. *Cell* **183**, 1103-1116.e20 (2020).
18. Stuart, T. *et al.* Comprehensive Integration of Single-Cell Data. *Cell* **177**, 1888-1902.e21 (2019).
19. Welch, J. D. *et al.* Single-Cell Multi-omic Integration Compares and Contrasts Features of Brain Cell Identity. *Cell* **177**, 1873-1887.e17 (2019).
20. Granja, J. M. *et al.* ArchR is a scalable software package for integrative single-cell chromatin accessibility analysis. *Nat. Genet.* **53**, 403–411 (2021).
21. Fang, R. *et al.* Comprehensive analysis of single cell ATAC-seq data with SnapATAC. *Nat. Commun.* **12**, 1337 (2021).
22. Korsunsky, I. *et al.* Fast, sensitive and accurate integration of single-cell data with Harmony. *Nat. Methods* **16**, 1289–1296 (2019).
23. Yao, Z. *et al.* An integrated transcriptomic and epigenomic atlas of mouse primary motor cortex cell types. *Cold Spring Harbor Laboratory* 2020.02.29.970558 (2020) doi:10.1101/2020.02.29.970558.
24. Fischer, S., Crow, M., Harris, B. D. & Gillis, J. Scaling up reproducible research for single-cell transcriptomics using MetaNeighbor. *Nat. Protoc.* **16**, 4031–4067 (2021).
25. Wang, J. *et al.* Tracing cell-type evolution by cross-species comparison of cell atlases. *Cell Rep.* **34**,

- 108803 (2021).
26. Crow, M., Paul, A., Ballouz, S., Huang, Z. J. & Gillis, J. Characterizing the replicability of cell types defined by single cell RNA-sequencing data using MetaNeighbor. *Nat. Commun.* **9**, 884 (2018).
 27. Cordingley, M. G., Riegel, A. T. & Hager, G. L. Steroid-dependent interaction of transcription factors with the inducible promoter of mouse mammary tumor virus in vivo. *Cell* **48**, 261–270 (1987).
 28. Archer, T. K., Lefebvre, P., Wolford, R. G. & Hager, G. L. Transcription factor loading on the MMTV promoter: a bimodal mechanism for promoter activation. *Science* **255**, 1573–1576 (1992).
 29. Hebbar, P. B. & Archer, T. K. Chromatin-dependent cooperativity between site-specific transcription factors in vivo. *J. Biol. Chem.* **282**, 8284–8291 (2007).
 30. Di Croce, L. *et al.* Two-step synergism between the progesterone receptor and the DNA-binding domain of nuclear factor 1 on MMTV minichromosomes. *Mol. Cell* **4**, 45–54 (1999).
 31. Vicent, G. P. *et al.* Nuclear factor 1 synergizes with progesterone receptor on the mouse mammary tumor virus promoter wrapped around a histone H3/H4 tetramer by facilitating access to the central hormone-responsive elements. *J. Biol. Chem.* **285**, 2622–2631 (2010).
 32. Grabowska, M. M. *et al.* NFI transcription factors interact with FOXA1 to regulate prostate-specific gene expression. *Mol. Endocrinol.* **28**, 949–964 (2014).
 33. Grabowska, M. M. *et al.* Nfib Regulates Transcriptional Networks That Control the Development of Prostatic Hyperplasia. *Endocrinology* **157**, 1094–1109 (2016).
 34. Pooley, J. R. *et al.* Genome-wide identification of basic helix-loop-helix and NF-1 motifs underlying GR binding sites in male rat hippocampus. *Endocrinology* **158**, 1486–1501 (2017).
 35. Paul, A. *et al.* Transcriptional Architecture of Synaptic Communication Delineates GABAergic Neuron Identity. *Cell* **171**, 522–539.e20 (2017).
 36. Yao, Z. *et al.* A taxonomy of transcriptomic cell types across the isocortex and hippocampal formation. *Cold Spring Harbor Laboratory* 2020.03.30.015214 (2020) doi:10.1101/2020.03.30.015214.
 37. Cooke, B. M. & Simerly, R. B. Ontogeny of bidirectional connections between the medial nucleus of the amygdala and the principal bed nucleus of the stria terminalis in the rat. *J. Comp. Neurol.* **489**, 42–58

- (2005).
38. Gotsiridze, T., Kang, N., Jacob, D. & Forger, N. G. Development of sex differences in the principal nucleus of the bed nucleus of the stria terminalis of mice: role of Bax-dependent cell death. *Dev. Neurobiol.* **67**, 355–362 (2007).
 39. Weissman, M. M. & Klerman, G. L. Sex differences and the epidemiology of depression. *Arch. Gen. Psychiatry* **34**, 98–111 (1977).
 40. Goldstein, J. M. & Link, B. G. Gender and the expression of schizophrenia. *J. Psychiatr. Res.* **22**, 141–155 (1988).
 41. Walf, A. A. & Frye, C. A. A review and update of mechanisms of estrogen in the hippocampus and amygdala for anxiety and depression behavior. *Neuropsychopharmacology* **31**, 1097–1111 (2006).
 42. Sherwin, B. B. & Gelfand, M. M. Sex steroids and affect in the surgical menopause: a double-blind, cross-over study. *Psychoneuroendocrinology* **10**, 325–335 (1985).
 43. Kulkarni, J. *et al.* Estrogen - a potential treatment for schizophrenia. *Schizophr. Res.* **48**, 137–144 (2001).
 44. Soares, C. N., Almeida, O. P., Joffe, H. & Cohen, L. S. Efficacy of estradiol for the treatment of depressive disorders in perimenopausal women: a double-blind, randomized, placebo-controlled trial. *Arch. Gen. Psychiatry* **58**, 529–534 (2001).
 45. Kulkarni, J. *et al.* Estrogen in severe mental illness: a potential new treatment approach. *Arch. Gen. Psychiatry* **65**, 955–960 (2008).
 46. Baron-Cohen, S. *et al.* Foetal oestrogens and autism. *Mol. Psychiatry* **25**, 2970–2978 (2020).
 47. Willsey, H. R. *et al.* Parallel in vivo analysis of large-effect autism genes implicates cortical neurogenesis and estrogen in risk and resilience. *Neuron* **109**, 788–804.e8 (2021).
 48. Moffitt, J. R. *et al.* Molecular, spatial, and functional single-cell profiling of the hypothalamic preoptic region. *Science* **362**, (2018).
 49. Gurney, M. E. & Konishi, M. Hormone-induced sexual differentiation of brain and behavior in zebra finches. *Science* **208**, 1380–1383 (1980).
 50. Tobet, S. A., Zahniser, D. J. & Baum, M. J. Differentiation in male ferrets of a sexually dimorphic nucleus

- of the preoptic/anterior hypothalamic area requires prenatal estrogen. *Neuroendocrinology* **44**, 299–308 (1986).
51. McEwen, B. S. Neural gonadal steroid actions. *Science* **211**, 1303–1311 (1981).
 52. Shang, Y., Hu, X., DiRenzo, J., Lazar, M. A. & Brown, M. Cofactor dynamics and sufficiency in estrogen receptor-regulated transcription. *Cell* **103**, 843–852 (2000).
 53. Xu, X. *et al.* Modular Genetic Control of Sexually Dimorphic Behaviors. *Cell* **148**, 596–607 (2012).
 54. Gould, E., Woolley, C. S., Frankfurt, M. & McEwen, B. S. Gonadal steroids regulate dendritic spine density in hippocampal pyramidal cells in adulthood. *J. Neurosci.* **10**, 1286–1291 (1990).
 55. de Castilhos, J., Forti, C. D., Achaval, M. & Rasia-Filho, A. A. Dendritic spine density of posterodorsal medial amygdala neurons can be affected by gonadectomy and sex steroid manipulations in adult rats: a Golgi study. *Brain Res.* **1240**, 73–81 (2008).
 56. Mukamel, E. A. Multiple Comparisons and Inappropriate Statistical Testing Lead to Spurious Sex Differences in Gene Expression. *Biol. Psychiatry* (2021) doi:10.1016/j.biopsych.2021.06.026.
 57. Xu, Y. *et al.* ER α is an RNA-binding protein sustaining tumor cell survival and drug resistance. *Cell* (2021) doi:10.1016/j.cell.2021.08.036.
 58. Björnström, L. & Sjöberg, M. Mechanisms of estrogen receptor signaling: convergence of genomic and nongenomic actions on target genes. *Mol. Endocrinol.* **19**, 833–842 (2005).
 59. Stender, J. D. *et al.* Genome-wide analysis of estrogen receptor alpha DNA binding and tethering mechanisms identifies Runx1 as a novel tethering factor in receptor-mediated transcriptional activation. *Mol. Cell. Biol.* **30**, 3943–3955 (2010).
 60. Heldring, N. *et al.* Multiple sequence-specific DNA-binding proteins mediate estrogen receptor signaling through a tethering pathway. *Mol. Endocrinol.* **25**, 564–574 (2011).
 61. Cvoro, A. *et al.* Distinct roles of unliganded and liganded estrogen receptors in transcriptional repression. *Mol. Cell* **21**, 555–564 (2006).
 62. Grober, O. M. V. *et al.* Global analysis of estrogen receptor beta binding to breast cancer cell genome reveals an extensive interplay with estrogen receptor alpha for target gene regulation. *BMC Genomics* **12**,

36 (2011).

63. Bronson, F. H. & Desjardins, C. Aggression in adult mice: modification by neonatal injections of gonadal hormones. *Science* **161**, 705–706 (1968).
64. Wu, M. V. *et al.* Estrogen masculinizes neural pathways and sex-specific behaviors. *Cell* **139**, 61–72 (2009).
65. Gertz, J. *et al.* Distinct properties of cell-type-specific and shared transcription factor binding sites. *Mol. Cell* **52**, 25–36 (2013).
66. Paakinaho, V., Swinstead, E. E., Presman, D. M., Grøntved, L. & Hager, G. L. Meta-analysis of Chromatin Programming by Steroid Receptors. *Cell Rep.* **28**, 3523-3534.e2 (2019).

Reviewer Reports on the First Revision:

Referees' comments:

Referee #1 (Remarks to the Author):

The authors have satisfactorily addressed most of the issues that I raised in my initial review. They have also modified somewhat their conclusions as a result of their new experiment on ATAC-seq in BNSTp ER α neurons in gonadally intact adults. I think the most important observation in the manuscript now is that the male-biased transcriptional program established in neonatal animals does not persist into adults, but only until P14. That result will surprise many people who have assumed that changes in open chromatin established at birth by ER α should persist into adulthood.

The authors seem a bit defensive about this result in their rebuttal (“we stand by our initial claim that neonatal activation of ER α coordinates a lasting male-biased transcriptional program”). While semantically accurate, the meaning of “lasting” has now changed relative to the initial submission. Rather than digging in with this message while simply acknowledging “that puberty represents an additional critical window,” I would urge the authors to embrace their new findings and to emphasize them more strongly in the Summary and the Discussion. E.g., rather than saying in the Summary that “ER α directs...activation of a sustained male-biased gene expression program,” where the use of the word “sustained” is technically accurate but vague regarding duration, I would suggest they state more explicitly that the male-biased gene expression program only persists until P14, and that the transcriptional program in the adult is substantially different from what is observed in neonatal animals.

I also appreciate the authors' inclusion of data from genetic deletion of ER α in BNSTp, which is a demanding but powerful experiment. Finally, I think the authors' revised view regarding the role of Nfix, from ER α co-factor regulating male-biased transcription to determinant of neuronal subtype identity, is important and deserves emphasis as well.

Referee #2 (Remarks to the Author):

The authors have added a number of additional experiments to address reviewer concerns. I remain in awe of what they have achieved technically: this study takes the field forward in leaps and bounds technique-wise. However, I am less confident of the conclusions conceptually. A vast quantity of data is presented, but the overall story is not clear. The difficulty in seeing a clear story line is partly related to the amount of data presented in the space allowed, but also may be related to difficulties in communication. I also have some quibbles with the interpretation of several experiments, as detailed below.

The final sentence of many of the paragraphs overstates what was actually shown. For example, “These results demonstrate that estrogen receptor genomic binding, rather than cell membrane-initiated ER signaling or presynaptic firing of BNST-projecting Esr1+ neurons, drives estradiol

transcription regulation...” is too strong since you didn’t directly look at either membrane-initiated or presynaptic effects. “Our results show that E2 induces expression of genes involved in axonogenesis and synaptic organization that we predict are important for early-life sexual differentiation of BNSTp circuitry,” seems like a strange conclusion for a study strictly involving adult animals and responses due to activational effects of E2. A few other examples are indicated below, some of which could be resolved by replacing “reveal that” or “demonstrate that” with “suggest that.”

“These genes [306 genes with a differential estradiol response between sexes] lacked enrichment of E2-responsive ATAC peaks, suggesting additional regulation by estradiol may occur at the translational level.” This observation points to a potential problem. There are over 7,200 chromatin regions that increase accessibility with E2 treatment and about 2,000 sites identified where ER “sits down” on the DNA, but only 306 genes that are differentially expressed in response to EB, and those do not show enrichment of E2-responsive ATAC peaks. What do the data from the ATAC and CUT&RUN approaches really tell us then? It seems that something should be said to address the order-of-magnitude differences between the different approaches.

The section, “Sex differences in gene regulation are defined by gonadal hormones” concludes: “Collectively, our findings demonstrate that sex hormone receptors drive adult sex differences in gene expression, and that these sex differences are a consequence of acute [i.e., adult, my addition] hormonal state.” However, the conclusion of the very next paragraph (and, reflected in the title of this section) is: “Together, these results demonstrate that neonatal ER activation drives sustained sex differences in the chromatin landscape.” This is contradictory. If effects of neonatal ER were sustained, there would not be a requirement for the animals to be gonadally intact in adulthood to see them. Indeed, further down you state, “...these results reveal that adult sex differences in chromatin accessibility largely derive from gonadal hormones released following puberty.” I suggest that you delete the word “sustained” throughout this paragraph.

The following four comments pertain to the section, “ER organizes a sustained gene expression program of brain sexual differentiation:”

I could not make sense of the second paragraph in this section. (e.g., “TFs with highly correlated RNA expression and motif accessibility...”)

“Together these results reveal neuron identity TFs can influence patterns of ER genomic recruitment, thereby expanding the cellular response to a common signaling event.” I don’t think your experiment allows you to conclude this. The TFs you identified correlate with different responses, but you don’t know that those TFs are what caused the different patterns of ER genomic recruitment.

“Notably, inhibitory neurons lacking Cyp19a1/aromatase expression, such as i2:Tac2 neurons, displayed NE-open peaks as well as sex-biased genes, revealing neural estradiol controls gene expression via autocrine and paracrine activation of ER.” How do you know that the activation in cells lacking aromatase was via estradiol? Couldn’t it be via synaptic activity from ER-containing cells, or a factor other than estradiol released by ER-containing cells? Also, “autocrine” means that a cell

secretes a hormone that then binds to receptors in/on that same cell. Is that really what you mean?

The data do not seem to support a “sustained gene expression program.” In Figure 4h, it appears that the overwhelming majority of sex-biased genes at P4 are no longer sex biased at P14 (a 84-99% decline, depending on cell sub-type). This indicates that the large majority of the neonatal response to estradiol is not sustained.

“VgatCre; Esr1^{lxlx} males had feminized abundance of male-biased i1;Nfix and i3:Esr2 neurons, suggesting neonatal ER activation promotes the survival of these two neuron types.” This seems like a bridge too far. How do you know that ER activation promotes the survival, versus promoting the differentiation, of those cell types (e.g., at the expense of differentiation of Esr1/Tac2 cells, which are lower in males)?

The remaining comments relate to relatively minor issues of clarity:

Last paragraph in the section “Estradiol induces a gene regulatory program....” It would help the reader if instead of referring to “the TRAP data,” “across ER CUT&RUN and ATAC-seq modalities,” you reminded the reader of what these mean (e.g., for gene expression as assessed by TRAP, etc...)

“BNSTp Esr1⁺ neurons are GABAergic, but the transcriptomic identity of male-biased neurons, and whether they constitute sex-shared or sex-specific populations, remains unknown. To characterize these neurons...” It’s not clear what you mean by “male-biased neurons” here, or why you are assuming at this point in the study that there is such a thing.

“Esr1 predicted the degree of sex-biased genes better than other TFs in the genome - nearly all sex differences were detected in Esr1⁺ neurons, primarily within the BNSTp, consistent with their central role in the regulation of sex-typical behaviors.” The phrase “primarily with the BNSTp” throws me here, because I thought your whole analysis was specific to the BNSTp. Are you making a distinction between the BNST and BNSTp? If so, please make this nuance clear.

In the statement, “Only a small proportion of neonatal estradiol-regulated sites (~10%) maintained corresponding female- or male-biased accessibility in adulthood,” please specify whether the comparison is with GDX or intact adults.

Referee #3 (Remarks to the Author):

The authors have done a nice job at responding to the many many comments from the reviewers. I am particularly sympathetic to the challenges of working with newborns and the inability to exploit many of the tools used in the adult to manipulate specific genes, such as lentivirus etc., as the time is too short. The authors inclusion of additional data using ATAC-Seq and CUT and RUN is sufficient to provide high confidence in their conclusions regarding the role of Er-alpha

To this reviewer there is only one remaining concern which is the authors portrayal of the androgen surge as a purely postnatal event. They cite a study by Herbison which does measure testosterone

postnatally but no other times. Unfortunately that article also includes a schematic depicting androgen levels across development but it is incorrect, the levels after birth are not as high as those prenatally. There is an older and long established literature that the greatest surge in testosterone occurs prenatally, with the peak levels in males being at embryonic day 18 in the rat, demonstrated in an iconic 1980 paper by Weisz and Ward in *Endocrinology* 106:306. The same has been shown for the mouse, and this is also why giving an ER-antagonist post-natally often has no effect because the process of steroid mediated differentiation is already well on its way. There is nothing wrong with the design or interpretation of the current study but the notion that there is only a postnatal surge (more like a small wave) should not be propagated

Referee #4 (Remarks to the Author):

The authors have made a substantial effort to address the reviewers' suggestions and concerns. I think the paper is considerably improved, reports important finding of general interest and is appropriate for publication in *Nature*. My one concern is that the paper is still tough to get through and is not effectively written for a general audience. I suggest the authors get input from colleagues outside the field to help make this terrific study more accessible.

Author Rebuttals to First Revision:

We appreciate the comments of the Referees, and address their remaining issues below. As requested, we have substantially streamlined our manuscript to make it more accessible for a general audience. We have also modified or softened several conclusions.

Referee #1 (Remarks to the Author):

The authors have satisfactorily addressed most of the issues that I raised in my initial review. They have also modified somewhat their conclusions as a result of their new experiment on ATAC-seq in BNSTp ER α neurons in gonadally intact adults. I think the most important observation in the manuscript now is that the male-biased transcriptional program established in neonatal animals does not persist into adults, but only until P14. That result will surprise many people who have assumed that changes in open chromatin established at birth by ER α should persist into adulthood.

The authors seem a bit defensive about this result in their rebuttal (“we stand by our initial claim that neonatal activation of ER α coordinates a lasting male-biased transcriptional program”). While semantically accurate, the meaning of “lasting” has now changed relative to the initial submission. Rather than digging in with this message while simply acknowledging “that puberty represents an additional critical window,” I would urge the authors to embrace their new findings and to emphasize them more strongly in the Summary and the Discussion. E.g., rather than saying in the Summary that “ER α directs...activation of a sustained male-biased gene expression program,” where the use of the word “sustained” is technically accurate but vague regarding duration, I would suggest they state more explicitly that the male-biased gene expression program only persists until P14, and that the transcriptional program in the adult is substantially different from what is observed in neonatal animals.

We appreciate the reviewer’s comment. In fact, several of the genes that are male-biased at P14 are also male-biased in adulthood (*Col25a1*, *Etl4*, *Kcnab1*, *Oxr1*, *Sox5*, many others), suggesting certain genes do persist as sex-biased in *expression* throughout life. However, it is apparent that the *chromatin loci* controlling these sex differences change at puberty, as thousands of sex-biased regions emerge as a result of gonadal hormone production. As stated, these findings suggest: “... certain male-biased genes undergo sequential regulation by ER α and AR in early life and adulthood, respectively.” Therefore, characterizing the functional interplay between ER α and AR will be crucial to understanding whether and how genomic changes at birth influence the response to subsequent hormone exposure during puberty. We now end our Discussion by highlighting the importance of puberty and stating that our work provides an archetype for understanding hormone receptor action across life stages, brain regions, and species.

I also appreciate the authors’ inclusion of data from genetic deletion of ER α in BNSTp, which is a demanding but powerful experiment. Finally, I think the authors’ revised view regarding the role of Nfix, from ER α co-factor regulating male-biased transcription to determinant of neuronal subtype identity, is important and deserves emphasis as well.

Thank you for this comment.

Referee #2 (Remarks to the Author):

The authors have added a number of additional experiments to address reviewer concerns. I remain in awe of what they have achieved technically: this study takes the field forward in leaps and bounds technique-wise. However, I am less confident of the conclusions conceptually. A vast quantity of data is presented, but the overall story is not clear. The difficulty in seeing a clear story line is partly related to the amount of data presented in the space allowed, but also may be related to difficulties in communication. I also have some quibbles with the interpretation of several experiments, as detailed below.

The final sentence of many of the paragraphs overstates what was actually shown. For example, “These results demonstrate that estrogen receptor genomic binding, rather than cell membrane-initiated ER signaling or presynaptic firing of BNST-projecting Esr1+ neurons, drives estradiol transcription regulation...” is too strong since you didn’t directly look at either membrane-initiated or presynaptic effects. “Our results show that E2 induces expression of genes involved in axonogenesis and synaptic organization that we predict are important for early-life sexual differentiation of BNSTp circuitry,” seems like a strange conclusion for a study strictly involving adult animals and responses due to activational effects of E2. A few other examples are indicated below, some of which could be resolved by replacing “reveal that” or “demonstrate that” with “suggest that.”

We appreciate these comments. We have systematically revised the text (including the examples mentioned here) to soften and/or eliminate overstating sentences, allowing readers to arrive at their own conclusions based on the data.

“These genes [306 genes with a differential estradiol response between sexes] lacked enrichment of E2-responsive ATAC peaks, suggesting additional regulation by estradiol may occur at the translational level.” This observation points to a potential problem. There are over 7,200 chromatin regions that increase accessibility with E2 treatment and about 2,000 sites identified where ER “sits down” on the DNA, but only 306 genes that are differentially expressed in response to EB, and those do not show enrichment of E2-responsive ATAC peaks. What do the data from the ATAC and CUT&RUN approaches really tell us then? It seems that something should be said to address the order-of-magnitude differences between the different approaches.

Thank you for raising this important point. We believe this comment has confused certain results in the paper; 358 genes were differentially expressed in response to estradiol (E2/EB) across sexes (Fig. 1e), whereas 306 genes responded differently to estradiol *between sexes*. The 358 genes that change expression across sexes are indeed enriched for E2-open chromatin regions and ER α binding sites, indicating that TF binding/enhancer activation contributes to their regulation. The 306 genes that differentially respond between sexes are not enriched for E2-responsive loci, suggesting a non-genomic mechanism regulates these genes.

The order-of-magnitude difference between differential gene expression and chromatin accessibility/TF binding is not unique to the datasets generated in this paper (see Hurtado et al., 2011, *Nature Genetics* as one example involving ER α); rather, it reflects general properties of gene regulation. The simplest explanation is that multiple E2-responsive loci can regulate

a single gene. In fact, most E2-induced genes have more than one proximal E2-open chromatin region - an extreme example of this is *Pgr*, which has 13 (!) E2-open chromatin regions within 300Kb of the TSS. Another explanation is that not every TF binding event is sufficient to alter transcription of the neighboring gene. Often, the number of TF binding sites, duration of TF binding, and presence of additional TF binding partners influences whether a TF alters transcription. A third explanation is that transcription occurs dynamically across time, whereas our RNA-seq experiment captured only a single snapshot at 4 hr post-treatment. Therefore, it is possible that some genes turn on and return to baseline prior to 4 hr, while others do not turn on until after 4 hr.

Regardless of the explanation, chromatin data (i.e., ATAC-seq, CUT&RUN) remain essential to the field for identifying mechanisms of gene regulation. Without these data, we would not know which factors and enhancers mediate the effects of estradiol on gene expression. Our data reinforce central principles of hormone receptor action previously identified *in vitro* and provide a framework for future exploration of hormonal regulation of gene expression in the brain.

The section, “Sex differences in gene regulation are defined by gonadal hormones” concludes: “Collectively, our findings demonstrate that sex hormone receptors drive adult sex differences in gene expression, and that these sex differences are a consequence of acute [i.e., adult, my addition] hormonal state.” However, the conclusion of the very next paragraph (and, reflected in the title of this section) is: “Together, these results demonstrate that neonatal ER activation drives sustained sex differences in the chromatin landscape.” This is contradictory. If effects of neonatal ER were sustained, there would not be a requirement for the animals to be gonadally intact in adulthood to see them. Indeed, further down you state, “...these results reveal that adult sex differences in chromatin accessibility largely derive from gonadal hormones released following puberty.” I suggest that you delete the word “sustained” throughout this paragraph.

As suggested, we have removed “sustained” from this paragraph to prevent confusion.

The following four comments pertain to the section, “ER organizes a sustained gene expression program of brain sexual differentiation:”

I could not make sense of the second paragraph in this section. (e.g., “TFs with highly correlated RNA expression and motif accessibility...”)

Thank you. We have re-written this paragraph to improve readability.

“Together these results reveal neuron identity TFs can influence patterns of ER genomic recruitment, thereby expanding the cellular response to a common signaling event.” I don’t think your experiment allows you to conclude this. The TFs you identified correlate with different responses, but you don’t know that those TFs are what caused the different patterns of ER genomic recruitment.

We agree with this comment and have softened this conclusion accordingly.

“Notably, inhibitory neurons lacking *Cyp19a1*/aromatase expression, such as *i2: Tac2* neurons, displayed NE-open peaks as well as sex-biased genes, revealing neural estradiol controls gene expression via autocrine and paracrine activation of ER.” How do you know that the activation in cells lacking aromatase was via estradiol? Couldn't it be via synaptic activity from ER-containing cells, or a factor other than estradiol released by ER-containing cells? Also, “autocrine” means that a cell secretes a hormone that then binds to receptors in/on that same cell. Is that really what you mean?

The neonatal estradiol (NE)-open ATAC peaks have elevated accessibility in neonatal vehicle-treated males and neonatal estradiol-treated females compared to neonatal vehicle-treated females. Importantly, ER α also binds the majority of these peaks. Therefore, the fact that these peaks are male-biased within cells that lack aromatase expression indicates that estradiol acts within cells that do not synthesize estradiol. We find this to be strong and novel evidence of non-cell-autonomous, or paracrine, estradiol signaling, as it is only possible to detect with single-cell multiome data.

The data do not seem to support a “sustained gene expression program.” In Figure 4h, it appears that the overwhelming majority of sex-biased genes at P4 are no longer sex biased at P14 (a 84-99% decline, depending on cell sub-type). This indicates that the large majority of the neonatal response to estradiol is not sustained.

Thank you. We agree that the number of genes that persist as sex-biased decreases over developmental time, which we state in the text (“While the total number of sex-biased genes declined between P4 and P14...”). However, certain genes do persist as sex-biased; their function does not seem to be random but rather relates to the cellular changes that are known to occur during brain sexual differentiation (i.e., axonogenesis, axon pathfinding, synapse formation). Hence, the neonatal surge drives sustained sex-biased expression of certain genes that may facilitate sexual differentiation of BNSTp circuitry. Notably, the neuron type with the highest number of sustained sex-biased genes (*i1:Nfix*) is also more abundant in males than in females, suggesting that some of these genes may drive male-biased cell survival.

Moreover, we also believe that the exact number of sex-biased genes detected at P14 is influenced by statistical power, as we observed a higher proportion of genes sustained as sex-biased in our P14 snRNA-seq dataset, which contains more cells and higher transcripts per cell, than in our P14 multiome experiment. Minor discrepancies in cell and transcript recovery between snRNA-seq and multiome reflect current technical and cost limitations of the multiome approach, which has only recently been released commercially.

“*VgatCre; Esr1^{lox}* males had feminized abundance of male-biased *i1:Nfix* and *i3:Esr2* neurons, suggesting neonatal ER activation promotes the survival of these two neuron types.” This seems like a bridge too far. How do you know that ER activation promotes the survival, versus promoting the differentiation, of those cell types (e.g., at the expense of differentiation of *Esr1/Tac2* cells, which are lower in males)?

We agree that we cannot exclude the possibility that ER α KO causes the identity of the neurons to change from *i1:Nfix* and *i3:Esr2* to a different inhibitory neuron type. It is unlikely that this is the case, as ER α was not predicted to be an identity regulator TF in our multiome

dataset. However, our P14 snRNA-seq experiment only permits us to examine whether a new identity forms in the absence of ER α , not a shift to an already existing Esr1+ identity. For this reason, we have removed “survival” from the conclusion of this section.

The remaining comments relate to relatively minor issues of clarity:

Last paragraph in the section “Estradiol induces a gene regulatory program....” It would help the reader if instead of referring to “the TRAP data,” “across ER CUT&RUN and ATAC-seq modalities,” you reminded the reader of what these mean (e.g., for gene expression as assessed by TRAP, etc...)

We have made this change.

“BNSTp Esr1+ neurons are GABAergic, but the transcriptomic identity of male-biased neurons, and whether they constitute sex-shared or sex-specific populations, remains unknown. To characterize these neurons...” It’s not clear what you mean by “male-biased neurons” here, or why you are assuming at this point in the study that there is such a thing.

Thank you. We now clearly state that the BNSTp is known to contain more cells in males than in females due to the neonatal hormone surge promoting male-specific cell survival.

“Esr1 predicted the degree of sex-biased genes better than other TFs in the genome - nearly all sex differences were detected in Esr1+ neurons, primarily within the BNSTp, consistent with their central role in the regulation of sex-typical behaviors.” The phrase “primarily with the BNSTp” throws me here, because I thought your whole analysis was specific to the BNSTp. Are you making a distinction between the BNST and BNSTp? If so, please make this nuance clear.

We have removed this statement to avoid confusion. Indeed, we performed differential expression analysis within each cell type throughout the entire BNST, which has additional subregions. Neuron types annotated to the BNSTp contained a higher number of sex-biased genes than types annotated to the anterior region (Extended Data Fig. 2.2e).

In the statement, “Only a small proportion of neonatal estradiol-regulated sites (~10%) maintained corresponding female- or male-biased accessibility in adulthood,” please specify whether the comparison is with GDX or intact adults.

We now specify that this comparison is made with gonadally intact adults.

Referee #3 (Remarks to the Author):

The authors have done a nice job at responding to the many many comments from the reviewers. I am particularly sympathetic to the challenges of working with newborns and the inability to exploit many of the tools used in the adult to manipulate specific genes, such as lentivirus etc., as the time is too short. The authors inclusion of additional data using ATAC-Seq and CUT and RUN is sufficient to provide high confidence in their conclusions regarding the role of Er-alpha

To this reviewer there is only one remaining concern which is the authors portrayal of the androgen surge as a purely postnatal event. They cite a study by Herbison which does measure testosterone postnatally but no other times. Unfortunately that article also includes a schematic depicting androgen levels across development but it is incorrect, the levels after birth are not as high as those prenatally. There is an older and long established literature that the greatest surge in testosterone occurs prenatally, with the peak levels in males being at embryonic day 18 in the rat, demonstrated in an iconic 1980 paper by Weisz and Ward in *Endocrinology* 106:306. The same has been shown for the mouse, and this is also why giving an ER-antagonist post-natally often has no effect because the process of steroid mediated differentiation is already well on its way. There is nothing wrong with the design or interpretation of the current study but the notion that there is only a postnatal surge (more like a small wave) should not be propagated

We agree that there is conclusive literature on the presence of a prenatal surge in rats. In addition to Weisz and Ward, 1980, *Endocrinology*, McAbee and DonCarlos 1998 showed that a male-bias in BNSTp *Ar* expression emerges between E20 and P0 in rats, indicative of prenatal ER α activation. We also do not disagree that there may be prenatal testosterone production in mice, based on indirect observations from 0M and 2M E17 female fetuses in vom Saal and Bronson, 1980, *Science*. However, the prenatal surge is poorly characterized in mice compared to rats. Despite our best efforts, we have not been able to locate a paper directly examining testosterone in serum or plasma of fetal mice. Besides the Herbison paper, there is an older study (Motelica-Heino et al., 1987, *Physiology & Behavior*) measuring plasma testosterone levels in spontaneously-born mice and mice harvested by Caesarian section (C-newborns). C-newborn mice had low T immediately upon removal from the uterine horn; however, remarkably, they had a surge in T between 30 min and 2 hr after removal, similar to newborn mice. Moreover, the level of T detected at birth was comparable to that of an adult male mouse, indicating the surge generates a considerable amount of hormone, as opposed to a small wave.

Regardless of whether embryonic hormone production occurs in mice, there are countless studies using P0 orchidectomy to conclusively demonstrate that the neonatal hormone surge controls sexual differentiation of the brain and behavior. These studies include: Edwards, 1969, *Physiology & Behavior*; Corbier et al., 1983, *Physiology & Behavior*; Roffi et al., 1987, *Physiology & Behavior*; Matuszcyk et al., 1988, *Hormones and Behavior*; Motelica-Heino et al., 1993, *Physiology & Behavior*; Davis et al., 1996, *Brain Research*; Gu et al., 2003, *J Comp Neurol*; Polston et al., 2004, *Neuroscience*. For this reason, our schematic solely portrays the P0 surge, as it is necessary for brain sexual differentiation to occur.

Referee #4 (Remarks to the Author):

The authors have made a substantial effort to address the reviewers' suggestions and concerns. I think the paper is considerably improved, reports important finding of general interest and is appropriate for publication in *Nature*. My one concern is that the paper is still tough to get through and is not effectively written for a general audience. I suggest the authors get input from colleagues outside the field to help make this terrific study more accessible.

We appreciate this comment. We have sought advice from colleagues and revised the text to improve readability.